# Tensor Product Attention Is All You Need

**Yifan Zhang**[*◇1,4]  **Yifeng Liu**[*3]  **Huizhuo Yuan**[3]  **Zhen Qin**[3]
**Yang Yuan**[1,2]  **Quanquan Gu**[3]  **Andrew Chi-Chih Yao**[1,2†]

[1]IIIS, Tsinghua University  [2]Shanghai Qi Zhi Institute
[3]University of California, Los Angeles  [4]Princeton University
yifzhang@princeton.edu, liuyifeng@cs.ucla.edu
qgu@cs.ucla.edu, andrewcyao@tsinghua.edu.cn

## Abstract

Scaling language models to handle longer input sequences typically necessitates large key-value (KV) caches, resulting in substantial memory overhead during inference. In this paper, we propose **T**ensor **P**roduct **A**ttention (TPA), a novel attention mechanism that uses tensor decompositions to represent queries, keys, and values compactly, substantially shrinking the KV cache size at inference time. By factorizing these representations into contextual low-rank components and seamlessly integrating with RoPE and any possible position encoding mechanisms, TPA achieves improved model quality alongside memory efficiency. Based on TPA, we introduce the **T**ensor Produc**T** A**TT**en**T**ion **T**ransformer (T6), a new model architecture for sequence modeling. Through extensive empirical evaluation on language modeling tasks, we demonstrate that T6 surpasses or matches the performance of standard Transformer baselines, including Multi-Head Attention (MHA), Multi-Query Attention (MQA), Grouped-Query Attention (GQA), and Multi-Head Latent Attention (MLA) across various metrics, including perplexity and a range of established evaluation benchmarks. Notably, TPA's memory efficiency and computational efficiency at the decoding stage enable processing longer sequences under fixed resource constraints, addressing a critical scalability challenge in modern language models. Project Page: https://github.com/tensorgi/TPA.

## 1  Introduction

Large language models (LLMs) have revolutionized natural language processing, demonstrating exceptional performance across tasks [5, 12, 58, 6]. As these models evolve, their ability to process longer contexts becomes increasingly important for sophisticated applications such as document analysis, complex reasoning, and code completion. However, managing longer sequences during inference poses significant computational and memory challenges, particularly due to the storage of key-value (KV) caches [70, 34]. Because memory consumption grows linearly with sequence length, the maximum context window is limited by practical hardware constraints.

A variety of solutions have been explored to address this memory bottleneck. Some approaches compress or selectively prune cached states through sparse attention patterns [10] or token eviction strategies [70, 62, 42], though such methods risk discarding tokens that may later prove important. Other work proposes off-chip storage of key-value states [17], at the expense of increased I/O latency. Attention variants like Multi-Query Attention (MQA) [46] and Grouped-Query Attention (GQA) [2] reduce per-token cache requirements by sharing keys and values across heads, but often compromise flexibility or require significant architectural modifications. Meanwhile, low-rank weight factorization methods such as LoRA [20] effectively reduce fine-tuning memory, yet do not address the KV cache overhead that dominates inference at runtime. The recently introduced Multi-Head Latent Attention

---

[*]Equal contribution; [◇]Project lead; [†]Corresponding author.

39th Conference on Neural Information Processing Systems (NeurIPS 2025).

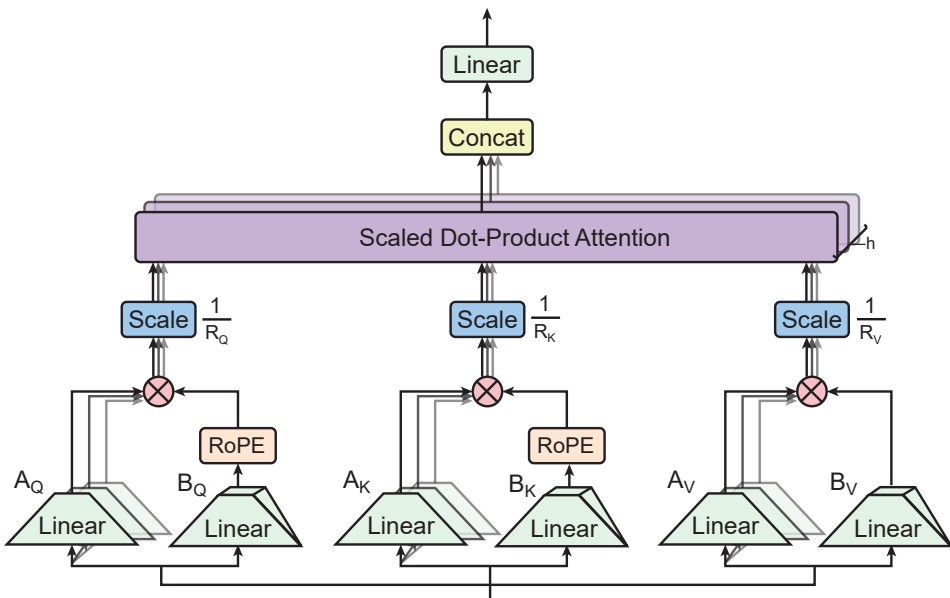

Figure 1: Tensor Product Attention (TPA) within the **T**ensor Produc**T** A**TT**en**T**ion **T**ransformer (T6). In each TPA layer, the input hidden state $\mathbf{x}_t$ is processed by linear layers to produce latent factor matrices for query (e.g., $\mathbf{A}_Q(\mathbf{x}_t), \mathbf{B}_Q(\mathbf{x}_t)$), key (e.g., $\mathbf{A}_K(\mathbf{x}_t), \mathbf{B}_K(\mathbf{x}_t)$), and value (e.g., $\mathbf{A}_V(\mathbf{x}_t), \mathbf{B}_V(\mathbf{x}_t)$). Rotary Position Embedding (RoPE) is applied to the $\mathbf{B}_Q(\mathbf{x}_t)$ and $\mathbf{B}_K(\mathbf{x}_t)$ factors. The query, key, and value tensors for each attention head are then formed by the tensor product of these factor matrices (e.g., $\mathbf{Q}_t = \frac{1}{R_Q}\mathbf{A}_Q(\mathbf{x}_t)^\top \mathbf{B}_Q(\mathbf{x}_t)$). Finally, the TPA output is computed using scaled dot-product attention, followed by a linear projection of the concatenated results from all heads.

(MLA) in Deepseek-V2 [32] caches compressed key-value representations but encounters difficulties with efficient Rotary Position Embedding (RoPE) [52] integration, necessitating additional position-encoded parameters per head.

To overcome the limitations of existing approaches, we introduce Tensor Product Attention (TPA), illustrated in Figure 1. TPA is a novel attention mechanism that employs tensor factorizations for queries (Q), keys (K), and values (V). By dynamically factorizing *activations* rather than static weights (as in LoRA), TPA constructs low-rank, contextual representations. This approach substantially reduces KV cache memory usage while offering improved representational capacity. In practice, TPA can decrease memory overhead by an order of magnitude compared to standard Multi-Head Attention (MHA), alongside achieving lower pretraining validation loss (perplexity) and better downstream performance. A key advantage of TPA is its native compatibility with rotary positional embeddings (RoPE) [52] and any possible position encodings, enabling a straightforward drop-in replacement for multi-head attention (MHA) layers in modern LLM architectures such as LLaMA [58], Qwen [3], and Gemma [56].

Our main contributions are summarized as follows:

1. We propose **Tensor Product Attention (TPA)**, a mechanism that factorizes $\mathbf{Q}$, $\mathbf{K}$, and $\mathbf{V}$ activations using *contextual* tensor decompositions. This achieves a substantial reduction in inference-time KV cache size relative to standard attention mechanisms [60], MHA, MQA, GQA, and MLA, while also improving performance. In addition, we analyze existing attention mechanisms and reveal that MHA, MQA, and GQA can be expressed as non-contextual variants of TPA.

2. We introduce the **T**ensor Produc**T** A**TT**en**T**ion **T**ransformer (T6), a new TPA-based model architecture for sequence modeling. In language modeling experiments, T6 consistently improves or matches validation perplexity and downstream evaluation performance, all while maintaining a reduced KV cache size.

3. We demonstrate that TPA integrates seamlessly with RoPE [52] and any possible position encodings as well as output gate and KV shifting, facilitating its easy adoption in popular foundation model architectures like LLaMA, Gemma, and Qwen.

4. We develop **FlashTPA Decoding**, an efficient autoregressive inference algorithm for TPA. Our empirical results show that FlashTPA Decoding can be faster than optimized MHA, MQA, GQA, and MLA decoding methods, particularly for long sequences.

## 2   Background

In this section, we briefly review Scaled Dot-Product Attention, Multi-Head Attention [60], and introduce key notations. Other attention mechanisms like Multi-Query Attention (MQA) [46], Grouped Query Attention (GQA) [2], Multi-head Latent Attention (MLA) [32, 33], and Rotary Position Embedding (RoPE) [52] are further discussed in the Appendix F.

**Notations.** We use bold uppercase letters (e.g., $\mathbf{X}$, $\mathbf{Q}$) for matrices, bold lowercase (e.g., $\mathbf{a}$, $\mathbf{b}$) for vectors, and italic uppercase (e.g., $\boldsymbol{W}_i^Q$) for learnable parameter matrices. We denote by $[n]$ the set $\{1, \ldots, n\}$ for some positive integer $n$. We use $\top$ to denote the transpose of a vector or a matrix. Let $d_{\text{model}}$ be the embedding dimension, $h$ the number of attention heads, $d_h$ the dimension per head, $\mathbf{x}_t \in \mathbb{R}^{d_{\text{model}}}$ the input for the $t$-th token at a given attention layer, $\mathbf{X} \in \mathbb{R}^{T \times d_{\text{model}}}$ denotes the input embeddings for $T$ tokens, and $\mathbf{Q}, \mathbf{K}, \mathbf{V} \in \mathbb{R}^{T \times h \times d_h}$ denote the queries, keys, and values of $h$ heads for $T$ tokens. With a little abuse of notation, $\mathbf{Q}_i, \mathbf{K}_i, \mathbf{V}_i \in \mathbb{R}^{T \times d_h}$ denote the $i$-th head of queries, keys, and values, and $\mathbf{Q}_t, \mathbf{K}_t, \mathbf{V}_t \in \mathbb{R}^{h \times d_h}$ denote the heads of the query, key, and value for $t$-th token. Throughout the paper, $\boldsymbol{W}^Q, \boldsymbol{W}^K, \boldsymbol{W}^V$ denote projection matrices for queries, keys, and values, respectively. In multi-head attention, each head is associated with its own set of $\boldsymbol{W}_i^Q, \boldsymbol{W}_i^K, \boldsymbol{W}_i^V$, and each has dimension $\boldsymbol{W}_i^Q, \boldsymbol{W}_i^K, \boldsymbol{W}_i^V \in \mathbb{R}^{d_{\text{model}} \times d_h}$.[5] Similarly, we have an output projection matrix $\boldsymbol{W}^O \in \mathbb{R}^{(h \cdot d_h) \times d_{\text{model}}}$.

We define the tensor product of two vectors as follows: for vectors $\mathbf{a} \in \mathbb{R}^m, \mathbf{b} \in \mathbb{R}^n$, the tensor product of $\mathbf{a}$ and $\mathbf{b}$ is: $\mathbf{a} \otimes \mathbf{b} = \mathbf{C} \in \mathbb{R}^{m \times n}$, with $C_{ij} = a_i b_j$, where $a_i$ is the $i$-th element of $\mathbf{a}$, $b_j$ is the $j$-th element of $\mathbf{b}$, and $C_{ij}$ is the $(i, j)$-th entry of $\mathbf{C}$. The vectorization of a matrix $\mathbf{C} \in \mathbb{R}^{m \times n}$, denoted $\text{vec}(\mathbf{C}) \in \mathbb{R}^{mn}$, stacks the columns of $\mathbf{C}$ into a single column vector. For example, if $\mathbf{C} = [\mathbf{c}_1, \mathbf{c}_2, \ldots, \mathbf{c}_n]$ where $\mathbf{c}_j$ are columns, then $\text{vec}(\mathbf{C}) = [\mathbf{c}_1^\top, \mathbf{c}_2^\top, \ldots, \mathbf{c}_n^\top]^\top$.

### 2.1   Scaled Dot-Product Attention

Scaled dot-product attention [60] determines how to focus on different parts of an input sequence by comparing queries ($\mathbf{Q}$) and keys ($\mathbf{K}$). It produces a weighted combination of the values ($\mathbf{V}$). Formally, the attention output is:

$$\text{Attention}(\mathbf{Q}, \mathbf{K}, \mathbf{V}) = \text{Softmax}\left(\frac{\mathbf{Q}\mathbf{K}^\top}{\sqrt{d_h}}\right) \mathbf{V},$$

where $\mathbf{Q} \in \mathbb{R}^{n \times d_h}$, $\mathbf{K} \in \mathbb{R}^{n \times d_h}$, and $\mathbf{V} \in \mathbb{R}^{n \times d_v}$ for $n$ tokens. The softmax is applied row-wise over the $n$ keys for each query.

### 2.2   Multi-Head Attention (MHA)

Multi-Head Attention (MHA) [60] extends scaled dot-product attention by dividing the model's internal representation into several *heads*. Each head learns different projections for queries, keys, and values, allowing the model to attend to different types of information from different representational subspaces. For each token embedding $\mathbf{x}_t \in \mathbb{R}^{d_{\text{model}}}$, MHA computes each head $i$ as follows:

$$\mathbf{Q}_{t,i} = (\boldsymbol{W}_i^Q)^\top \mathbf{x}_t \in \mathbb{R}^{d_h}, \ \mathbf{K}_{t,i} = (\boldsymbol{W}_i^K)^\top \mathbf{x}_t \in \mathbb{R}^{d_h}, \ \mathbf{V}_{t,i} = (\boldsymbol{W}_i^V)^\top \mathbf{x}_t \in \mathbb{R}^{d_h},$$

$$\mathbf{head}_i = \text{Attention}\Big(\mathbf{Q}_i, \mathbf{K}_i, \mathbf{V}_i\Big),$$

where $\boldsymbol{W}_i^Q, \boldsymbol{W}_i^K, \boldsymbol{W}_i^V \in \mathbb{R}^{d_{\text{model}} \times d_h}$ are learnable projection matrices for the $i$-th head, and $\mathbf{Q}_i, \mathbf{K}_i, \mathbf{V}_i \in \mathbb{R}^{T \times d_h}$ are the query, key, and value matrices for the $i$-th head over $T$ tokens. After computing each head's attention output, the results are concatenated and mapped back to the model's original dimension via another learnable linear projection matrix $\boldsymbol{W}^O \in \mathbb{R}^{h d_h \times d_{\text{model}}}$:

$$\text{MHA}(\mathbf{X}) = \text{Concat}\big(\mathbf{head}_1, \ldots, \mathbf{head}_h\big) \boldsymbol{W}^O.$$

MHA enables the model to capture a rich set of dependencies by allowing each head to focus on different aspects of the input sequence. We also discuss how MHA, MQA, and GQA relate to TPA in the Section 4.

---

[5]Often, $h \times d_h = d_{\text{model}}$, so each head has query/key/value dimension $d_h$.

# 3 Tensor Product Attention

In this section, we provide a detailed description of our proposed **Tensor Product Attention** (TPA), which enables *contextual* low-rank factorization for queries, keys, and values. First, we explain how TPA factorizes these components, specifying tensor shapes. Next, we describe TPA's integration into the multi-head attention framework and its benefits for reducing KV cache memory consumption during inference. Finally, we demonstrate RoPE's seamless integration with TPA, including a pre-rotated variant for efficiency.

## 3.1 Tensor Factorization of Queries, Keys, and Values

Let $d_{\text{attn}} := h \, d_h$ denote the total attention projection dimension. Typically one sets $d_{\text{attn}} = d_{\text{model}}$, but this is not required: when $d_{\text{attn}} \neq d_{\text{model}}$, the projection matrices $\boldsymbol{W}^Q, \boldsymbol{W}^K, \boldsymbol{W}^V$ map from $\mathbb{R}^{d_{\text{model}}}$ into $\mathbb{R}^{d_{\text{attn}}}$ and $\boldsymbol{W}^O$ maps $\mathbb{R}^{d_{\text{attn}}}$ back to $\mathbb{R}^{d_{\text{model}}}$. Standard attention projects the entire sequence into three tensors, $\mathbf{Q}, \mathbf{K}, \mathbf{V} \in \mathbb{R}^{T \times h \times d_h}$, where $\mathbf{Q}_t, \mathbf{K}_t, \mathbf{V}_t \in \mathbb{R}^{h \times d_h}$ denote the slices for the $t$-th token.

**Contextual Factorization.** Instead of forming each head's query, key, or value via a single linear map, TPA factorizes each $\mathbf{Q}_t, \mathbf{K}_t, \mathbf{V}_t$ into a sum of (contextual) tensor products whose ranks are $R_Q$, $R_K$, and $R_V$, respectively, and may differ. Specifically, for each token $t$, with a small abuse of notation, we define:

$$\mathbf{Q}_t = \frac{1}{R_Q} \sum_{r=1}^{R_Q} \mathbf{a}_r^Q(\mathbf{x}_t) \otimes \mathbf{b}_r^Q(\mathbf{x}_t), \qquad \mathbf{K}_t = \frac{1}{R_K} \sum_{r=1}^{R_K} \mathbf{a}_r^K(\mathbf{x}_t) \otimes \mathbf{b}_r^K(\mathbf{x}_t),$$

$$\mathbf{V}_t = \frac{1}{R_V} \sum_{r=1}^{R_V} \mathbf{a}_r^V(\mathbf{x}_t) \otimes \mathbf{b}_r^V(\mathbf{x}_t), \tag{3.1}$$

where $\mathbf{a}_r^Q(\mathbf{x}_t), \mathbf{a}_r^K(\mathbf{x}_t), \mathbf{a}_r^V(\mathbf{x}_t) \in \mathbb{R}^h, \mathbf{b}_r^Q(\mathbf{x}_t), \mathbf{b}_r^K(\mathbf{x}_t), \mathbf{b}_r^V(\mathbf{x}_t) \in \mathbb{R}^{d_h}$. Hence, for queries, each tensor product $\mathbf{a}_r^Q(\mathbf{x}_t) \otimes \mathbf{b}_r^Q(\mathbf{x}_t): \mathbb{R}^h \times \mathbb{R}^{d_h} \to \mathbb{R}^{h \times d_h}$ contributes to the query slice $\mathbf{Q}_t \in \mathbb{R}^{h \times d_h}$. Analogous definitions apply to the key slice $\mathbf{K}_t$ and value slice $\mathbf{V}_t$.

**Latent Factor Maps.** Each factor in the tensor product depends on the token's hidden state $\mathbf{x}_t$. For example, for queries, we can write:

$$\mathbf{a}_r^Q(\mathbf{x}_t) = \boldsymbol{W}_r^{a^Q} \mathbf{x}_t \in \mathbb{R}^h, \quad \mathbf{b}_r^Q(\mathbf{x}_t) = \boldsymbol{W}_r^{b^Q} \mathbf{x}_t \in \mathbb{R}^{d_h},$$

where $\boldsymbol{W}_r^{a^Q} \in \mathbb{R}^{h \times d_{\text{model}}}$ and $\boldsymbol{W}_r^{b^Q} \in \mathbb{R}^{d_h \times d_{\text{model}}}$ are learnable weight matrices. Similar linear maps produce the factors for keys and values.

One often merges the rank index into a single output dimension. For instance, for queries:

$$\mathbf{a}^Q(\mathbf{x}_t) = \boldsymbol{W}^{a^Q} \mathbf{x}_t \in \mathbb{R}^{R_Q \cdot h}, \ \mathbf{b}^Q(\mathbf{x}_t) = \boldsymbol{W}^{b^Q} \mathbf{x}_t \in \mathbb{R}^{R_Q \cdot d_h},$$

which are then reshaped into $\mathbf{A}_Q(\mathbf{x}_t) \in \mathbb{R}^{R_Q \times h}$ and $\mathbf{B}_Q(\mathbf{x}_t) \in \mathbb{R}^{R_Q \times d_h}$ (where each row of $\mathbf{A}_Q(\mathbf{x}_t)$ corresponds to an $\mathbf{a}_r^Q(\mathbf{x}_t)^\top$ and each row of $\mathbf{B}_Q(\mathbf{x}_t)$ to a $\mathbf{b}_r^Q(\mathbf{x}_t)^\top$). The query tensor for token $t$ can then be expressed as:

$$\mathbf{Q}_t = \frac{1}{R_Q} \mathbf{A}_Q(\mathbf{x}_t)^\top \mathbf{B}_Q(\mathbf{x}_t) \in \mathbb{R}^{h \times d_h}.$$

This operation is equivalent to $\mathbf{Q}_t = \frac{1}{R_Q} \sum_{r=1}^{R_Q} \mathbf{a}_r^Q(\mathbf{x}_t)(\mathbf{b}_r^Q(\mathbf{x}_t))^\top$, where $\mathbf{a}_r^Q$ is the $r$-th column of $\mathbf{A}_Q(\mathbf{x}_t)^\top$ and $(\mathbf{b}_r^Q)^\top$ is the $r$-th row of $\mathbf{B}_Q(\mathbf{x}_t)$. Repeating for all tokens reconstitutes $\mathbf{Q} \in \mathbb{R}^{T \times h \times d_h}$. Similar procedures are applied to obtain $\mathbf{K}$ and $\mathbf{V}$ with ranks $R_K$ and $R_V$, respectively.

**Scaled Dot-Product Attention.** Once $\mathbf{Q}, \mathbf{K}, \mathbf{V}$ are factorized, multi-head attention proceeds as in standard Transformers. For each head $i \in \{1, \ldots, h\}$:

$$\mathbf{head}_i = \text{Softmax}\left(\frac{1}{\sqrt{d_h}} \mathbf{Q}_i (\mathbf{K}_i)^\top\right) \mathbf{V}_i, \tag{3.2}$$

where $\mathbf{Q}_i, \mathbf{K}_i, \mathbf{V}_i \in \mathbb{R}^{T \times d_h}$ are the slices along the head dimension. Concatenating these $h$ heads along the last dimension yields an $\mathbb{R}^{T \times (h \cdot d_h)}$ tensor, which is projected back to $\mathbb{R}^{T \times d_{\text{model}}}$ by an output weight matrix $\boldsymbol{W}^O \in \mathbb{R}^{(h \cdot d_h) \times d_{\text{model}}}$:

$$\text{TPA}(\mathbf{Q}, \mathbf{K}, \mathbf{V}) = \text{Concat}(\mathbf{head}_1, \ldots, \mathbf{head}_h) \boldsymbol{W}^O. \tag{3.3}$$

**Parameter Initialization.** We use Xavier initialization [15] for the factor weight matrices; details are in the Appendix G.

## 3.2 RoPE Compatibility and Acceleration

In a typical workflow of adding RoPE to standard multi-head attention, one first computes $\mathbf{Q}_t, \mathbf{K}_s \in \mathbb{R}^{h \times d_h}$ of the $t$-th token and $s$-th token and then applies:

$$\mathbf{Q}_t \mapsto \widetilde{\mathbf{Q}}_t = \mathrm{RoPE}_t(\mathbf{Q}_t), \qquad \mathbf{K}_s \mapsto \widetilde{\mathbf{K}}_s = \mathrm{RoPE}_s(\mathbf{K}_s). \tag{3.4}$$

**Direct Integration.** A useful optimization is to integrate RoPE directly into the TPA factorization. For example, one can *pre-rotate* the token-dimension factors:

$$\widetilde{\mathbf{B}}_K(\mathbf{x}_t) := \mathrm{RoPE}_t\big(\mathbf{B}_K(\mathbf{x}_t)\big) = \mathbf{B}_K(\mathbf{x}_t)\mathbf{T}_t, \tag{3.5}$$

yielding a *pre-rotated* key representation:

$$\widetilde{\mathbf{K}}_t = \frac{1}{R_K} \sum_{r=1}^{R_K} \mathbf{a}_r^K(\mathbf{x}_t) \otimes \mathrm{RoPE}_t\big(\mathbf{b}_r^K(\mathbf{x}_t)\big) = \frac{1}{R_K} \mathbf{A}_K(\mathbf{x}_t)^\top \widetilde{\mathbf{B}}_K(\mathbf{x}_t).$$

Here, $\mathrm{RoPE}_t$ is applied to each row of $\mathbf{B}_K(\mathbf{x}_t)$ (i.e., to each $\mathbf{b}_r^K(\mathbf{x}_t)$ vector). Thus, each cached key factor corresponds to a RoPE-rotated key slice. This removes the need to rotate *cached* keys at decoding time; the current-step query (which is not cached) can still be rotated on the fly at negligible cost. Depending on hardware and performance requirements, different RoPE integration strategies can be adopted for training and inference.

**Theorem 3.1** (RoPE's Compatibility with TPA). Let $\mathbf{Q}_t$ be factorized by TPA as

$$\mathbf{Q}_t = \frac{1}{R_Q} \mathbf{A}_Q(\mathbf{x}_t)^\top \mathbf{B}_Q(\mathbf{x}_t) \in \mathbb{R}^{h \times d_h},$$

where $\mathbf{A}_Q(\mathbf{x}_t) \in \mathbb{R}^{R_Q \times h}$ and $\mathbf{B}_Q(\mathbf{x}_t) \in \mathbb{R}^{R_Q \times d_h}$. Then we have:

$$\mathrm{RoPE}_t(\mathbf{Q}_t) = \mathbf{Q}_t \mathbf{T}_t = \frac{1}{R_Q} \mathbf{A}_Q(\mathbf{x}_t)^\top \widetilde{\mathbf{B}}_Q(\mathbf{x}_t), \tag{3.6}$$

where $\widetilde{\mathbf{B}}_Q(\mathbf{x}_t) := \mathbf{B}_Q(\mathbf{x}_t)\mathbf{T}_t = \mathrm{RoPE}_t\big(\mathbf{B}_Q(\mathbf{x}_t)\big)$ (RoPE applied row-wise to $\mathbf{B}_Q(\mathbf{x}_t)$). Furthermore, let $\widetilde{\mathbf{Q}}_t = \mathrm{RoPE}_t(\mathbf{Q}_t) = \mathbf{Q}_t \mathbf{T}_t$ and $\widetilde{\mathbf{K}}_s = \mathrm{RoPE}_s(\mathbf{K}_s) = \mathbf{K}_s \mathbf{T}_s$ be the RoPE-transformed query/key slices. Then RoPE's standard relative-position identity is preserved:

$$\widetilde{\mathbf{Q}}_t \widetilde{\mathbf{K}}_s^\top = \mathbf{Q}_t \mathbf{T}_{t-s} \mathbf{K}_s^\top, \qquad \text{equivalently} \qquad \mathrm{RoPE}_{t-s}(\mathbf{Q}_t) \mathbf{K}_s^\top = \widetilde{\mathbf{Q}}_t \widetilde{\mathbf{K}}_s^\top,$$

where $\mathbf{T}_{t-s} := \mathbf{T}_t \mathbf{T}_s^\top$. In particular, for any head $i$ (the $i$-th row), if $\mathbf{q}_{t,i}, \mathbf{k}_{s,i} \in \mathbb{R}^{1 \times d_h}$ and $\widetilde{\mathbf{q}}_{t,i} = \mathbf{q}_{t,i} \mathbf{T}_t$, $\widetilde{\mathbf{k}}_{s,i} = \mathbf{k}_{s,i} \mathbf{T}_s$, then $\widetilde{\mathbf{q}}_{t,i} \widetilde{\mathbf{k}}_{s,i}^\top = \mathbf{q}_{t,i} \mathbf{T}_{t-s} \mathbf{k}_{s,i}^\top$.

Theorem 3.1 indicates that TPA does not break RoPE's relative translational property. We prove it in the Appendix D.1.

## 3.3 KV Caching and Memory Reduction

In autoregressive decoding, standard attention caches $\mathbf{K}_t, \mathbf{V}_t \in \mathbb{R}^{h \times d_h}$ for each past token $t$. This accumulates to $\mathbb{R}^{T \times h \times d_h}$ for keys and $\mathbb{R}^{T \times h \times d_h}$ for values, i.e., $2 T h d_h$ total.

**TPA Factorized KV Caching.** Instead of storing the full $\mathbf{K}_t$ and $\mathbf{V}_t$, TPA stores only their factor components. Specifically, for each past token $t$, we cache:

$$\mathbf{A}_K(\mathbf{x}_t), \widetilde{\mathbf{B}}_K(\mathbf{x}_t) \quad \text{and} \quad \mathbf{A}_V(\mathbf{x}_t), \mathbf{B}_V(\mathbf{x}_t),$$

where $\mathbf{A}_K(\mathbf{x}_t) \in \mathbb{R}^{R_K \times h}$, $\widetilde{\mathbf{B}}_K(\mathbf{x}_t) \in \mathbb{R}^{R_K \times d_h}$ (pre-rotated), $\mathbf{A}_V(\mathbf{x}_t) \in \mathbb{R}^{R_V \times h}$, $\mathbf{B}_V(\mathbf{x}_t) \in \mathbb{R}^{R_V \times d_h}$.

Hence, the memory cost per token is $\underbrace{R_K(h + d_h)}_{\text{for K}} + \underbrace{R_V(h + d_h)}_{\text{for V}} = (R_K + R_V)(h + d_h)$.

Compared to the standard caching cost of $2 h d_h$, the ratio is $\frac{(R_K + R_V)(h + d_h)}{2 h d_h}$. For large $h$ and $d_h$ (typically $d_h = 64$ or $128$), setting $R_K, R_V \ll h$ (e.g., rank 1 or 2) often yields substantial reduction of KV cache size. Table 1 provides a comparative overview of different attention mechanisms, including TPA and its variants, focusing on KV cache size per token and the number of parameters in an attention layer.

Table 1: Comparison of different attention mechanisms. Here, $R_Q$, $R_K$, and $R_V$ denote the ranks for queries, keys, and values in TPA, respectively. Variants of TPA, such as TPA (KVonly), TPA (Non-contextual A), and TPA (Non-contextual B), are detailed in the Appendix G. For MLA, $d_h^R$ and $d_h$ are the dimensions for RoPE and non-RoPE parts; $d_c'$ and $d_c$ are the dimensions of compressed vectors for query and key-value, respectively. The MLA parameter count includes the output projection.

| METHOD | KV CACHE | # PARAMETERS | # QUERY HEADS | # KV HEADS |
|---|---|---|---|---|
| MHA | $2hd_h$ | $4d_{\text{model}}\,h\,d_h$ | $h$ | $h$ |
| MQA | $2d_h$ | $2d_{\text{model}}\,d_h\,(h+1)$ | $h$ | $1$ |
| GQA | $2Gd_h$ | $2d_{\text{model}}\,d_h\,(h+G)$ | $h$ | $G$ |
| MLA | $d_c + d_h^R$ | $d_c'(d_{\text{model}} + hd_h + hd_h^R)$ $+d_c(d_{\text{model}} + 2hd_h)$ $+d_{\text{model}}(hd_h + d_h^R)$ | $h$ | $h$ |
| TPA | $(R_K + R_V)(h + d_h)$ | $d_{\text{model}}(R_Q + R_K + R_V)(h + d_h) + d_{\text{model}}\,hd_h$ | $h$ | $h$ |
| TPA (KVonly) | $(R_K + R_V)(h + d_h)$ | $d_{\text{model}}(R_K + R_V)(h + d_h) + 2d_{\text{model}}\,hd_h$ | $h$ | $h$ |
| TPA (Non-contextual A) | $(R_K + R_V)d_h$ | $(R_Q + R_K + R_V)(d_{\text{model}}d_h + h) + d_{\text{model}}\,hd_h$ | $h$ | $h$ |
| TPA (Non-contextual B) | $(R_K + R_V)h$ | $(R_Q + R_K + R_V)(d_{\text{model}}h + d_h) + d_{\text{model}}\,hd_h$ | $h$ | $h$ |

## 4 Expressing MHA, MQA, GQA as Non-contextual TPA

We demonstrate that standard Multi-Head Attention (MHA), Multi-Query Attention (MQA), and Grouped-Query Attention (GQA) can be expressed as special, non-contextual variants of Tensor Product Attention (TPA). This is achieved by imposing specific constraints on the TPA factors, particularly by making the head-dimension factors ($\mathbf{a}$) independent of the input token ($\mathbf{x}_t$).

### 4.1 MHA as Non-contextual TPA

Standard Multi-Head Attention (MHA) can be precisely formulated as a TPA where the rank is equal to the number of heads ($R_Q = R_K = R_V = h$), and the head-dimension factors are fixed, non-contextual basis vectors. To recover MHA, we set the rank $R_Q = h$ and define the factors for each head $i \in [h]$ as follows:

- **Contextual token factor**: This is the standard linear projection for the $i$-th head's query:

$$\mathbf{b}_i^Q(\mathbf{x}_t) = (\boldsymbol{W}_i^Q)^\top \mathbf{x}_t \in \mathbb{R}^{d_h}$$

- **Non-contextual head factor**: This factor is a scaled standard basis vector, independent of $\mathbf{x}_t$:

$$\mathbf{a}_i^Q = h \cdot \mathbf{e}_i \in \mathbb{R}^h$$

where $\mathbf{e}_i$ is the $i$-th standard basis vector (a vector of zeros with a one at the $i$-th position).

Substituting these into the TPA equation, the $1/R_Q = 1/h$ scaling factor cancels with the scaling of the $\mathbf{a}_i^Q$ factor:

$$\mathbf{Q}_t = \frac{1}{h}\sum_{i=1}^h (h \cdot \mathbf{e}_i) \otimes \left((\boldsymbol{W}_i^Q)^\top \mathbf{x}_t\right) = \sum_{i=1}^h \mathbf{e}_i \otimes \left((\boldsymbol{W}_i^Q)^\top \mathbf{x}_t\right)$$

The resulting tensor product, $\mathbf{e}_i \otimes \mathbf{b}_i^Q(\mathbf{x}_t)$, produces an $h \times d_h$ matrix where only the $i$-th row is non-zero and contains the vector $(\mathbf{b}_i^Q(\mathbf{x}_t))^\top$. Summing these matrices for $i = 1, \ldots, h$ assembles the complete query tensor $\mathbf{Q}_t$, where the $i$-th row is precisely the query vector for the $i$-th head in standard MHA. An analogous construction applies to the key ($\mathbf{K}_t$) and value ($\mathbf{V}_t$) tensors.

Thus, MHA is equivalent to a non-contextual TPA where the head-dimension factors are fixed and orthogonal, effectively assigning a dedicated rank component to each attention head.

### 4.2 MQA and GQA as Non-contextual TPA

Similarly, Multi-Query Attention (MQA) and Grouped-Query Attention (GQA) can be seen as non-contextual TPAs where the key and value tensors are formed with a rank lower than the number of heads.

- **MQA as Rank-1 TPA (for K and V).** In MQA, all $h$ query heads share a single key and value. This corresponds to a TPA with ranks $R_K = 1$ and $R_V = 1$. The key tensor $\mathbf{K}_t$ is formed using a

single, non-contextual head-dimension factor $\mathbf{a}^K = \mathbf{1}_h$ (a vector of all ones) and a single contextual token-dimension factor $\mathbf{b}^K(\mathbf{x}_t) = (\boldsymbol{W}^K)^\top \mathbf{x}_t$:

$$\mathbf{K}_t = \frac{1}{1}\left(\mathbf{1}_h \otimes \mathbf{b}^K(\mathbf{x}_t)\right)$$

This creates an $h \times d_h$ matrix where every row is the same shared key vector $(\mathbf{b}^K(\mathbf{x}_t))^\top$. The same logic applies to the value tensor $\mathbf{V}_t$. The queries remain full-rank ($R_Q = h$) as in MHA.

- **GQA as Rank-G TPA (for K and V).** GQA is an intermediate approach where $h$ heads are divided into $G$ groups, with heads in the same group sharing a key and value. This is equivalent to a TPA with ranks $R_K = G$ and $R_V = G$. The key tensor is formed by summing $G$ components:

$$\mathbf{K}_t = \frac{1}{G}\sum_{j=1}^{G}\mathbf{a}_j^K \otimes \mathbf{b}_j^K(\mathbf{x}_t)$$

Here, $\mathbf{b}_j^K(\mathbf{x}_t)$ is the shared key vector for group $j$. The non-contextual factor $\mathbf{a}_j^K$ is a scaled mask vector, defined as $\mathbf{a}_j^K = G \cdot \text{mask}_j$, where the $\text{mask}_j$ vector has ones for heads belonging to group $j$ and zeros elsewhere. This scaling cancels the $1/G$ pre-factor:

$$\mathbf{K}_t = \frac{1}{G}\sum_{j=1}^{G}(G \cdot \text{mask}_j) \otimes \mathbf{b}_j^K(\mathbf{x}_t) = \sum_{j=1}^{G}\text{mask}_j \otimes \mathbf{b}_j^K(\mathbf{x}_t)$$

For example, with $h = 8$ heads and $G = 2$ groups (2 KV heads), the factor for the first group of 4 heads would be $\mathbf{a}_1^K = 2 \cdot [1, 1, 1, 1, 0, 0, 0, 0]^\top$. This construction correctly assembles the final key tensor by broadcasting each group's shared key to its designated heads without any unintended extra scaling.

This perspective highlights that MHA, MQA, and GQA are specific instances of a more general TPA framework, where expressiveness and parameter sharing are controlled by the rank and the nature (contextual vs. non-contextual) of the tensor factors.

## 4.3 Model Architectures

We propose a new architecture called **T**ensor Produc**T** A**TT**en**T**ion **T**ransformer (T6), which uses our **Tensor Product Attention** (TPA) in place of standard MHA (multi-head attention) or GQA (grouped-query attention). Building upon the query, key, and value tensors $\mathbf{Q}, \mathbf{K}, \mathbf{V} \in \mathbb{R}^{T \times h \times d_h}$ defined in Section 3.1, T6 utilizes the overall architecture of LLaMA [58] while changing the self-attention block to our TPA-based version. The feed-forward network (FFN) adopts a SwiGLU layer, as in [47, 58].

**Rotary Positional Embedding (RoPE).** As discussed in Section 3.2, RoPE [52] is applied to the $\mathbf{Q}$ and $\mathbf{K}$. Within TPA, we *pre-rotate* the factor $\mathbf{b}_t^Q(\mathbf{x}_t)$ and $\mathbf{b}_s^K(\mathbf{x}_s)$ directly, so that each $\mathbf{K}_s$ is already rotated prior to caching, see Equation (3.5) and Theorem 3.1.

**SwiGLU Feed-Forward Network.** Following [47, 58], our T6 uses a SwiGLU-based Feed-Forward Network (FFN): $\text{FFN}(\mathbf{x}) = \left[\sigma(\mathbf{x}\,\boldsymbol{W}_1) \odot (\mathbf{x}\,\boldsymbol{W}_2)\right]\boldsymbol{W}_3$, where $\sigma$ is the SiLU (a.k.a., swish) nonlinearity, $\odot$ is element-wise product, and $\boldsymbol{W}_1, \boldsymbol{W}_2, \boldsymbol{W}_3$ are learnable parameters. Note that other activation functions can also be used.

**Overall T6 Block Structure.** Putting everything together, one T6 block consists of:

$$\mathbf{x} \leftarrow \mathbf{x} + \text{TPA}\big(\text{RMSNorm}(\mathbf{x})\big),$$
$$\mathbf{x} \leftarrow \mathbf{x} + \text{SwiGLU-FFN}\big(\text{RMSNorm}(\mathbf{x})\big).$$

We place norm layers (e.g., RMSNorm) before each sub-layer. Stacking $L$ such blocks yields a T6 model architecture with $L$ layers.

## 5 FlashTPA Decoding Algorithm

For efficient autoregressive inference with Tensor Product Attention (TPA), we introduce FlashTPA Decoding. This algorithm is optimized for generating one token at a time by leveraging the factorized

representation of queries, keys, and values. The core idea, illustrated in Figure 2, is to perform attention computations using a sequence of Einstein summations ("einsum") that operate directly on these factorized components. This avoids materializing the full query, key, and value tensors, which is particularly beneficial as the Key-Value (KV) cache grows with sequence length. The detailed definitions of the input factorized components and the step-by-step pseudo-code for FlashTPA Decoding are provided in Algorithm 2. An optimized Triton kernel implementation is outlined in Algorithm 3 (see Appendix B.1).

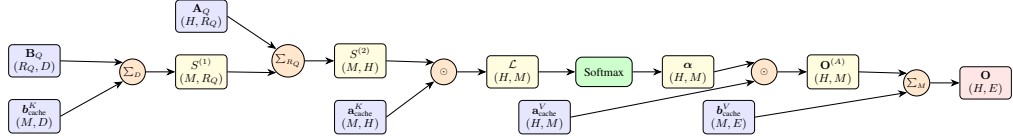

Figure 2: Data flow diagram for FlashTPA Decoding. Rectangles represent tensors (blue for inputs, yellow for intermediates, red for final output), circles with $\sum$ or $\odot$ denote Einstein summation contractions or element-wise products respectively, and the green rounded rectangle is the softmax operation. Shapes are shown for a single query ($N = 1$) interacting with $M$ cached items in the common rank-1 setting $R_K = R_V = 1$. We use a head-first layout $(H, M)$ for logits and attention weights; the cached head factors $\mathbf{a}_{\text{cache}}^K$ and $\mathbf{a}_{\text{cache}}^V$ are shown transposed relative to their natural token-major layout for readability. $H$ is the number of heads, $R_Q$ is the query rank, and $D, E$ are respective feature dimensions for the $\mathbf{B}_Q/\boldsymbol{b}_{\text{cache}}^K$ and $\boldsymbol{b}_{\text{cache}}^V$ factors. Scaling factors are omitted for visual clarity.

This sequence of factorized operations allows FlashTPA Decoding to compute the attention output efficiently. Consequently, TPA is not only memory-efficient due to its smaller KV cache footprint but can also be computationally efficient during inference. The experimental results for FlashTPA decoding time are presented in Section 6.2.

## 6 Experiments

### 6.1 Language Modeling Tasks

All experiments reported in this paper are implemented based on the `nanoGPT` codebase [24], and we pretrain our models using the FineWeb-Edu 100B dataset [37]. The dataset contains 100 billion tokens for training and 0.1 billion tokens for validation. We compare T6 against the baseline Llama architecture [58] with SwiGLU activation [47] and RoPE embeddings [52], as well as Llama variants that replace Multi-Head Attention (MHA; [60]) with Multi-Query Attention (MQA; [46]), Grouped Query Attention (GQA; [2]), or Multi-head Latent Attention (MLA; [32]). In our experiments, the number of heads $h$ is adjusted for each attention mechanism to ensure that all attention mechanisms have the same number of parameters as the standard Multi-Head Attention (MHA), which has $4d_{\text{model}}^2$ parameters per attention layer. We train models at four scales: *small* (124M parameters), *medium* (353M), *large* (773M), and *XL* (1.5B). We pretrain all models for 50B tokens (roughly half an epoch over FineWeb-Edu-100B). Details on architecture hyperparameters and training hardware are shown in Appendix H.1.

**Training & Validation Curves.** Figure 4 compares validation loss curves for the *medium* (353M), *large* (773M), and *XL* (1.5B) models on FineWeb-Edu-100B. Training loss curves are provided in Appendix Figure 3. Overall, **TPA** (red curves) and its simpler variant **TPA-KVonly** (pink curves) (see Appendix G) converge as fast as or faster than the baselines (MHA, MQA, GQA, MLA) while also achieving visibly lower final validation losses. For instance, in Figure 4(b), TPA and TPA-KVonly remain below the MHA baseline in terms of validation loss at nearly all training stages. Meanwhile, Multi-Head Latent Attention (MLA) [32] (blue curves) generally trains more slowly and yields higher validation losses.

**Validation Perplexity.** Figure 9 (in the Appendix) shows the validation perplexities of the *medium*- and *large*-scale models. Mirroring the loss curves, **TPA** and **TPA-KVonly** steadily outperform MHA, MQA, GQA, and MLA over the course of training. By the end of pretraining (around 49B tokens), TPA-based approaches achieve the lowest perplexities in most configurations.

**Downstream Evaluation.** We evaluate zero-shot and two-shot performance on standard benchmarks, including ARC [63], BoolQ [13], HellaSwag [64], OBQA [39], PIQA [4], WinoGrande [43],

and MMLU [18], using the `lm-evaluation-harness` codebase [14]. For ARC-E, ARC-C, HellaSwag, OBQA, PIQA, and SciQ, we report accuracy norm; for other tasks, we report standard accuracy. Due to the page limitation, we only display the zero-shot evaluation results of *medium* and *large* models here in Tables 2 and 3. Zero-shot evaluation of *small* and *XL* models are displayed in Tables 11 and 12 in the appendix. Moreover, we also present 2-shot evaluation results in Tables 13, 14, 15 and 16 in the appendix.

For the *medium*-size (353M) models (Table 2 for 0-shot and Table 14 in appendix for 2-shot), TPA generally ties or outperforms all competing methods, achieving, for example, an average of 51.41% in zero-shot mode versus MHA's 50.11%, MQA's 50.44%, and MLA's 50.13%. When given two-shot prompts, TPA again leads with 53.12% average accuracy. A similar trend appears for the *large*-size (773M) models (Table 3), where TPA-KVonly attains the highest average (53.52% zero-shot). For the *XL* size models (1.5B) (Table 12 in the appendix), TPA-KV only achieves the highest average (55.03% zero-shot). Our experiments confirm that TPA consistently matches or exceeds the performance of established attention mechanisms (MHA, MQA, GQA, MLA) across *medium* and *large* model scales.

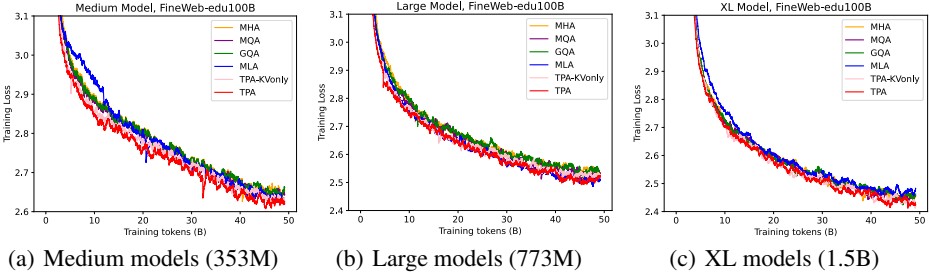

(a) Medium models (353M)    (b) Large models (773M)    (c) XL models (1.5B)

Figure 3: The training loss of medium-size (353M), large-size (773M) as well as XL-size (1.5B) models, with different attention mechanisms on the FineWeb-Edu 100B dataset.

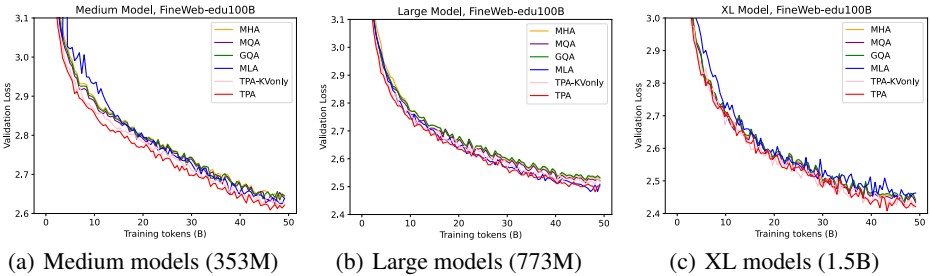

(a) Medium models (353M)    (b) Large models (773M)    (c) XL models (1.5B)

Figure 4: The validation loss of medium-size (353M), large-size (773M) as well as XL-size (1.5B) models, with different attention mechanisms on the FineWeb-Edu 100B dataset.

Table 2: The evaluation results of medium models with different attention mechanisms pre-trained using FineWeb-Edu 100B dataset (0-shot with lm-evaluation-harness). The best scores in each column are **bolded**. Abbreviations: HellaSw. = HellaSwag, W.G. = WinoGrande.

| Method | ARC-E | ARC-C | BoolQ | HellaSw. | OBQA | PIQA | W.G. | MMLU | SciQ | Avg. |
|---|---|---|---|---|---|---|---|---|---|---|
| MHA | **59.51** | 29.52 | **59.60** | 45.68 | 34.20 | 68.82 | 53.43 | 23.33 | 76.90 | 50.11 |
| MQA | 57.62 | **31.91** | 59.45 | 45.69 | 35.40 | 69.31 | 53.51 | **26.47** | 74.60 | 50.44 |
| GQA | 58.67 | 31.48 | 58.29 | 45.45 | 35.20 | 68.50 | **54.46** | 24.58 | 76.50 | 50.35 |
| MLA | 56.65 | 29.52 | 57.83 | 46.05 | 34.60 | 69.42 | 52.80 | 24.62 | 79.70 | 50.13 |
| **TPA-KVonly** | 58.01 | 30.12 | 58.01 | 45.95 | 35.60 | 69.10 | 53.12 | 25.39 | 75.10 | 50.04 |
| **TPA** | 58.38 | 31.57 | 59.39 | **46.83** | **37.00** | **70.02** | 54.06 | 25.52 | **79.90** | **51.41** |

## 6.2 Experimental Results on FlashTPA Decoding

This section presents an evaluation of FlashTPA's decoding time in comparison to several other optimized attention mechanisms. We benchmark FlashTPA against FlashMHA [45], FlashGQA, FlashMQA, and FlashMLA [23]. It is important to note that our current FlashTPA implementation utilizes Triton [57]. While the compared methods are typically available as highly optimized CUDA kernels, these experiments provide initial insights into FlashTPA's potential. Development of a CUDA-based FlashTPA kernel is ongoing and is expected to yield further performance improvements.

Table 3: The evaluation results of large models with different attention mechanisms pre-trained using the FineWeb-Edu 100B dataset (0-shot with lm-evaluation-harness). The best scores in each column are **bolded**. Abbreviations: HellaSw. = HellaSwag, W.G. = WinoGrande.

| Method | ARC-E | ARC-C | BoolQ | HellaSw. | OBQA | PIQA | W.G. | MMLU | SciQ | Avg. |
|--------|-------|-------|-------|----------|------|------|------|------|------|------|
| MHA | 59.93 | 33.62 | 61.93 | 50.63 | 36.00 | 71.06 | 55.41 | 22.87 | 81.20 | 52.52 |
| MQA | 60.73 | 33.62 | 57.34 | 50.09 | 37.00 | 69.97 | 55.49 | 25.30 | 79.60 | 52.13 |
| GQA | 61.66 | 34.30 | 58.72 | 49.85 | 38.40 | 71.16 | 53.75 | 25.23 | 77.60 | 52.30 |
| MLA | **63.55** | 32.85 | 60.95 | **51.72** | **38.80** | 70.51 | 55.01 | 24.55 | **81.90** | 53.32 |
| **TPA-KVonly** | 63.26 | 34.13 | **61.96** | 50.66 | 37.20 | **72.09** | 55.25 | **26.06** | 81.10 | **53.52** |
| **TPA** | 63.22 | **35.58** | 60.03 | 51.26 | 36.80 | 71.44 | **55.56** | 24.77 | 79.60 | 53.10 |

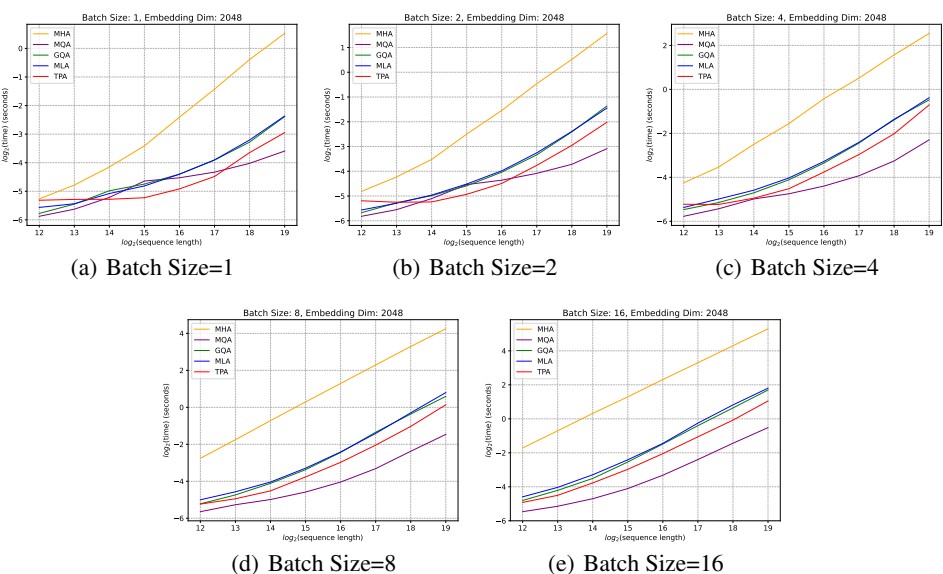

(a) Batch Size=1     (b) Batch Size=2     (c) Batch Size=4

(d) Batch Size=8     (e) Batch Size=16

Figure 5: Decoding time comparison of different attention mechanisms with an embedding dimension of 2048 and $d_h = 64$. The y-axis represents $\log_2(\text{time})$ in seconds, and the x-axis represents $\log_2(\text{sequence length})$. Each subfigure corresponds to a different batch size.

The evaluations were performed with batch sizes selected from $\{1, 2, 4, 8, 16\}$, model embedding dimensions ($d_{\text{model}}$) chosen from $\{1024, 2048, 3072\}$, and sequence lengths ranging from $2^{12}$ (4,096) to $2^{19}$ (524,288). For all experiments, the dimension per head ($d_h$) was fixed at 64. The ranks for TPA's factorized components ($R_Q, R_K, R_V$) were set to $(16, 1, 1)$, and for GQA configurations, the number of key-value head groups was 4. The decoding time per token, measured as $\log_2(\text{time})$ in seconds, is plotted against $\log_2(\text{sequence length})$. Lower values on the y-axis indicate faster decoding times. Results are presented in Figure 5 for an embedding dimension of 2048 (corresponding to 32 attention heads). Additional results for embedding dimensions of 1024 (16 heads, Figure 8) and 3072 (48 heads, Figure 7) are provided in Appendix B. Figure 5 depicts these speed comparisons for an embedding dimension of 2048. The results indicate that FlashTPA (blue line) is highly competitive and often outperforms other attention mechanisms, especially as the sequence length increases.

# 7 Conclusion

We introduced Tensor Product Attention (TPA), which factorizes query, key, and value matrices into rank-$R$ tensor products dependent on the token's hidden state. Storing only the factorized key/value components during autoregressive decoding substantially decreases the KV memory size with improved performance compared with MHA, MQA, GQA, and MLA. The approach is fully compatible with RoPE (and can store pre-rotated keys). Variants of TPA include factorizing only the key/value or sharing basis vectors across tokens. Overall, TPA offers a powerful mechanism for compressing KV storage while improving the model performance, thereby enabling longer sequence contexts under constrained memory.

## Acknowledgements

We thank the anonymous reviewers and area chairs for their helpful comments. We acknowledge the compute credits provided by Fetch.ai.

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

# Appendix

# A Toward Faster Computation Without Materializing Q, K and V

Our objective in this section is to compute attention *without* explicitly forming $\mathbf{Q}, \mathbf{K}, \mathbf{V}$, by contracting their *factorized* representations in a cache- and throughput-friendly order. Recall from Equation (3.1) that each per-token slice $\mathbf{Q}_t, \mathbf{K}_t, \mathbf{V}_t \in \mathbb{R}^{h \times d_h}$ is a sum of rank-1 outer products. Unless otherwise stated we use the per-factor normalizations $s_Q = 1/R_Q$, $s_K = 1/R_K$, $s_V = 1/R_V$.

We make the batch/time/head/rank/value dimensions explicit and introduce the shorthands $D := d_h$ and $E := d_v$ (typically $E = D$):

$$\mathbf{A}_Q \in \mathbb{R}^{B \times T_q \times R_Q \times H}, \quad \mathbf{B}_Q \in \mathbb{R}^{B \times T_q \times R_Q \times D}, \quad \mathbf{A}_K \in \mathbb{R}^{B \times T_k \times R_K \times H}, \quad \mathbf{B}_K \in \mathbb{R}^{B \times T_k \times R_K \times D},$$

$$\mathbf{A}_V \in \mathbb{R}^{B \times T_k \times R_V \times H}, \quad \mathbf{B}_V \in \mathbb{R}^{B \times T_k \times R_V \times E}.$$

Indices $b, q, k, h, r, s, u, d, e$ denote batch, query position, key position, head, query-rank, key-rank, value-rank, feature ($D$), and value feature ($E$). We write $T := T_q = T_k$ for full-sequence attention; in decoding, $T_q = 1$ and we denote the cache length by $M = T_k$.

**Convention.** For a single token, the main text defines $\mathbf{A}_*(\mathbf{x}_t) \in \mathbb{R}^{R_* \times H}$ and $\mathbf{B}_*(\mathbf{x}_t) \in \mathbb{R}^{R_* \times D}$, with $\mathbf{Q}_t = \frac{1}{R_Q} \mathbf{A}_Q(\mathbf{x}_t)^\top \mathbf{B}_Q(\mathbf{x}_t)$. Accordingly, throughout this appendix we index $\mathbf{A}_Q$ as $\mathbf{A}_Q[b, q, r, h]$ (rank-major). Some implementations may store $\mathbf{A}_*$ transposed as $(H \times R_*)$ for memory layout, this is equivalent, since all uses contract over the rank index.

**High-level idea.** We first compute head-*shared* feature-space dot products between $\mathbf{B}_Q$ and $\mathbf{B}_K$, then mix them with head-specific $\mathbf{A}_Q, \mathbf{A}_K$ to obtain logits, apply the masked softmax, and finally aggregate values via $\mathbf{A}_V, \mathbf{B}_V$. This ordering avoids materializing any $T_q \times h \times D$ queries/keys/values.

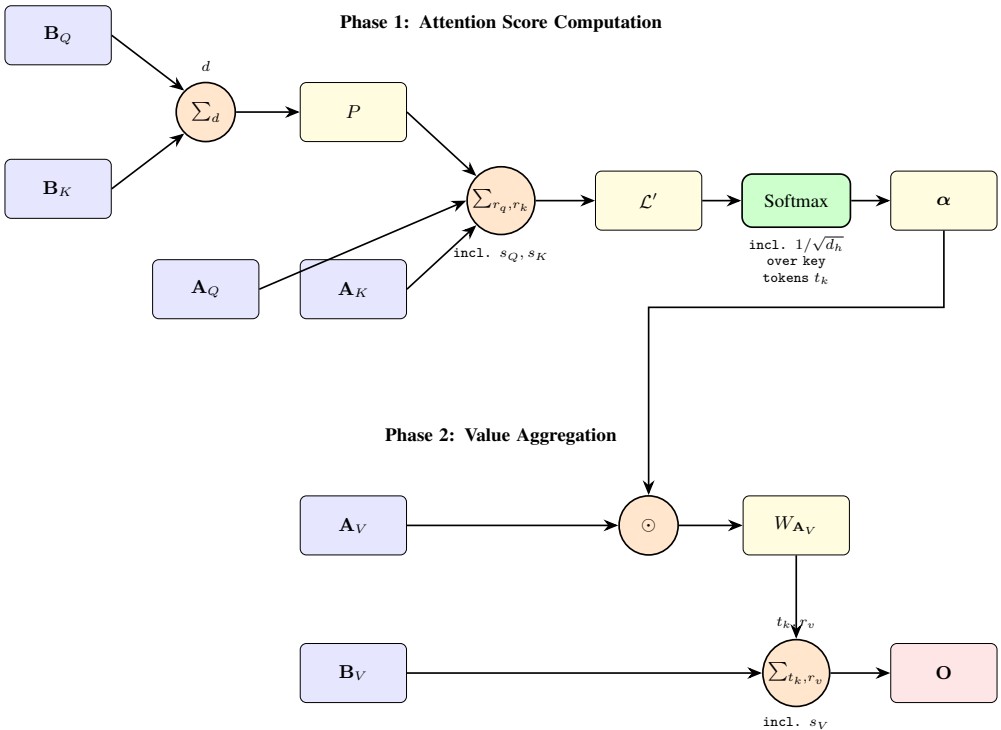

Figure 6: Specialized TPA computation *without* materializing $\mathbf{Q}, \mathbf{K}, \mathbf{V}$. **Phase 1** (top): compute head-shared feature-space dot products $P[b, q, k, r, s] = \langle \mathbf{B}_Q[b, q, r, :], \mathbf{B}_K[b, k, s, :] \rangle$ and mix them with head-specific factors $\mathbf{A}_Q, \mathbf{A}_K$ to obtain logits $\mathcal{L}[b, h, q, k]$. **Phase 2** (bottom): apply the causal/padding mask and softmax to get $\alpha[b, h, q, k]$, then aggregate values via $\mathbf{A}_V, \mathbf{B}_V$. Scalings $s_Q, s_K, s_V$ and $1/\sqrt{D}$ are folded into the corresponding phases. Dropout is omitted for clarity. Batch $B$, heads $H$, ranks $R_Q, R_K, R_V$, and feature dims $D, E$ are indicated in the nodes.

## A.1 Direct computation in factor space

**Single head.** For a fixed head $h \in [H]$ and token indices $(q, k)$, using $s_Q = 1/R_Q$ and $s_K = 1/R_K$ we have

$$\left[\mathbf{Q}^{(h)}(\mathbf{K}^{(h)})^\top\right]_{q,k} = \frac{1}{R_Q R_K}\sum_{r=1}^{R_Q}\sum_{s=1}^{R_K} a_{q,h}^{Q,(r)}(\mathbf{x}_q)\, a_{k,h}^{K,(s)}(\mathbf{x}_k)\, \langle \mathbf{b}^{Q,(r)}(\mathbf{x}_q), \mathbf{b}^{K,(s)}(\mathbf{x}_k)\rangle, \quad \text{(A.1)}$$

and for values (with $s_V = 1/R_V$), $\mathbf{V}_k^{(h)} = \frac{1}{R_V}\sum_{u=1}^{R_V} a_{k,h}^{V,(u)}(\mathbf{x}_k)\, \mathbf{b}^{V,(u)}(\mathbf{x}_k)$. The per-head attention output at query position $q$ is then $\sum_k \mathrm{softmax}\left(\frac{1}{\sqrt{D}}[\mathbf{Q}^{(h)}(\mathbf{K}^{(h)})^\top]_{q,:}\right)_k \mathbf{V}_k^{(h)}$.

**Multi-head with head-shared feature dot-products.** Define head-shared feature-space dot products $P[b, q, k, r, s] = \langle \mathbf{B}_Q[b, q, r, :], \mathbf{B}_K[b, k, s, :]\rangle$. With $\mathcal{S} = \frac{s_Q s_K}{\sqrt{D}}$, we compute

$$\mathcal{L}[b, h, q, k] = \mathcal{S}\sum_{r=1}^{R_Q}\sum_{s=1}^{R_K} \mathbf{A}_Q[b, q, r, h]\, \mathbf{A}_K[b, k, s, h]\, P[b, q, k, r, s], \quad \text{(A.2)}$$

$$\alpha[b, h, q, k] = \mathrm{Softmax}_k(\mathcal{L}[b, h, q, k] + \mathsf{mask}[b, q, k]),$$

$$\mathbf{O}[b, h, q, e] = s_V \sum_{k=1}^{T_k}\sum_{u=1}^{R_V} \alpha[b, h, q, k]\, \mathbf{A}_V[b, k, u, h]\, \mathbf{B}_V[b, k, u, e]. \quad \text{(A.3)}$$

Here $\mathsf{mask}[b, q, k] \in \{0, -\infty\}$ is an *additive* mask in logit space that enforces causality and padding. Eqs. (A.2)–(A.3) make explicit that (i) feature-space dot products $P$ are *head-shared*, and (ii) the rank normalizations $s_Q, s_K, s_V$ can be absorbed into the corresponding factor tensors (or into the scalar prefactors) without changing the computed attention output.

## A.2 Complexity: materialized vs. specialized computation

We compare two execution strategies. (i) *Naïve/materialized:* form $\mathbf{Q}, \mathbf{K}, \mathbf{V}$ explicitly and call standard kernels. (ii) *Specialized:* compute via Eq. (A.2)–(A.3) using head-shared feature-dot products and per-head rank contractions.

**Standard MHA (baseline).** Ignoring projections, full-sequence attention uses $\Theta(BHT^2 D)$ FLOPs for scores *and* $\Theta(BHT^2 D)$ for value aggregation, i.e., $\mathcal{F}_{\mathrm{MHA}} = 2\Theta(BHT^2 D)$.

**TPA (materialized).** Forming $\mathbf{Q}, \mathbf{K}, \mathbf{V}$ from factors costs $\Theta(BTHD(R_Q + R_K + R_V))$ after the linear projections; subsequent attention uses the same $2\Theta(BHT^2 D)$ as MHA.

**TPA (specialized).** Using Eqs. (A.2)–(A.3) and writing $T_q = T_k = T$, the dominant FLOPs are

$$\underbrace{\Theta(BT^2 R_Q R_K D)}_{\text{feature dots } P} + \underbrace{\Theta(BHT^2 R_Q R_K)}_{\text{per-head rank combine}} + \underbrace{\Theta(BHT^2 R_V E)}_{\text{value aggregation}}.$$

Compared to $\mathcal{F}_{\mathrm{MHA}}$, the specialized path reduces FLOPs whenever

$$R_Q R_K D + H R_Q R_K + H R_V E < 2HD. \quad \text{(A.4)}$$

Dividing by $HD$ yields $(R_Q R_K/H) + (R_Q R_K/D) + R_V(E/D) < 2$. For $E = D$ and small ranks (e.g., $R_Q = R_K = R_V = 1$), the inequality holds for typical $H, D \geq 2$ and the benefit grows with larger $H$ or $D$.

**Memory traffic and peak working set.** For full-sequence attention the naive path streams $\mathbf{Q}, \mathbf{K}, \mathbf{V}$ of size $\Theta(BTHD)$ each. The specialized path streams factors only and needs the head-shared $P$ tiles of size $\Theta(B T_q^{\mathrm{tile}} T_k^{\mathrm{tile}} R_Q R_K)$ plus per-head tiles for the rank combine/value aggregation. In decoding with cache length $M$, the factorized KV cache uses $(R_K + R_V)(h + D)$ numbers per token (cf. Section 3.3), vs. $2hD$ for MHA; this reduction directly lowers memory bandwidth pressure.

---

**Algorithm 1** Specialized TPA (no explicit $\mathbf{Q}, \mathbf{K}, \mathbf{V}$; causal)

---

**Require:** $\mathbf{A}_Q \in \mathbb{R}^{B \times T_q \times R_Q \times H}$, $\mathbf{B}_Q \in \mathbb{R}^{B \times T_q \times R_Q \times D}$
**Require:** $\mathbf{A}_K \in \mathbb{R}^{B \times T_k \times R_K \times H}$, $\mathbf{B}_K \in \mathbb{R}^{B \times T_k \times R_K \times D}$
**Require:** $\mathbf{A}_V \in \mathbb{R}^{B \times T_k \times R_V \times H}$, $\mathbf{B}_V \in \mathbb{R}^{B \times T_k \times R_V \times E}$
**Require:** scales $s_Q{=}1/R_Q$, $s_K{=}1/R_K$, $s_V{=}1/R_V$; mask $\mathsf{mask} \in \{0, -\infty\}^{B \times T_q \times T_k}$
**Ensure:** $\mathbf{O} \in \mathbb{R}^{B \times T_q \times H \times E}$
 1: $P \leftarrow \mathrm{einsum}(\text{"bqrd,bksd->bqkrs"}, \mathbf{B}_Q, \mathbf{B}_K)$       $\triangleright \in \mathbb{R}^{B \times T_q \times T_k \times R_Q \times R_K}$
 2: $\mathcal{L} \leftarrow (s_Q s_K / \sqrt{D}) \cdot \mathrm{einsum}(\text{"bqrh,bksh,bqkrs->bhqk"}, \mathbf{A}_Q, \mathbf{A}_K, P)$
 3: $\mathcal{L} \leftarrow \mathcal{L} + \mathrm{broadcast}(\mathsf{mask})$          $\triangleright$ causal/padding mask
 4: $\alpha \leftarrow \mathrm{Softmax}_k(\mathcal{L})$       $\triangleright \in \mathbb{R}^{B \times H \times T_q \times T_k}$; online/LSE in practice
 5: $\mathbf{O} \leftarrow s_V \cdot \mathrm{einsum}(\text{"bhqk,bkuh,bkue->bhqe"}, \alpha, \mathbf{A}_V, \mathbf{B}_V)$
 6: **return** $\mathrm{transpose}(\mathbf{O}, \text{"bhqe"} \to \text{"bqhe"})$

---

### A.3 Complexity of the specialized path

Combining the terms above gives complexity $\mathcal{F}_{\text{TPA-spec}} = \Theta(BT^2 R_Q R_K D) + \Theta(BHT^2 R_Q R_K) + \Theta(BHT^2 R_V E)$, with the speed condition Eq. (A.4).

For a single query ($T_q{=}1$) against a cache of length $M$, the specialized FLOPs are

$$\Theta(B\,M\,R_Q R_K\,D) \;+\; \Theta(B\,H\,M\,R_Q R_K) \;+\; \Theta(B\,H\,M\,R_V\,E),$$

while MHA uses $2\,\Theta(BHMD)$. This matches the asymptotics embodied in FLASHTPA (Section 5) and explains the regimes where $R_Q{\ll}D$ and $R_K{=}R_V \in \{1, 2\}$ yield the largest gains.

We apply the *causal mask* before softmax and use an *online log-sum-exp* update for numerical stability (as in FlashAttention). The intermediate $P \in \mathbb{R}^{B \times T_q \times T_k \times R_Q \times R_K}$ is evaluated *blockwise* in $T_k$ to keep peak memory linear in the block size; the same blocking naturally fuses with the masked softmax and the value aggregation step.

The constants $s_Q, s_K, s_V$ can be absorbed into either $\mathbf{A}_{(\cdot)}$ or $\mathbf{B}_{(\cdot)}$ at training time. We expose them explicitly only to make Eq. (A.4) transparent; The choice has no effect on softmax invariance or gradients.

The Triton kernel in Section 5 implements the blocked computation of $P$, the masked online softmax over $k$, and the fused value aggregation, mirroring Algorithm 1. This avoids creating any $\mathbf{Q}, \mathbf{K}, \mathbf{V}$ or full $T_q \times T_k$ temporaries beyond working tiles.

Compared with $2\,\Theta(BHT^2 D)$ for MHA, the specialized path improves with small $(R_Q, R_K, R_V)$ and benefits further from pre-rotating $\mathbf{B}_K$ for RoPE (cf. Section 3.2), which removes per-step rotations in decoding. Practical speed also depends on tiling, memory bandwidth, and kernel fusion; our measured gains in Section 6.2 align with the regime predicted by Eq. (A.4).

### A.4 Inference-time decoding cost across mechanisms

In autoregressive decoding, we generate the output for the current token $\mathbf{x}_T$ given cached keys and values from $T{-}1$ previous tokens. We analyze the FLOPs for computing the attention output for this single query token and use $M$ for the current cache length. For all mechanisms, we analyze the total Floating Point Operations (FLOPs) and the number of parameters in the attention layer, including the cost of projecting the current token's hidden state $\mathbf{x}_T$ into its respective Query, Key, and Value representations. The parameter count formulas are taken from Table 1.

For **Multi-Head Attention (MHA)**, with $H$ query heads and $H$ distinct Key/Value heads, the complexity is determined by the dot-product attention and value aggregation steps.

- Projection: Projecting $\mathbf{x}_T$ to get a query, key, and value vector for each of the $H$ heads costs $\Theta(d_{\text{model}} H d_h)$.
- Attention: Dot products and value aggregation over a cache of length $M$ cost $\Theta(2MHd_h)$ (ignoring softmax constants).
- Total MHA: The complexity is $\Theta(d_{\text{model}} H d_h + 2MHd_h)$.

**Multi-Query Attention (MQA)** uses $H$ query heads but shares a single Key/Value head ($H_{kv} = 1$). The arithmetic complexity remains the same as MHA for the same number of query heads.

- **Projection:** Projecting for $H$ query heads and 1 shared K/V head costs $\Theta(d_{\text{model}}(Hd_h + 2d_h))$.
- **Attention:** The interaction with the cache costs $\Theta(2MHd_h)$.
- **Total MQA:** The complexity is $\Theta(d_{\text{model}}d_h(H + 2) + 2MHd_h)$.

**Grouped-Query Attention (GQA)** uses $H$ query heads and $G$ Key/Value head groups ($H_{kv} = G$). The arithmetic complexity is also identical to MHA.

- **Projection:** Projecting for $H$ query heads and $G$ K/V head groups costs $\Theta(d_{\text{model}}(Hd_h + 2Gd_h))$.
- **Attention:** The interaction with the cache costs $\Theta(2MHd_h)$.
- **Total GQA:** The complexity is $\Theta(d_{\text{model}}d_h(H + 2G) + 2MHd_h)$.

MQA and GQA significantly reduce the KV cache *size* and memory bandwidth compared to MHA. While the arithmetic FLOP count for the core attention computation (dot products and weighted sums) is $2MHd_h$ for all three (for fixed $H, d_h$), practical speedups for MQA/GQA arise from improved memory locality due to smaller K/V caches.

**Multi-Head Latent Attention (MLA)**, as described in Appendix F.3, uses $H$ heads. Each head's (up-projected) query/key vectors have dimension $d_h + d_h^R$. During decoding, however, the score computation against the cache can be decomposed into (i) a dot product in the cached latent space $\mathbb{R}^{d_c}$ for the content part and (ii) an additional RoPE dot product in $\mathbb{R}^{d_h^R}$ for the positional part. Concretely, MLA caches $\mathbf{c}_s^{KV} \in \mathbb{R}^{d_c}$ per past token $s$, aggregates values in $\mathbb{R}^{d_c}$, and then up-projects once per step.

- **Cached state:** MLA caches the compressed KV latent $\mathbf{c}_s^{KV} \in \mathbb{R}^{d_c}$ and the shared RoPE key component $\mathbf{k}_s^R \in \mathbb{R}^{d_h^R}$ per past token $s$.
- **Projection (current token):** Computing the query latents and the new cache entry (up to constant factors) costs
$$\Theta\big(d_{\text{model}}d_c' + d_c'H(d_h + d_h^R) + d_{\text{model}}(d_c + d_h^R)\big),$$
corresponding to forming $\mathbf{c}^Q, \mathbf{Q}^C, \mathbf{Q}^R$, and computing/storing $\mathbf{c}^{KV}$ and $\mathbf{k}^R$ for the current token.
- **Attention (cache interaction):** Using the identity $\mathbf{q}_{t,i}^{C\top}\mathbf{k}_{s,i}^C = (\boldsymbol{W}_i^{UK}\mathbf{q}_{t,i}^C)^\top \mathbf{c}_s^{KV}$, the score against each cached token can be computed via a dot product in $\mathbb{R}^{d_c}$ plus the RoPE dot product in $\mathbb{R}^{d_h^R}$. The latent value can be aggregated in $\mathbb{R}^{d_c}$ and then up-projected once. The dominant cache-dependent cost is
$$\Theta\big(MH(2d_c + d_h^R)\big),$$
up to lower-order per-step terms such as $\Theta(Hd_cd_h)$.
- **Total MLA:** $\Theta\big(d_{\text{model}}d_c' + d_c'H(d_h + d_h^R) + d_{\text{model}}(d_c + d_h^R) + MH(2d_c + d_h^R)\big)$.

**TPA.** We use the FlashTPA Decoding algorithm (Algorithm 2) for FLOPs analysis, with $N = 1$ query token, $M$ cached items, $D$ as feature dimension for $\mathbf{B}_Q/\boldsymbol{b}^K$ (typically $d_h$), and $E$ for $\boldsymbol{b}^V$ (typically $d_h$). For ranks $(R_Q, R_K, R_V)$:

- **Projection:** Projecting the current token $\mathbf{x}_T$ to all Q/K/V factors costs $\Theta\big(d_{\text{model}}(R_Q + R_K + R_V)(H + d_h)\big)$.
- **Attention (cache interaction):** Using Algorithm 2 with cache length $M$, the dominant cache-dependent FLOPs are
$$\Theta\big(M(R_QR_KD + HR_QR_K + HR_VE)\big),$$
up to lower-order terms (masking/element-wise products and online-softmax bookkeeping).
- **Total for TPA decoding:** $\Theta\big(d_{\text{model}}(R_Q + R_K + R_V)(H + d_h) + M(R_QR_KD + HR_QR_K + HR_VE)\big)$.

**Example Comparison I.**

We compare the total Floating Point Operations (FLOPs) required to process a single token during autoregressive inference. This analysis separates the initial, constant projection cost from the attention cost, which scales linearly with the cache length $M$.

The following parameters are used for the comparison:

- Model Dimension: $d_{\text{model}} = 2048$
- Heads: $H = 32$
- Head Dimension: $d_h = 64$ (so $D = E = d_h$)

- GQA Groups: $G = 4$
- MLA Dimensions: $d_c = 256$, $d_h^R = 32$, and $d_c' = 768$

**MHA (16.8M parameters):**

$$\text{Parameters} = 4d_{\text{model}}Hd_h = 4 \cdot 2048 \cdot (32 \cdot 64) \approx 16.8 \times 10^6$$

$$\text{Projection} = 3 \cdot d_{\text{model}} \cdot H \cdot d_h = 3 \cdot 2048 \cdot 32 \cdot 64 \approx 12.6 \times 10^6$$

$$\text{Attention} = 2 \cdot M \cdot H \cdot d_h = 4096M$$

**GQA ($G = 4$, 9.4M parameters):**

$$\text{Parameters} = d_{\text{model}}d_h(2H + 2G) = 2048 \cdot 64 \cdot (2 \cdot 32 + 2 \cdot 4) \approx 9.4 \times 10^6$$

$$\text{Projection} = d_{\text{model}}(H + 2G)d_h = 2048 \cdot (32 + 8) \cdot 64 \approx 5.2 \times 10^6$$

$$\text{Attention} = 2 \cdot M \cdot H \cdot d_h = 4096M$$

**MLA (9.8M parameters):**

$$\text{Parameters} = 768(2048 + 2048 + 1024) + 2048(32 + 2048) + 256(2048 + 4096) \approx 9.8 \times 10^6$$

$$\begin{aligned}\text{Projection} &\approx d_{\text{model}}d_c' + d_c'H(d_h + d_h^R) + d_{\text{model}}(d_c + d_h^R) + Hd_cd_h \\ &= 2048 \cdot 768 + 768 \cdot 32 \cdot (64 + 32) + 2048 \cdot (256 + 32) + 32 \cdot 256 \cdot 64 \\ &\approx 5.0 \times 10^6\end{aligned}$$

$$\text{Attention} = M \cdot H \cdot (2d_c + d_h^R) = M \cdot 32 \cdot (512 + 32) = 17408M$$

**TPA ($R_Q = 16, R_K = 1, R_V = 1$, 7.7M parameters):**

$$\text{Parameters} = d_{\text{model}}(16 + 1 + 1)(H + d_h) + d_{\text{model}}Hd_h = 2048(18)(96) + 2048^2 \approx 7.7 \times 10^6$$

$$\text{Projection} = d_{\text{model}}(16 + 1 + 1)(H + d_h) = 2048 \cdot (18) \cdot (96) \approx 3.5 \times 10^6$$

$$\text{Attention} = M \cdot [1(1536) + 1(2048)] = 3584M$$

**TPA ($R_Q = 16, R_K = 2, R_V = 2$, 8.1M parameters):**

$$\text{Parameters} = d_{\text{model}}(16 + 2 + 2)(H + d_h) + d_{\text{model}}Hd_h = 2048(20)(96) + 2048^2 \approx 8.1 \times 10^6$$

$$\text{Projection} = d_{\text{model}}(16 + 2 + 2)(H + d_h) = 2048 \cdot (20) \cdot (96) \approx 3.9 \times 10^6$$

$$\text{Attention} = M \cdot [2(1536) + 2(2048)] = 7168M$$

**TPA ($R_Q = 8, R_K = 1, R_V = 1$, 6.2M parameters):**

$$\text{Parameters} = d_{\text{model}}(8 + 1 + 1)(H + d_h) + d_{\text{model}}Hd_h = 2048(10)(96) + 2048^2 \approx 6.2 \times 10^6$$

$$\text{Projection} = d_{\text{model}}(8 + 1 + 1)(H + d_h) = 2048 \cdot (10) \cdot (96) \approx 2.0 \times 10^6$$

$$\text{Attention} = M \cdot [1(768) + 1(2048)] = 2816M$$

**TPA ($R_Q = 8, R_K = 2, R_V = 2$, 6.6M parameters):**

$$\text{Parameters} = d_{\text{model}}(8 + 2 + 2)(H + d_h) + d_{\text{model}}Hd_h = 2048(12)(96) + 2048^2 \approx 6.6 \times 10^6$$

$$\text{Projection} = d_{\text{model}}(8 + 2 + 2)(H + d_h) = 2048 \cdot (12) \cdot (96) \approx 2.4 \times 10^6$$

$$\text{Attention} = M \cdot [2(768) + 2(2048)] = 5632M$$

The analysis shows that TPA with low ranks offers a favorable trade-off. Reducing the query rank ($R_Q$) from 16 to 8 further decreases both the projection and attention costs, making the TPA ($R_Q=8, R_K=1, R_V=1$) configuration the most computationally efficient in this comparison. Increasing key/value ranks (e.g., to $R_K=2, R_V=2$) raises the attention cost linearly, remaining competitive with MHA for sufficiently long contexts where kernel fusion and blocking amortize memory traffic.

**Example Comparison II.**

We now repeat the analysis for a larger model configuration to observe how these trade-offs scale.

The following parameters for a larger model are used for this comparison:

- Model Dimension: $d_{\text{model}} = 4096$
- Heads: $H = 32$
- Head Dimension: $d_h = 128$ (so $D = E = d_h$)
- GQA Groups: $G = 4$
- MLA Dimensions: $d_c = 512$, $d_h^R = 64$, and $d_c' = 1536$

**MHA (67.1M parameters):**
$$\text{Parameters} = 4d_{\text{model}}Hd_h = 4 \cdot 4096^2 \approx 67.1 \times 10^6$$
$$\text{Projection} = 3 \cdot 4096 \cdot 32 \cdot 128 \approx 50.3 \times 10^6$$
$$\text{Attention} = 2 \cdot M \cdot 32 \cdot 128 = 8192M$$

**GQA ($G = 4$, 37.7M parameters):**
$$\text{Parameters} = d_{\text{model}}d_h(2H + 2G) = 4096 \cdot 128 \cdot (2 \cdot 32 + 2 \cdot 4) \approx 37.7 \times 10^6$$
$$\text{Projection} = 4096 \cdot (32 + 8) \cdot 128 \approx 21.0 \times 10^6$$
$$\text{Attention} = 2 \cdot M \cdot 32 \cdot 128 = 8192M$$

**MLA (39.1M parameters):**

$\text{Parameters} = 1536(4096 + 4096 + 2048) + 4096(64 + 4096) + 512(4096 + 8192) \approx 39.1 \times 10^6$

$$\begin{aligned}\text{Projection} &\approx d_{\text{model}}d_c' + d_c'H(d_h + d_h^R) + d_{\text{model}}(d_c + d_h^R) + Hd_cd_h \\ &= 4096 \cdot 1536 + 1536 \cdot 32 \cdot (128 + 64) + 4096 \cdot (512 + 64) + 32 \cdot 512 \cdot 128 \\ &\approx 20.2 \times 10^6\end{aligned}$$

$\text{Attention} = M \cdot 32 \cdot (1024 + 64) = 34816M$

**TPA ($R_Q = 16, R_K = 1, R_V = 1$, 28.6M parameters):**
$$\text{Parameters} = 4096(18)(160) + 4096^2 \approx 28.6 \times 10^6$$
$$\text{Projection} = 4096 \cdot (16 + 1 + 1) \cdot (32 + 128) \approx 11.8 \times 10^6$$
$$\text{Attention} = M \cdot [1(2560) + 1(4096)] = 6656M$$

**TPA ($R_Q = 16, R_K = 2, R_V = 2$, 29.9M parameters):**
$$\text{Parameters} = 4096(20)(160) + 4096^2 \approx 29.9 \times 10^6$$
$$\text{Projection} = 4096 \cdot (16 + 2 + 2) \cdot (32 + 128) \approx 13.1 \times 10^6$$
$$\text{Attention} = M \cdot [2(2560) + 2(4096)] = 13312M$$

**TPA ($R_Q = 8, R_K = 1, R_V = 1$, 23.3M parameters):**
$$\text{Parameters} = 4096(10)(160) + 4096^2 \approx 23.3 \times 10^6$$
$$\text{Projection} = 4096 \cdot (8 + 1 + 1) \cdot (32 + 128) \approx 6.6 \times 10^6$$

$$\text{Attention} = M \cdot [1(1280) + 1(4096)] = 5376M$$

**TPA ($R_Q = 8, R_K = 2, R_V = 2$, 24.6M parameters):**

$$\text{Parameters} = 4096(12)(160) + 4096^2 \approx 24.6 \times 10^6$$
$$\text{Projection} = 4096 \cdot (8 + 2 + 2) \cdot (32 + 128) \approx 7.9 \times 10^6$$
$$\text{Attention} = M \cdot [2(1280) + 2(4096)] = 10752M$$

For this larger configuration, TPA ($R_Q$=8, $R_K$=1, $R_V$=1) remains the clear leader in computational efficiency, with the lowest projection and attention costs. This highlights the value of tuning TPA ranks to balance expressiveness against compute.

**Example Comparison III.**

Then we analyze a very large model configuration (e.g. MoE model with 1∼2T parameters) to examine the scaling properties of each architecture, where $d_{\text{model}} \neq H \cdot d_h$ to align MLA with other attention mechanisms. We also denote the number of parameters in the attention part for each layer.

The following parameters are used for this comparison:

- Model Dimension: $d_{\text{model}} = 7168$
- Heads: $H = 64$
- Head Dimension: $d_h = 128$ (so $D = E = d_h$)
- GQA Groups: $G = 8$
- MLA Dimensions: $d_c = 512$, $d_h^R = 64$, and $d_c' = 1536$

**MHA (235M parameters):**

$$\text{Parameters} = 4d_{\text{model}}Hd_h = 4 \cdot 7168 \cdot 8192 \approx 235 \times 10^6$$
$$\text{Projection} = 3 \cdot 7168 \cdot 64 \cdot 128 \approx 176.2 \times 10^6$$
$$\text{Attention} = 2 \cdot M \cdot 64 \cdot 128 = 16384M$$

**GQA ($G = 8$, 132M parameters):**

$$\text{Parameters} = d_{\text{model}}d_h(2H + 2G) = 7168 \cdot 128 \cdot (2 \cdot 64 + 2 \cdot 8) \approx 132 \times 10^6$$
$$\text{Projection} = 7168 \cdot (64 + 16) \cdot 128 \approx 73.4 \times 10^6$$
$$\text{Attention} = 2 \cdot M \cdot 64 \cdot 128 = 16384M$$

**MLA (101M parameters):**

$$\text{Parameters} = 1536(7168 + 8192 + 4096) + 7168(64 + 8192) + 512(7168 + 16384) \approx 101 \times 10^6$$
$$\begin{aligned}\text{Projection} &\approx d_{\text{model}}d_c' + d_c'H(d_h + d_h^R) + d_{\text{model}}(d_c + d_h^R) + Hd_cd_h \\ &= 7168 \cdot 1536 + 1536 \cdot 64 \cdot (128 + 64) + 7168 \cdot (512 + 64) + 64 \cdot 512 \cdot 128 \\ &\approx 38.2 \times 10^6\end{aligned}$$
$$\text{Attention} = M \cdot 64 \cdot (2 \cdot 512 + 64) = 69632M$$

**TPA ($R_Q = 16, R_K = 1, R_V = 1$, 83M parameters):**

$$\text{Parameters} = 7168(18)(192) + 7168 \cdot 8192 \approx 83.5 \times 10^6$$
$$\text{Projection} = 7168 \cdot (16 + 1 + 1) \cdot (64 + 128) \approx 24.8 \times 10^6$$
$$\text{Attention} = M \cdot [1(3072) + 1(8192)] = 11264M$$

**TPA ($R_Q = 16, R_K = 2, R_V = 2$, 86.2M parameters):**

$$\text{Parameters} = 7168(20)(192) + 7168 \cdot 8192 \approx 86.2 \times 10^6$$
$$\text{Projection} = 7168 \cdot (16 + 2 + 2) \cdot (64 + 128) \approx 27.5 \times 10^6$$
$$\text{Attention} = M \cdot [2(3072) + 2(8192)] = 22528M$$

**TPA ($R_Q = 8, R_K = 1, R_V = 1$, 72.5M parameters):**

$$\text{Parameters} = 7168(10)(192) + 7168 \cdot 8192 \approx 72.5 \times 10^6$$

$$\text{Projection} = 7168 \cdot (8 + 1 + 1) \cdot (64 + 128) \approx 13.8 \times 10^6$$

$$\text{Attention} = M \cdot [1(1536) + 1(8192)] = 9728M$$

**TPA ($R_Q = 8, R_K = 2, R_V = 2$, 75.2M parameters):**

$$\text{Parameters} = 7168(12)(192) + 7168 \cdot 8192 \approx 75.2 \times 10^6$$

$$\text{Projection} = 7168 \cdot (8 + 2 + 2) \cdot (64 + 128) \approx 16.5 \times 10^6$$

$$\text{Attention} = M \cdot [2(1536) + 2(8192)] = 19456M$$

At this very large scale, the cost of MHA projections becomes prohibitive. While MLA's projection cost can be competitive, its attention cost scales with $(2d_c + d_h^R)$ and exceeds MHA for long sequences. TPA with low ranks ($R_Q=8, R_K=1, R_V=1$) yields the lowest attention cost and a substantially smaller projection cost, strengthening its advantage as model size increases.

# B  More on FlashTPA Decoding Algorithm

In this section, we present FlashTPA for decoding in a hardware–friendly, numerically stable form and extend it to general key/value ranks $R_K, R_V \geq 1$. The algorithm computes attention *without* materializing $\mathbf{Q}, \mathbf{K}, \mathbf{V}$ or the full $N \times M$ attention matrix, by (i) forming head-shared feature–space dot products, (ii) mixing them with head-specific factors to obtain logits as in Eq. (A.2), and (iii) aggregating values as in Eq. (A.3) in a single online softmax pass.

**Notation and shapes.** We allow $N$ query positions but decoding uses $N=1$. Let $B$ be batch, $M$ the cache length, $H$ heads, $R_Q, R_K, R_V$ ranks, and $D, E$ feature sizes (typically $D=E=d_h$). Inputs:

$$\mathbf{A}_Q \in \mathbb{R}^{B \times N \times R_Q \times H}, \ \mathbf{B}_Q \in \mathbb{R}^{B \times N \times R_Q \times D}, \ \mathbf{A}_K^{\text{cache}} \in \mathbb{R}^{B \times M \times R_K \times H}, \ \mathbf{B}_K^{\text{cache}} \in \mathbb{R}^{B \times M \times R_K \times D},$$

$$\mathbf{A}_V^{\text{cache}} \in \mathbb{R}^{B \times M \times R_V \times H}, \ \mathbf{B}_V^{\text{cache}} \in \mathbb{R}^{B \times M \times R_V \times E}.$$

We use scalings $s_Q=1/R_Q$, $s_K=1/R_K$, $s_V=1/R_V$, and $s_{\text{total}}=1/\sqrt{D}$. Let mask $\in \{0, -\infty\}^{B \times N \times M}$ encode causality/padding. If RoPE pre-rotation is used (Section 3.2), $\mathbf{B}_K^{\text{cache}}$ already includes positional phases; otherwise apply RoPE to $\mathbf{B}_K$ on load.

---

**Algorithm 2** FlashTPA Decoding (general $R_K, R_V$, masked, online-LSE)

---

**Require:** $\mathbf{A}_Q, \mathbf{B}_Q, \mathbf{A}_K^{\text{cache}}, \mathbf{B}_K^{\text{cache}}, \mathbf{A}_V^{\text{cache}}, \mathbf{B}_V^{\text{cache}}$, mask; $s_Q, s_K, s_V, s_{\text{total}}$
**Ensure:** $\mathbf{O} \in \mathbb{R}^{B \times N \times H \times E}$

1: Initialize $\mathbf{y} \leftarrow 0^{B \times H \times N \times E}$, $\quad \mathbf{s} \leftarrow 0^{B \times H \times N}$, $\quad \mathbf{m} \leftarrow (-\infty)^{B \times H \times N}$ $\quad\quad\quad \triangleright$ $\mathbf{s}$ accumulates $\sum \exp(\cdot)$; log-sum-exp is $\log \mathbf{s} + \mathbf{m}$
2: **for each** cache block $m{:}m+\Delta m \leq M$ **do**
3: $\quad$ Load $\mathbf{B}_{K,\text{blk}} \in \mathbb{R}^{B \times \Delta m \times R_K \times D}$, $\mathbf{A}_{K,\text{blk}} \in \mathbb{R}^{B \times \Delta m \times R_K \times H}$
4: $\quad$ Load $\mathbf{A}_{V,\text{blk}} \in \mathbb{R}^{B \times \Delta m \times R_V \times H}$, $\quad \mathbf{B}_{V,\text{blk}} \in \mathbb{R}^{B \times \Delta m \times R_V \times E}$, $\quad$ mask$_{\text{blk}} \in \mathbb{R}^{B \times N \times \Delta m}$
5: $\quad$ **(1) Head-shared feature dots:** $\mathbf{P} \leftarrow \text{einsum}(\text{"bnrd,bmsd}\rightarrow\text{bnmrs"}, \mathbf{B}_Q, \mathbf{B}_{K,\text{blk}})$ $\quad\quad \triangleright$ $\mathbb{R}^{B \times N \times \Delta m \times R_Q \times R_K}$
6: $\quad$ **(2) Per-head rank mixing to logits:**
7: $\quad$ $\mathcal{L}_{\text{blk}} \leftarrow (s_{\text{total}} s_Q s_K) \cdot \text{einsum}(\text{"bnrh,bmsh,bnmrs}\rightarrow\text{bhnm"}, \mathbf{A}_Q, \mathbf{A}_{K,\text{blk}}, \mathbf{P})$ $\quad\quad \triangleright$ $\mathbb{R}^{B \times H \times N \times \Delta m}$
8: $\quad$ $\mathcal{L}_{\text{blk}} \leftarrow \mathcal{L}_{\text{blk}} + \text{broadcast}(\text{mask}_{\text{blk}})$
9: $\quad$ **(3) Online softmax update (no $\alpha$ materialization):**
10: $\quad$ $\mathbf{m}_{\text{blk}} \leftarrow \max_m(\mathcal{L}_{\text{blk}})$; $\quad \mathbf{p}_{\text{blk}} \leftarrow \exp(\mathcal{L}_{\text{blk}} - \mathbf{m}_{\text{blk}})$; $\quad \mathbf{s}_{\text{blk}} \leftarrow \sum_m \mathbf{p}_{\text{blk}}$
11: $\quad$ **(4) Block value aggregation (fused over $m, u$):**
12: $\quad$ $\mathbf{y}_{\text{blk}} \leftarrow \text{einsum}(\text{"bhnm,bmuh,bmue}\rightarrow\text{bhne"}, \mathbf{p}_{\text{blk}}, \mathbf{A}_{V,\text{blk}}, \mathbf{B}_{V,\text{blk}})$ $\quad\quad \triangleright$ $\mathbb{R}^{B \times H \times N \times E}$
13: $\quad$ **(5) Fuse blocks with log-sum-exp:**
14: $\quad$ $\mathbf{m}_{\text{new}} \leftarrow \max(\mathbf{m}, \mathbf{m}_{\text{blk}})$; $\quad \mathbf{y} \leftarrow \exp(\mathbf{m} - \mathbf{m}_{\text{new}})[..., None] \odot \mathbf{y} + \exp(\mathbf{m}_{\text{blk}} - \mathbf{m}_{\text{new}})[..., None] \odot \mathbf{y}_{\text{blk}}$
15: $\quad$ $\mathbf{s} \leftarrow \exp(\mathbf{m} - \mathbf{m}_{\text{new}}) \odot \mathbf{s} + \exp(\mathbf{m}_{\text{blk}} - \mathbf{m}_{\text{new}}) \odot \mathbf{s}_{\text{blk}}$; $\quad \mathbf{m} \leftarrow \mathbf{m}_{\text{new}}$
16: **end for**
17: **return** $\mathbf{O} \leftarrow s_V \cdot \frac{\mathbf{y}}{\mathbf{s}[..., None]}$ permuted to $(B, N, H, E)$

---

Step (1)–(2) implements Eq. (A.2); step (4)–(5) implements Eq. (A.3) while fusing the masked softmax with value aggregation via online log-sum-exp (as in FlashAttention), thereby avoiding any $\alpha$ materialization. When $R_K = R_V = 1$ the contractions reduce to the simpler einsums in Figure 2.

**Complexity and working set.** Per block of $\Delta m$ cache items, the dominant FLOPs are

$$\Theta(B\,N\,\Delta m\,R_Q R_K D) \;+\; \Theta(B\,H\,N\,\Delta m\,R_Q R_K) \;+\; \Theta(B\,H\,\Delta m\,R_V E),$$

matching the specialized analysis in Appendix A.2 and the decoding bounds in Appendix A.4. Peak memory scales with tiles of $\mathbf{B}_K, \mathbf{A}_K, \mathbf{A}_V, \mathbf{B}_V$ and the small temporaries $\mathbf{P}$ and $\mathbf{V}_{\text{blk}}$; neither $\mathbf{Q}, \mathbf{K}, \mathbf{V}$ nor the full $N \times M$ attention matrix is formed.

**RoPE and masking.** If keys are pre-rotated (Eq. (3.5)), $\mathbf{B}_K^{\text{cache}}$ needs no decoding-time rotation. Otherwise apply RoPE to $\mathbf{B}_{K,\text{blk}}$ row-wise before step (1). The mask mask (zeros or $-\infty$) is added to logits in step (2) and supports both causal and padding masks.

## B.1    Triton FlashTPA Decoding Kernel

We implement the experiments using Triton [57]; Algorithm 3 sketches the kernel corresponding to Algorithm 2. The provided kernel outline specializes to the frequently used case $R_K = R_V = 1$; general ranks follow by tiling over $R_K, R_V$ and replacing the rank-1 vector–matrix products with the corresponding small GEMMs in steps $S1/S2$ and the value mixing path.

## B.2    Additional Experimental Results

The following figures present additional speed comparisons for different embedding dimensions, with $d_h = 64$ maintained. The y-axis represents $\log_2(\text{time})$ in seconds (lower is faster), and the x-axis represents $\log_2(\text{sequence length})$.

**Detailed Analysis of Figure 5 (Embedding Dimension 2048):** Figure 5 in the main paper depicts speed comparisons for an embedding dimension of 2048. The results indicate that FlashTPA (blue line) is highly competitive. Across all tested batch sizes (1 to 16) for $d_{\text{model}} = 2048$:

- MHA (orange line) is consistently the slowest mechanism, with its decoding time increasing most rapidly with sequence length.
- MQA (purple line) and GQA (green line) offer significant speedups over MHA and perform very similarly to each other, often overlapping in the plots.
- MLA (blue line) demonstrates strong performance, generally being faster than GQA, particularly at longer sequence lengths.
- FlashTPA shows excellent scalability. While at very short sequence lengths (e.g., $2^{12}$ to $2^{13}$), its performance is comparable to MQA/GQA and MLA, its decoding time increases at a notably slower rate with sequence length. Consequently, FlashTPA becomes significantly faster than GQA for sequences longer than approximately $2^{14}$.
- Compared to MLA, FlashTPA is consistently among the top two performers. In many instances, particularly at sequence lengths greater than $2^{14}$ or $2^{15}$, FlashTPA matches or slightly surpasses MLA in speed. The logarithmic scale for time suggests that these differences can be substantial in practice for very long contexts. For example, at a sequence length of $2^{19}$ across various batch sizes, FlashTPA often shows a visible advantage over MLA.

**Figure 7 (Embedding Dimension 3072):** With a larger embedding dimension of 3072, the relative performance trends observed in Figure 5 largely persist.

- FlashTPA (red line) remains one of the most efficient decoding methods. MHA (orange line) is consistently the slowest, while MQA (purple line) and GQA (green line) offer considerable improvements over MHA.
- MLA (blue line) and FlashTPA are the top two performers. FlashTPA consistently matches or exceeds the speed of MLA, particularly at longer sequence lengths (e.g., beyond $2^{15}$ or $2^{16}$ depending on the batch size). Its advantage often becomes more pronounced at the longest sequences tested ($2^{19}$). For instance, in batch size 1, TPA is clearly faster than MLA for sequence lengths $2^{16}$ and above. A similar trend is seen across other batch sizes, where TPA maintains a competitive edge or becomes superior at longer contexts.

---

**Algorithm 3** Triton FlashTPA Decoding Kernel

---

**Require:** Input Tensors: $\mathbf{A}_Q(B, N, R_Q, H)$, $\mathbf{a}^K(B, M, H)$, $\mathbf{a}^V(B, M, H)$, $\mathbf{B}_Q(B, N, R_Q, D)$, $\boldsymbol{b}^K(B, M, D)$, $\boldsymbol{b}^V(B, M, E)$
**Require:** Scaling factors: $s_{\text{total}}, s_Q, s_K, s_V$; Dimensions: $B, N(=1), M, H, R_Q, D, E$
**Require:** Kernel Block dims: $B_H, B_R, B_D, B_E$; Sequence Blocking: $M_{\text{block}}, M_{\text{chunk}}$
**Require:** Program IDs: $p_{id_B}, p_{id_H}, p_{id_M}$
**Ensure:** Partial Output $\mathbf{O}_{\text{partial}}(B, \text{Num}_M, N, H, E)$, Log-Sum-Exp $\mathbf{LSE}_{\text{partial}}(B, \text{Num}_M, H)$

1: $b \leftarrow p_{id_B}$; $h_{\text{start}} \leftarrow p_{id_H} \cdot B_H$
2: $m_{\text{block\_start}} \leftarrow p_{id_M} \cdot M_{\text{block}}$; $m_{\text{block\_end}} \leftarrow \min((p_{id_M} + 1) \cdot M_{\text{block}}, M)$
3:                      $\triangleright$ $B_H, B_R, B_D, B_E$ are tile sizes for dimensions H, R, D, E respectively.

4:                         $\triangleright$ Initialize accumulators for the head block
5: $\mathbf{o}_{\text{accum}} \leftarrow \mathbf{0}^{(E \times B_H)}$; $\mathbf{m}_{\text{max}} \leftarrow -\infty^{(B_H)}$; $\mathbf{s}_{\text{exp\_sum}} \leftarrow \mathbf{0}^{(B_H)}$; $c_{\text{scale}} \leftarrow s_{\text{total}} \cdot s_Q \cdot s_K$

6:                       $\triangleright$ Load query factors (fixed for this program as N=1)
7: Load $\mathbf{A}_{Q,\text{local}}^{(R_Q \times B_H)}$ from $\mathbf{A}_Q[b, 0, :, h_{\text{start}} \dots]$
8: Load $\mathbf{B}_{Q,\text{local}}^{(D \times R_Q)}$ from $\mathbf{B}_Q[b, 0, :, :]$    $\triangleright$ Dimensions may be transposed after loading for matmul

9:                   $\triangleright$ Iterate over $M_{\text{chunk}}$-sized chunks within the K/V block
10: **for** $m_{\text{chunk\_start}}$ from $m_{\text{block\_start}}$ to $m_{\text{block\_end}} - 1$ step $M_{\text{chunk}}$ **do**
11:     $m_{\text{chunk\_end}} \leftarrow \min(m_{\text{chunk\_start}} + M_{\text{chunk}}, m_{\text{block\_end}})$
12:     $M_{\text{curr\_chunk}} \leftarrow m_{\text{chunk\_end}} - m_{\text{chunk\_start}}$
13:                       $\triangleright$ Load K/V factors for the current chunk
14:     Load     $\mathbf{a}_{\text{chunk}}^K(M_{\text{curr\_chunk}}, B_H)$;     $\mathbf{a}_{\text{chunk}}^V(M_{\text{curr\_chunk}}, B_H)$;     $\boldsymbol{b}_{\text{chunk}}^K(M_{\text{curr\_chunk}}, D)$; $\boldsymbol{b}_{\text{chunk}}^V(E, M_{\text{curr\_chunk}})$        $\triangleright$ Layouts optimized for memory access and matmuls
15:     $\boldsymbol{b}_{\text{chunk}}^V \leftarrow \boldsymbol{b}_{\text{chunk}}^V \cdot s_V$
16:                       $\triangleright$ Core TPA Score Calculation for the chunk
17:     $S1_{\text{chunk}} \leftarrow \text{MatMul}(\boldsymbol{b}_{\text{chunk}}^K, \mathbf{B}_{Q,\text{local}})$             $\triangleright$ Shape: $(M_{\text{curr\_chunk}}, R_Q)$
18:     $S2_{\text{chunk}} \leftarrow \text{MatMul}(S1_{\text{chunk}}, \mathbf{A}_{Q,\text{local}})$             $\triangleright$ Shape: $(M_{\text{curr\_chunk}}, B_H)$
19:     $S3_{\text{chunk}} \leftarrow S2_{\text{chunk}} \odot \mathbf{a}_{\text{chunk}}^K \cdot c_{\text{scale}}$             $\triangleright$ Shape: $(M_{\text{curr\_chunk}}, B_H)$
20:                       $\triangleright$ Online Softmax Update for the chunk
21:     $\mathbf{m}_{\text{max\_local}} \leftarrow \max_{axis=0}(S3_{\text{chunk}})$             $\triangleright$ Shape: $(B_H)$
22:     $\mathbf{m}_{\text{max\_new}} \leftarrow \max(\mathbf{m}_{\text{max}}, \mathbf{m}_{\text{max\_local}})$
23:     $\mathbf{p}_{\text{num}} \leftarrow \exp(S3_{\text{chunk}} - \mathbf{m}_{\text{max\_new}}[\text{None}, :])$
24:     $\mathbf{s}_{\text{exp\_sum\_local}} \leftarrow \sum_{axis=0}(\mathbf{p}_{\text{num}})$
25:     $\mathbf{p}_{\text{weighted\_av}} \leftarrow (\mathbf{p}_{\text{num}}/\mathbf{s}_{\text{exp\_sum\_local}}[\text{None}, :]) \odot \mathbf{a}_{\text{chunk}}^V$
26:     $\mathbf{o}_{\text{chunk}} \leftarrow \text{MatMul}(\boldsymbol{b}_{\text{chunk}}^V, \mathbf{p}_{\text{weighted\_av}})$             $\triangleright$ Shape: $(E, B_H)$
27:                       $\triangleright$ Update global (M-block level) accumulators
28:     $\mathbf{s}_{\text{exp\_sum\_prev\_rescaled}} \leftarrow \mathbf{s}_{\text{exp\_sum}} \cdot \exp(\mathbf{m}_{\text{max}} - \mathbf{m}_{\text{max\_new}})$
29:     $\mathbf{s}_{\text{exp\_sum}} \leftarrow \mathbf{s}_{\text{exp\_sum\_prev\_rescaled}} + \mathbf{s}_{\text{exp\_sum\_local}}$
30:     $\text{ratio} \leftarrow \mathbf{s}_{\text{exp\_sum\_local}}/\mathbf{s}_{\text{exp\_sum}}$             $\triangleright$ This is $\mathbf{s}_{\text{exp\_sum\_local}}/\mathbf{s}_{\text{exp\_sum\_new}}$
31:     $\mathbf{o}_{\text{accum}} \leftarrow (1 - \text{ratio}) \cdot \mathbf{o}_{\text{accum}} + \text{ratio} \cdot \mathbf{o}_{\text{chunk}}$
32:     $\mathbf{m}_{\text{max}} \leftarrow \mathbf{m}_{\text{max\_new}}$
33: **end for**

34:                  $\triangleright$ Store partial results for this program's (batch, head_block, M_block)
35: Store $\mathbf{o}_{\text{accum}}$ into $\mathbf{O}_{\text{partial}}[b, p_{id_M}, 0, h_{\text{start}} \dots, :]$
36: $\mathbf{LSE}_{\text{val}} \leftarrow \log(\mathbf{s}_{\text{exp\_sum}}) + \mathbf{m}_{\text{max}}$
37: Store $\mathbf{LSE}_{\text{val}}$ into $\mathbf{LSE}_{\text{partial}}[b, p_{id_M}, h_{\text{start}} \dots]$

---

This suggests that FlashTPA's efficiency is well-maintained even as the model's embedding dimension increases.

**Figure 8 (Embedding Dimension 1024):** For a smaller embedding dimension of 1024, similar trends are observed:

- FlashTPA (red line) is highly competitive. MHA (orange line) remains the least performant. MQA (purple line) and GQA (green line) are faster than MHA.
- However, as sequence length increases, both MLA (blue line) and FlashTPA demonstrate superior scalability. FlashTPA generally matches or outperforms MLA, particularly for sequences longer than $2^{15}$. For example, with a batch size of 16, TPA shows a clear speed advantage over MLA for sequence lengths $2^{16}$ and greater.

These results across different embedding dimensions highlight the robustness of FlashTPA's decoding speed advantages, especially for long sequences where it consistently ranks as one of the fastest, if not the fastest, attention mechanisms among those tested.

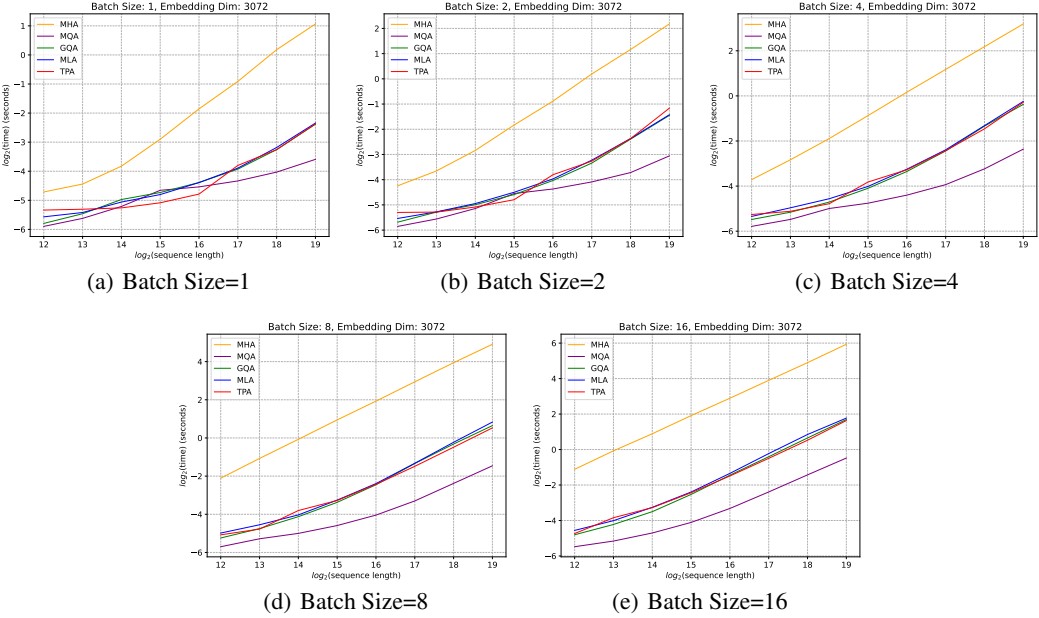

Figure 7: Decoding time comparison of different attention mechanisms with an embedding dimension of 3072 and $d_h = 64$.

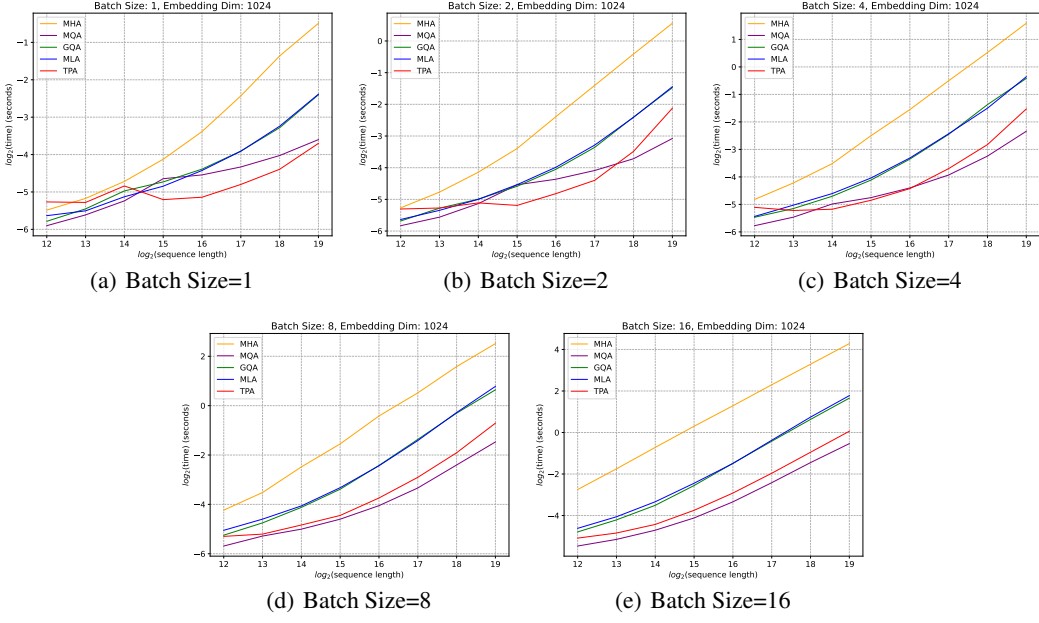

Figure 8: Decoding time comparison of different attention mechanisms with an embedding dimension of 1024 and $d_h = 64$.

## C  Higher-Order Tensor Product Attention

All prior discussions have focused on TPA where the query, key, and value matrices (e.g., $\mathbf{Q}_t \in \mathbb{R}^{h \times d_h}$) are formed as a sum of $R_Q$ components. Each component is an outer product of two context-dependent vectors, one spanning the head dimension ($\mathbb{R}^h$) and the other spanning the feature-per-head dimension ($\mathbb{R}^{d_h}$), as detailed in Section 3.1 (e.g., $\mathbf{Q}_t = \frac{1}{R_Q}\mathbf{A}_Q(\mathbf{x}_t)^\top \mathbf{B}_Q(\mathbf{x}_t)$ implies $\mathbf{Q}_t = \sum_r \mathbf{a}_r \mathbf{b}_r^\top$ where $\mathbf{a}_r$ are columns of $\mathbf{A}_Q^\top$ and $\mathbf{b}_r^\top$ are rows of $\mathbf{B}_Q$). We now generalize this by introducing additional latent factors in the construction of the feature-per-head vectors, leading to what we term *higher-order* TPA. This approach allows for more complex interactions in forming these feature vectors.

For instance, in a third-order factorization, the query tensor $\mathbf{Q}_t \in \mathbb{R}^{h \times d_h}$ for a single token $t$ is constructed as:

$$\mathbf{Q}_t = \frac{1}{R_Q} \sum_{r=1}^{R_Q} \mathbf{a}_r^Q(\mathbf{x}_t) \otimes \text{vec}\big(\mathbf{b}_r^Q(\mathbf{x}_t) \otimes \mathbf{c}_r^Q(\mathbf{x}_t)\big),$$

where $\mathbf{a}_r^Q(\mathbf{x}_t) \in \mathbb{R}^h$. The term $\mathbf{b}_r^Q(\mathbf{x}_t) \in \mathbb{R}^{d_b}$ and the newly introduced factor $\mathbf{c}_r^Q(\mathbf{x}_t) \in \mathbb{R}^{d_c}$ first form a matrix $\mathbf{b}_r^Q(\mathbf{x}_t) \otimes \mathbf{c}_r^Q(\mathbf{x}_t) \in \mathbb{R}^{d_b \times d_c}$ via an outer product (as defined in Section 2). This matrix is then vectorized by $\text{vec}(\cdot)$ into a column vector of dimension $d_h = d_b d_c$. The final query $\mathbf{Q}_t$ is formed by the sum of outer products between $\mathbf{a}_r^Q(\mathbf{x}_t)$ and these resulting $d_h$-dimensional vectors. Analogous expansions apply to $\mathbf{K}_t$ and $\mathbf{V}_t$.

The additional factor $\mathbf{c}_r^Q(\mathbf{x}_t)$ can be viewed as a learnable, context-dependent modulation or gating term for the features generated by $\mathbf{b}_r^Q(\mathbf{x}_t)$.

$$\mathbf{b}_r^Q(\mathbf{x}_t) \in \mathbb{R}^{d_b}, \quad \mathbf{c}_r^Q(\mathbf{x}_t) \in \mathbb{R}^{d_c}, \quad d_h = d_b d_c.$$

This higher-order construction can enhance expressiveness. While introducing $\mathbf{c}_r^Q$ increases the parameter count for the factors, it might allow for the use of smaller base ranks ($R_Q, R_K, R_V$) to achieve comparable representational power, thus offering a different design choice. One could also explore tying or sharing $\mathbf{c}_r^Q$ across queries, keys, and values to manage parameter overhead.

From a memory perspective, during inference, higher-order TPA maintains the benefit of factorized KV caching. Only the constituent factors $\mathbf{a}_K(\mathbf{x}_t), \mathbf{b}_K(\mathbf{x}_t), \mathbf{c}_K(\mathbf{x}_t)$ (and similarly for values) for each past token need to be stored. A trade-off arises between model capacity and the overhead of memory and computation. Higher-order tensor decompositions can provide additional flexibility and potentially increased capacity.

### C.1  RoPE Compatibility in Higher-Order TPA

Rotary positional embeddings (RoPE) remain compatible with higher-order factorizations. In second-order TPA, RoPE applies rotations to the $d_h$-dimensional feature vectors. This compatibility extends to higher-order TPA. Consider the case where RoPE is intended to primarily rotate feature pairs derived from the $\mathbf{b}_r^Q(\mathbf{x}_t)$ components, while the structural influence of $\mathbf{c}_r^Q(\mathbf{x}_t)$ components on the $d_h$-dimensional vector is preserved. More formally, RoPE acts on the $d_h$-dimensional vector $\text{vec}(\mathbf{b}_r^Q \otimes \mathbf{c}_r^Q)$ such that the transformation is equivalent to rotating $\mathbf{b}_r^Q$ to $\widetilde{\mathbf{b}}_r^Q = \mathbf{R}_t \mathbf{b}_r^Q$ (where $\mathbf{R}_t$ is the RoPE rotation matrix for $d_b$ dimensions) and then forming $\text{vec}(\widetilde{\mathbf{b}}_r^Q \otimes \mathbf{c}_r^Q)$. This is achieved by a specific RoPE transformation matrix $\mathbf{T}_t$ acting on the full $d_h$-dimensional vector, as stated in the following theorem.

**Theorem C.1** (RoPE Compatibility in Higher-Order TPA). Consider the higher-order (3-order) Tensor Product Attention (TPA) query factorization

$$\mathbf{Q}_t = \frac{1}{R_Q} \sum_{r=1}^{R_Q} \mathbf{a}_r^Q(\mathbf{x}_t) \otimes \text{vec}\big(\mathbf{b}_r^Q(\mathbf{x}_t) \otimes \mathbf{c}_r^Q(\mathbf{x}_t)\big) \in \mathbb{R}^{h \times d_h},$$

where $\mathbf{a}_r^Q(\mathbf{x}_t) \in \mathbb{R}^h$, $\mathbf{b}_r^Q(\mathbf{x}_t) \in \mathbb{R}^{d_b}$, $\mathbf{c}_r^Q(\mathbf{x}_t) \in \mathbb{R}^{d_c}$, with $d_h = d_b d_c$. Define the RoPE-transformed query as $\widetilde{\mathbf{Q}}_t = \mathrm{RoPE}_t(\mathbf{Q}_t) = \mathbf{Q}_t \mathbf{T}_t$, where

$$
\mathbf{T}_t = \mathbf{I}_{d_c} \otimes (\mathbf{R}_t)^\top = \begin{pmatrix} (\mathbf{R}_t)^\top & \cdots & \mathbf{0} & \mathbf{0} \\ \mathbf{0} & (\mathbf{R}_t)^\top & \cdots & \mathbf{0} \\ \vdots & \vdots & \ddots & \vdots \\ \mathbf{0} & \mathbf{0} & \cdots & (\mathbf{R}_t)^\top \end{pmatrix} \in \mathbb{R}^{d_h \times d_h},
$$

$\mathbf{I}_{d_c}$ is the identity matrix of size $d_c \times d_c$, and $\mathbf{R}_t \in \mathbb{R}^{d_b \times d_b}$ ($d_b \in \mathbb{Z}_+$ is even) is the standard RoPE block-diagonal matrix composed of $2 \times 2$ rotation matrices:

$$
\mathbf{R}_t = \begin{pmatrix} \cos(t\theta_1) & -\sin(t\theta_1) & & & & \\ \sin(t\theta_1) & \cos(t\theta_1) & & & & \\ & & \cos(t\theta_2) & -\sin(t\theta_2) & & \\ & & \sin(t\theta_2) & \cos(t\theta_2) & & \\ & & & & \ddots & \\ & & & & & \cos(t\theta_{d_b/2}) & -\sin(t\theta_{d_b/2}) \\ & & & & & \sin(t\theta_{d_b/2}) & \cos(t\theta_{d_b/2}) \end{pmatrix},
$$

for $t \in \{1, \dots, T\}$ and $j \in \{1, \dots, d_b/2\}$. The transformation $\mathbf{T}_t = \mathbf{I}_{d_c} \otimes (\mathbf{R}_t)^\top$ operates on the $d_h$-dimensional vectorized features by post-multiplication. This structure of $\mathbf{T}_t$ ensures that the rotation effectively applied to the $\mathbf{b}_r^Q(\mathbf{x}_t)$ component (which is a column vector) corresponds to a pre-multiplication by $\mathbf{R}_t$, as detailed in the proof (Appendix D.2). This preserves the structure induced by $\mathbf{c}_r^Q(\mathbf{x}_t)$ while rotating $\mathbf{b}_r^Q(\mathbf{x}_t)$.

Under these conditions, the RoPE-transformed query $\mathrm{RoPE}_t(\mathbf{Q}_t)$ admits a higher-order TPA factorization of the same rank $R_Q$:

$$
\frac{1}{R_Q} \sum_{r=1}^{R_Q} \mathbf{a}_r^Q(\mathbf{x}_t) \otimes \mathrm{vec}\left( \widetilde{\mathbf{b}}_r^Q(\mathbf{x}_t) \otimes \mathbf{c}_r^Q(\mathbf{x}_t) \right) = \mathrm{RoPE}_t(\mathbf{Q}_t), \tag{C.1}
$$

where $\widetilde{\mathbf{b}}_r^Q(\mathbf{x}_t) = \mathbf{R}_t \mathbf{b}_r^Q(\mathbf{x}_t)$.

Please see Appendix D.2 for the proof. For fourth-order or higher, this result still holds.

To assess its empirical performance, we implemented third-order TPA. Table 4 lists the evaluation results for a small model. These results provide an initial indication of its viability. A comprehensive comparison with second-order TPA variants of similar parameter counts or ranks would be necessary to fully evaluate the trade-offs.

Table 4: The evaluation results of small models with third-order TPA pre-trained using FineWeb-Edu 100B dataset with lm-evaluation-harness. Abbreviations: HellaSw. = HellaSwag, W.G. = WinoGrande.

| Few-shot | ARC-E | ARC-C | BoolQ | HellaSw. | OBQA | PIQA | W.G. | MMLU | SciQ | Avg. |
|---|---|---|---|---|---|---|---|---|---|---|
| **0-shot** | 49.24 | 24.91 | 57.06 | 34.01 | 31.80 | 63.33 | 50.59 | 23.23 | 66.9 | 44.56 |
| **2-shot** | 53.37 | 25.34 | 48.78 | 34.00 | 29.20 | 62.79 | 52.33 | 26.41 | 75.3 | 45.28 |

# D Proofs of Theorems

## D.1 Proof of Theorem 3.1

*Proof.* Because RoPE is a linear orthogonal transform, we can write

$$
\widetilde{\mathbf{Q}}_t = \mathbf{Q}_t \, \mathbf{T}_t = \frac{1}{R_Q} \left( \mathbf{A}_Q(\mathbf{x}_t)^\top \mathbf{B}_Q(\mathbf{x}_t) \right) \mathbf{T}_t = \frac{1}{R_Q} \mathbf{A}_Q(\mathbf{x}_t)^\top \left( \mathbf{B}_Q(\mathbf{x}_t) \, \mathbf{T}_t \right),
$$

where $\mathbf{T}_t$ is the block-diagonal matrix encoding RoPE. This allows us to define

$$
\widetilde{\mathbf{B}}_Q(\mathbf{x}_t) = \mathbf{B}_Q(\mathbf{x}_t) \, \mathbf{T}_t,
$$

thereby obtaining

$$\mathrm{RoPE}_t(\mathbf{Q}_t) = \frac{1}{R_Q} \mathbf{A}_Q(\mathbf{x}_t)^\top \widetilde{\mathbf{B}}_Q(\mathbf{x}_t).$$

Similarly, for the key tensor $\mathbf{K}_s$, we have

$$\widetilde{\mathbf{K}}_s = \mathbf{K}_s\, \mathbf{T}_s = \frac{1}{R_K}\big(\mathbf{A}_K(\mathbf{x}_s)^\top\, \mathbf{B}_K(\mathbf{x}_s)\big)\, \mathbf{T}_s = \frac{1}{R_K}\mathbf{A}_K(\mathbf{x}_s)^\top\big(\mathbf{B}_K(\mathbf{x}_s)\, \mathbf{T}_s\big),$$

which defines

$$\widetilde{\mathbf{B}}_K(\mathbf{x}_s) = \mathbf{B}_K(\mathbf{x}_s)\, \mathbf{T}_s,$$

and thus

$$\mathrm{RoPE}_s(\mathbf{K}_s) = \frac{1}{R_K}\mathbf{A}_K(\mathbf{x}_s)^\top \widetilde{\mathbf{B}}_K(\mathbf{x}_s).$$

Now, consider the product of the rotated queries and keys:

$$\widetilde{\mathbf{Q}}_t\, \widetilde{\mathbf{K}}_s^\top = \frac{1}{R_Q R_K}\left(\mathbf{A}_Q(\mathbf{x}_t)^\top \widetilde{\mathbf{B}}_Q(\mathbf{x}_t)\right)\left(\mathbf{A}_K(\mathbf{x}_s)^\top \widetilde{\mathbf{B}}_K(\mathbf{x}_s)\right)^\top$$

$$= \frac{1}{R_Q R_K}\mathbf{A}_Q(\mathbf{x}_t)^\top \widetilde{\mathbf{B}}_Q(\mathbf{x}_t)\widetilde{\mathbf{B}}_K(\mathbf{x}_s)^\top \mathbf{A}_K(\mathbf{x}_s),$$

Since $\mathbf{T}_t$ and $\mathbf{T}_s$ encode positional rotations, the product $\mathbf{T}_t\mathbf{T}_s^\top$ corresponds to a relative rotation $\mathbf{T}_{t-s}$. Therefore, we can express the above as

$$\widetilde{\mathbf{Q}}_t\, \widetilde{\mathbf{K}}_s^\top = \frac{1}{R_Q R_K}\mathbf{A}_Q(\mathbf{x}_t)^\top\left(\mathbf{B}_Q(\mathbf{x}_t)\mathbf{T}_t\mathbf{T}_s^\top\mathbf{B}_K(\mathbf{x}_s)^\top\right)\mathbf{A}_K(\mathbf{x}_s)$$

$$= \frac{1}{R_Q R_K}\mathbf{A}_Q(\mathbf{x}_t)^\top\left(\mathbf{B}_Q(\mathbf{x}_t)\mathbf{T}_{t-s}\mathbf{B}_K(\mathbf{x}_s)^\top\right)\mathbf{A}_K(\mathbf{x}_s)$$

$$= \frac{1}{R_Q R_K}\mathbf{A}_Q(\mathbf{x}_t)^\top\left(\mathbf{B}_Q(\mathbf{x}_t)\mathbf{T}_{t-s}\right)\left(\mathbf{B}_K(\mathbf{x}_s)^\top\mathbf{A}_K(\mathbf{x}_s)\right)$$

$$= \left(\frac{1}{R_Q}\mathbf{A}_Q(\mathbf{x}_t)^\top\mathbf{B}_Q(\mathbf{x}_t)\mathbf{T}_{t-s}\right)\left(\frac{1}{R_K}\mathbf{A}_K(\mathbf{x}_s)^\top\mathbf{B}_K(\mathbf{x}_s)\right)^\top,$$

This shows that

$$\mathrm{RoPE}_{t-s}(\mathbf{Q}_t)\mathbf{K}_s^\top = \widetilde{\mathbf{Q}}_t\, \widetilde{\mathbf{K}}_s^\top,$$

Focusing on individual heads $i$, the above matrix equality implies:

$$(\mathbf{q}_{t,i}\mathbf{T}_{t-s})\, \mathbf{k}_{s,i}^\top = (\mathbf{q}_{t,i}\mathbf{T}_t)\, (\mathbf{k}_{s,i}\mathbf{T}_s)^\top,$$

where

$$\widetilde{\mathbf{q}}_{t,i} = \mathrm{RoPE}_t(\mathbf{q}_{t,i}) = \mathbf{q}_{t,i}\mathbf{T}_t \in \mathbb{R}^{1\times d_h}, \quad \widetilde{\mathbf{k}}_{s,i} = \mathrm{RoPE}_s(\mathbf{k}_{s,i}) = \mathbf{k}_{s,i}\mathbf{T}_s \in \mathbb{R}^{1\times d_h}.$$

This equality confirms that the relative positional encoding between queries and keys is preserved under TPA's factorization and RoPE's rotation. Thus, TPA maintains compatibility with RoPE. This completes the proof of Theorem 3.1. $\qquad\square$

### D.2 Proof of Theorem C.1

Theorem C.1 addresses the compatibility of RoPE with higher-order (specifically, 3rd-order) Tensor Product Attention. The theorem considers the query factorization:

$$\mathbf{Q}_t = \frac{1}{R_Q}\sum_{r=1}^{R_Q}\mathbf{a}_r^Q(\mathbf{x}_t) \otimes \mathrm{vec}\big(\mathbf{b}_r^Q(\mathbf{x}_t) \otimes \mathbf{c}_r^Q(\mathbf{x}_t)\big) \in \mathbb{R}^{h\times d_h},$$

where $\mathbf{a}_r^Q(\mathbf{x}_t) \in \mathbb{R}^h$ (column vector), $\mathbf{b}_r^Q(\mathbf{x}_t) \in \mathbb{R}^{d_b}$ (column vector), $\mathbf{c}_r^Q(\mathbf{x}_t) \in \mathbb{R}^{d_c}$ (column vector), and $d_h = d_b d_c$. The term $\mathbf{b}_r^Q(\mathbf{x}_t) \otimes \mathbf{c}_r^Q(\mathbf{x}_t)$ is interpreted as the matrix $\mathbf{M}_r =$

$\mathbf{b}_r^Q(\mathbf{x}_t)(\mathbf{c}_r^Q(\mathbf{x}_t))^\top \in \mathbb{R}^{d_b \times d_c}$. The notation $\mathbf{a} \otimes \mathbf{v}$ for $\mathbf{a} \in \mathbb{R}^h$ and $\mathbf{v} \in \mathbb{R}^{d_h}$ (column vectors) implies the outer product $\mathbf{a}\mathbf{v}^\top$. Thus, $\mathbf{Q}_t = \frac{1}{R_Q} \sum_{r=1}^{R_Q} \mathbf{a}_r^Q(\mathbf{x}_t)(\text{vec}(\mathbf{M}_r))^\top$.

The RoPE-transformed query is defined as $\widetilde{\mathbf{Q}}_t = \text{RoPE}_t(\mathbf{Q}_t) = \mathbf{Q}_t \mathbf{T}_t$. Crucially, for the theorem's conclusion to hold as intended (i.e., that the $\mathbf{b}_r^Q$ component is transformed by pre-multiplication with the standard RoPE matrix $\mathbf{R}_t$), the global transformation matrix $\mathbf{T}_t \in \mathbb{R}^{d_h \times d_h}$ (that post-multiplies $\mathbf{Q}_t$) is given by:

$$\mathbf{T}_t = \mathbf{I}_{d_c} \otimes (\mathbf{R}_t)^\top,$$

where $\mathbf{I}_{d_c}$ is the $d_c \times d_c$ identity matrix, and $\mathbf{R}_t \in \mathbb{R}^{d_b \times d_b}$ is the standard RoPE block-diagonal matrix that pre-multiplies $d_b$-dimensional column vectors (as defined explicitly in the theorem statement in Section C).

The theorem claims that, under these conditions, $\widetilde{\mathbf{Q}}_t$ admits a higher-order TPA factorization:

$$\widetilde{\mathbf{Q}}_t = \frac{1}{R_Q} \sum_{r=1}^{R_Q} \mathbf{a}_r^Q(\mathbf{x}_t) \otimes \text{vec}\Big(\widetilde{\mathbf{b}}_r^Q(\mathbf{x}_t) \otimes \mathbf{c}_r^Q(\mathbf{x}_t)\Big),$$

where $\widetilde{\mathbf{b}}_r^Q(\mathbf{x}_t) = \mathbf{R}_t \mathbf{b}_r^Q(\mathbf{x}_t)$.

*Proof.* Let $\mathbf{a}_r^Q \equiv \mathbf{a}_r^Q(\mathbf{x}_t)$, $\mathbf{b}_r^Q \equiv \mathbf{b}_r^Q(\mathbf{x}_t)$, and $\mathbf{c}_r^Q \equiv \mathbf{c}_r^Q(\mathbf{x}_t)$ for brevity. Let $\mathbf{M}_r = \mathbf{b}_r^Q(\mathbf{c}_r^Q)^\top \in \mathbb{R}^{d_b \times d_c}$. Let $\mathbf{v}_r = \text{vec}(\mathbf{M}_r) \in \mathbb{R}^{d_h}$ be the column vector obtained by stacking the columns of $\mathbf{M}_r$. The query tensor is $\mathbf{Q}_t = \frac{1}{R_Q} \sum_{r=1}^{R_Q} \mathbf{a}_r^Q(\mathbf{v}_r)^\top$.

The RoPE transformation is $\widetilde{\mathbf{Q}}_t = \mathbf{Q}_t \mathbf{T}_t$. Substituting the factorization and the revised definition of $\mathbf{T}_t$:

$$\widetilde{\mathbf{Q}}_t = \left( \frac{1}{R_Q} \sum_{r=1}^{R_Q} \mathbf{a}_r^Q(\mathbf{v}_r)^\top \right) (\mathbf{I}_{d_c} \otimes (\mathbf{R}_t)^\top)$$

$$= \frac{1}{R_Q} \sum_{r=1}^{R_Q} \mathbf{a}_r^Q \left( (\mathbf{v}_r)^\top (\mathbf{I}_{d_c} \otimes (\mathbf{R}_t)^\top) \right).$$

Let's analyze the transformed vector part for the $r$-th component: $(\mathbf{v}_r)^\top (\mathbf{I}_{d_c} \otimes (\mathbf{R}_t)^\top)$. This row vector is the transpose of $((\mathbf{I}_{d_c} \otimes (\mathbf{R}_t)^\top)^\top \mathbf{v}_r)$. Let's compute the pre-multiplying matrix:

$$((\mathbf{I}_{d_c} \otimes (\mathbf{R}_t)^\top)^\top = (\mathbf{I}_{d_c})^\top \otimes ((\mathbf{R}_t)^\top)^\top = \mathbf{I}_{d_c} \otimes \mathbf{R}_t.$$

So, the column vector transformation is $(\mathbf{I}_{d_c} \otimes \mathbf{R}_t)\mathbf{v}_r$. Substitute $\mathbf{v}_r = \text{vec}(\mathbf{M}_r) = \text{vec}(\mathbf{b}_r^Q(\mathbf{c}_r^Q)^\top)$:

$$(\mathbf{I}_{d_c} \otimes \mathbf{R}_t) \text{vec}(\mathbf{b}_r^Q(\mathbf{c}_r^Q)^\top).$$

We use the Kronecker product identity: $(\mathbf{B_0}^\top \otimes \mathbf{A_0}) \text{vec}(\mathbf{X_0}) = \text{vec}(\mathbf{A_0}\mathbf{X_0}\mathbf{B_0})$. To match our expression $(\mathbf{I}_{d_c} \otimes \mathbf{R}_t) \text{vec}(\mathbf{M}_r)$, we identify: $\mathbf{A_0} = \mathbf{R}_t$, $\mathbf{B_0}^\top = \mathbf{I}_{d_c} \implies \mathbf{B_0} = \mathbf{I}_{d_c}$, $\mathbf{X_0} = \mathbf{M}_r = \mathbf{b}_r^Q(\mathbf{c}_r^Q)^\top$. Applying the identity, we get:

$$\text{vec}\left( \mathbf{R}_t(\mathbf{b}_r^Q(\mathbf{c}_r^Q)^\top)\mathbf{I}_{d_c} \right) = \text{vec}\left( (\mathbf{R}_t \mathbf{b}_r^Q)(\mathbf{c}_r^Q)^\top \right).$$

Let $\widetilde{\mathbf{b}}_r^Q = \mathbf{R}_t \mathbf{b}_r^Q$. This is precisely the transformation for the $\mathbf{b}_r^Q$ component as claimed in the theorem. So the transformed column vector is $\text{vec}(\widetilde{\mathbf{b}}_r^Q(\mathbf{c}_r^Q)^\top)$. The corresponding row vector in the sum for $\widetilde{\mathbf{Q}}_t$ is therefore $(\text{vec}(\widetilde{\mathbf{b}}_r^Q(\mathbf{c}_r^Q)^\top))^\top$.

Substituting this back into the expression for $\widetilde{\mathbf{Q}}_t$:

$$\widetilde{\mathbf{Q}}_t = \frac{1}{R_Q} \sum_{r=1}^{R_Q} \mathbf{a}_r^Q(\text{vec}(\widetilde{\mathbf{b}}_r^Q(\mathbf{c}_r^Q)^\top))^\top.$$

This is equivalent to the theorem's claimed factorization, using the definition $\mathbf{a} \otimes \mathbf{col\_vec} = \mathbf{a}(\mathbf{col\_vec})^\top$:

$$\widetilde{\mathbf{Q}}_t = \frac{1}{R_Q} \sum_{r=1}^{R_Q} \mathbf{a}_r^Q \otimes \text{vec}(\widetilde{\mathbf{b}}_r^Q \otimes \mathbf{c}_r^Q),$$

where $\widetilde{\mathbf{b}}_r^Q = \mathbf{R}_t \mathbf{b}_r^Q$. This completes the proof, showing that RoPE can be consistently applied to higher-order TPA representations if the global RoPE transformation matrix $\mathbf{T}_t$ (that post-multiplies $\mathbf{Q}_t$) is appropriately defined as $\mathbf{I}_{d_c} \otimes (\mathbf{R}_t)^\top$, ensuring that the standard RoPE matrix $\mathbf{R}_t$ effectively pre-multiplies the $\mathbf{b}_r^Q$ component. $\qquad\square$

## E  More Related Work

**Transformers and Attention.**  As a sequence-to-sequence architecture, Transformer [60] introduced Multi-Head Attention (MHA), enabling more effective capture of long-range dependencies. Subsequent work has explored a variety of attention mechanisms aimed at improving scalability and efficiency, including sparse patterns [10, 49, 16, 30, 27, 31], kernel-based projections [11], and linearized transformers [59, 25, 44, 69, 54, 67]. To decrease memory usage and circumvent the limitation of memory bandwidth in training, [46] proposed Multi-Query Attention (MQA) where multiple query heads share the same key head and value head. To tackle the issue of quality degradation and instability in training, Grouped-Query Attention (GQA) [2] divides queries into several groups, and each group of queries shares a single key head and value head. Recently, DeepSeek-V2 [32] applied multihead latent attention (MLA) to achieve better performance than MHA while reducing KV cache in inference time by sharing the same low-rank representation of key and value. Concurrently, [21] proposed Multi-matrix Factorization Attention (MFA), which can be simply seen as MQA with low-rank factorized Q. Compared to the approaches above, TPA applied contextual tensor decompositions to represent queries, keys, and values activations compactly, achieving better reduction on the size of KV cache with improved performance.

**KV Cache Optimization.**  During the auto-regressive inference of Transformers, key and value (KV) tensors from previous tokens are cached to avoid recomputation, a technique first proposed by [40]. This Key-Value (KV) cache, while crucial for efficiency, consumes significant memory and can introduce latency bottlenecks due to memory bandwidth limitations [1]. Consequently, various studies have explored methods to mitigate these issues. These include KV cache eviction strategies that discard less significant tokens [70, 62, 8, 1], dynamic sparse attention mechanisms focusing on selected keys and values [42, 55, 50], offloading the KV cache to CPU memory [17, 26, 53], and quantizing the KV cache [61, 34, 19]. In contrast to these approaches, TPA focuses on reducing the intrinsic size of the KV cache by employing tensor-decomposed key and value representations.

**Low-Rank Factorizations.**  Low-rank approximations are widely used to compress model parameters and reduce computational complexity. Notable examples include LoRA [20], which factorizes weight updates during fine-tuning, and its derivatives tailored for various training scenarios such as efficient pretraining (ReLoRA [28], MoRA [22]), long-context training (LongLoRA [9], SinkLoRA [66]), and continual training (InfLoRA [29], GS-LoRA [71], I-LoRA [41]). These methods generally produce static low-rank expansions that are independent of the input context. Theoretical justifications for the expressiveness of low-rank approximations have been provided by [38, 65]. Initialization strategies for these factorization matrices have also been explored: OLoRA [7] utilizes QR-decomposition of pretrained weights for improved language model performance, while LoLDU [48] employs LDU-decomposition to accelerate LoRA training. Furthermore, AdaLoRA [68] uses Singular Value Decomposition (SVD) on pretrained weights and introduces parameter importance scores to dynamically adjust ranks. TPA, in contrast, constructs Q, K, and V tensors using contextually-aware factorizations, allowing for dynamic adaptation based on the input.

## F  More on Attention Mechanisms

### F.1  Multi-Query Attention (MQA)

Multi-Query Attention (MQA) [46] significantly reduces memory usage, particularly for the KV cache, by sharing a single key and value projection across all attention heads, while each head maintains a unique query projection. Given a sequence of input embeddings $\mathbf{X} \in \mathbb{R}^{T \times d_{\text{model}}}$, the query, shared key, and shared value tensors are computed as:

$$\mathbf{Q}_i = \mathbf{X} \boldsymbol{W}_i^Q, \quad \mathbf{K}_{\text{shared}} = \mathbf{X} \boldsymbol{W}_{\text{shared}}^K, \quad \mathbf{V}_{\text{shared}} = \mathbf{X} \boldsymbol{W}_{\text{shared}}^V.$$

Thus, each head $i$ uses a distinct query projection $\mathbf{Q}_i \in \mathbb{R}^{T \times d_h}$ but shares the common key $\mathbf{K}_{\text{shared}} \in \mathbb{R}^{T \times d_h}$ and value $\mathbf{V}_{\text{shared}} \in \mathbb{R}^{T \times d_h}$ tensors. The weight matrices are:

$$\boldsymbol{W}_i^Q \in \mathbb{R}^{d_{\text{model}} \times d_h}, \quad \boldsymbol{W}_{\text{shared}}^K, \boldsymbol{W}_{\text{shared}}^V \in \mathbb{R}^{d_{\text{model}} \times d_h}.$$

The resulting MQA operation is:

$$\text{MQA}(\mathbf{X}) = \text{Concat}\Big(\mathbf{head}_1, \ldots, \mathbf{head}_h\Big)\boldsymbol{W}^O,$$

where

$$\mathbf{head}_i = \text{Attention}\big(\mathbf{Q}_i, \mathbf{K}_{\text{shared}}, \mathbf{V}_{\text{shared}}\big).$$

By sharing key and value projections, MQA substantially reduces memory demands, especially for the KV cache during autoregressive inference. However, this comes at the cost of reduced model expressivity, as all heads must utilize the same key and value representations.

### F.2 Grouped Query Attention (GQA)

Grouped Query Attention (GQA) [2] generalizes Multi-Head Attention (MHA) and MQA by dividing the total $h$ attention heads into $G$ groups. Within each group, heads share a common key and value projection, while each head maintains its own unique query projection. Formally, let $g(i)$ denote the group index for head $i \in \{1, \ldots, h\}$, where $g(i) \in \{1, \ldots, G\}$. The projections are:

$$\mathbf{K}_{g(i)} = \mathbf{X}\,\boldsymbol{W}_{g(i)}^K, \quad \mathbf{V}_{g(i)} = \mathbf{X}\,\boldsymbol{W}_{g(i)}^V, \quad \mathbf{Q}_i = \mathbf{X}\,\boldsymbol{W}_i^Q,$$

and

$$\text{head}_i = \text{Attention}\Big(\mathbf{Q}_i, \mathbf{K}_{g(i)}, \mathbf{V}_{g(i)}\Big).$$

Here, $\boldsymbol{W}_g^K$ and $\boldsymbol{W}_g^V$ are the shared weight matrices for group $g$, each in $\mathbb{R}^{d_{\text{model}} \times d_h}$, and $\boldsymbol{W}_i^Q \in \mathbb{R}^{d_{\text{model}} \times d_h}$ is the query weight matrix for head $i$. The complete output is again a concatenation of all heads:

$$\text{GQA}(\mathbf{X}) = \text{Concat}\Big(\text{head}_1, \ldots, \text{head}_h\Big)\boldsymbol{W}^O.$$

By varying $G$ from 1 (equivalent to MQA) to $h$ (equivalent to MHA), GQA offers a trade-off between memory efficiency and model capacity.

### F.3 Multi-head Latent Attention (MLA)

Multi-head Latent Attention (MLA), as used in DeepSeek-V2 [32] and DeepSeek-V3 [33], introduces low-rank compression for keys and values to reduce KV caching costs during inference.

$$\mathbf{C}^{KV} = \mathbf{X}\boldsymbol{W}^{DKV},$$
$$\text{Concat}\big(\mathbf{K}_1^C, \mathbf{K}_2^C, \ldots, \mathbf{K}_h^C\big) = \mathbf{K}^C = \mathbf{C}^{KV}\boldsymbol{W}^{UK},$$
$$\mathbf{K}^R = \text{RoPE}\big(\mathbf{X}\boldsymbol{W}^{KR}\big),$$
$$\mathbf{K}_i = \text{Concat}\big(\mathbf{K}_i^C, \mathbf{K}^R\big),$$
$$\text{Concat}\big(\mathbf{V}_1^C, \mathbf{V}_2^C, \ldots, \mathbf{V}_h^C\big) = \mathbf{V}^C = \mathbf{C}^{KV}\boldsymbol{W}^{UV},$$

Here, $\boldsymbol{W}^{DKV} \in \mathbb{R}^{d_{\text{model}} \times d_c}$ projects to a compressed dimension $d_c$, $\boldsymbol{W}^{UK} \in \mathbb{R}^{d_c \times (d_h h)}$ up-projects the compressed keys, $\boldsymbol{W}^{KR} \in \mathbb{R}^{d_{\text{model}} \times d_h^R}$ projects to a residual key component for RoPE, and $\boldsymbol{W}^{UV} \in \mathbb{R}^{d_c \times (d_h h)}$ up-projects the compressed values. $\mathbf{C}^{KV} \in \mathbb{R}^{T \times d_c}$ is the shared compressed KV latent (where $d_c \ll d_h h$). The RoPE transformation is applied to a separate key embedding $\mathbf{K}^R \in \mathbb{R}^{T \times d_h^R}$. Thus, only $\mathbf{C}^{KV}$ and $\mathbf{K}^R$ are cached, reducing KV memory usage while largely preserving performance compared to standard MHA [60].

MLA also compresses the queries, lowering their training-time memory footprint:

$$\mathbf{C}^Q = \mathbf{X}\boldsymbol{W}^{DQ},$$
$$\text{Concat}\big(\mathbf{Q}_1^C, \mathbf{Q}_2^C, \ldots, \mathbf{Q}_h^C\big) = \mathbf{Q}^C = \mathbf{C}^Q\boldsymbol{W}^{UQ},$$
$$\text{Concat}\big(\mathbf{Q}_1^R, \mathbf{Q}_2^R, \ldots, \mathbf{Q}_h^R\big) = \mathbf{Q}^R = \text{RoPE}\big(\mathbf{C}^Q\boldsymbol{W}^{QR}\big),$$
$$\mathbf{Q} = \text{Concat}\big(\mathbf{Q}^C, \mathbf{Q}^R\big).$$

The weight matrices are $\boldsymbol{W}^{DQ} \in \mathbb{R}^{d_{\text{model}} \times d'_c}$, $\boldsymbol{W}^{UQ} \in \mathbb{R}^{d'_c \times (d_h h)}$, and $\boldsymbol{W}^{QR} \in \mathbb{R}^{d'_c \times (d_h^R h)}$. Here, $\mathbf{C}^Q \in \mathbb{R}^{T \times d'_c}$ (where $d'_c \ll d_h h$) is the compressed query latent. The final query $\mathbf{Q}_i$ for each head, formed by concatenating $\mathbf{Q}_i^C$ and $\mathbf{Q}_i^R$, has a dimension of $d_h + d_h^R$.

Given compressed queries, keys, and values, the final attention output for the $t$-th token is:

$$\mathbf{O}_i = \text{Softmax}\left(\frac{\mathbf{Q}_i \mathbf{K}_i^\top}{\sqrt{d_h + d_h^R}}\right) \mathbf{V}_i^C,$$

$$\mathbf{U} = \text{Concat}(\mathbf{O}_1, \mathbf{O}_2, \ldots, \mathbf{O}_h) \boldsymbol{W}^O,$$

where $\mathbf{V}_i$ is typically $\mathbf{V}_i^C$ as no residual value component is explicitly defined, and $\boldsymbol{W}^O \in \mathbb{R}^{(d_h h) \times d_{\text{model}}}$ is the output projection.

During inference, $\mathbf{C}^{KV}$ and $\mathbf{K}^R$ are cached to accelerate decoding. In detail, if RoPE were ignored for the compressed components, the inner product $\mathbf{q}_{t,i}^\top \mathbf{k}_{s,i}$ (where $\mathbf{q}_{t,i}, \mathbf{k}_{s,i} \in \mathbb{R}^{d_h}$) of the $i$-th head between $t$-th token query and $s$-th token key could be calculated using the current hidden state $\mathbf{x}_t \in \mathbb{R}^{d_{\text{model}}}$ and the cached latent state $\mathbf{c}_s^{KV} \in \mathbb{R}^{d_c}$ for the $s$-th token:

$$\mathbf{q}_{t,i}^\top \mathbf{k}_{s,i} = [(\boldsymbol{W}_i^{UQ})^\top (\boldsymbol{W}_i^{DQ})^\top \mathbf{x}_t]^\top [(\boldsymbol{W}_i^{UK})^\top \mathbf{c}_s^{KV}] \tag{F.1}$$

$$= \mathbf{x}_t^\top [\boldsymbol{W}_i^{DQ} \boldsymbol{W}_i^{UQ} (\boldsymbol{W}_i^{UK})^\top] \mathbf{c}_s^{KV}, \tag{F.2}$$

where $\boldsymbol{W}_i^{(\cdot)}$ denotes the $i$-th head's portion of the respective weight matrix. The term $[\boldsymbol{W}_i^{DQ} \boldsymbol{W}_i^{UQ} (\boldsymbol{W}_i^{UK})^\top]$ could be pre-computed for faster decoding. However, as noted by [51], this pre-computation strategy is not directly compatible with RoPE if RoPE were applied to these compressed representations. RoPE applies a rotation matrix $\mathbf{T}_t \in \mathbb{R}^{d_h \times d_h}$ based on position $t$ (see Section F.5), satisfying $\mathbf{T}_t \mathbf{T}_s^\top = \mathbf{T}_{t-s}$ (Equation F.4). If RoPE were applied to the up-projected $Q^C$ and $K^C$:

$$\mathbf{q}_{t,i}^\top \mathbf{k}_{s,i} = [\mathbf{T}_t^\top (\boldsymbol{W}_i^{UQ})^\top (\boldsymbol{W}_i^{DQ})^\top \mathbf{x}_t]^\top [\mathbf{T}_s^\top (\boldsymbol{W}_i^{UK})^\top \mathbf{c}_s^{KV}]$$
$$= \mathbf{x}_t^\top [\boldsymbol{W}_i^{DQ} \boldsymbol{W}_i^{UQ} \mathbf{T}_{t-s} (\boldsymbol{W}_i^{UK})^\top] \mathbf{c}_s^{KV}. \tag{F.3}$$

Unlike Equation (F.2), acceleration by pre-computing the term $[\boldsymbol{W}_i^{DQ} \boldsymbol{W}_i^{UQ} \mathbf{T}_{t-s} (\boldsymbol{W}_i^{UK})^\top]$ is not possible because it depends on the relative position $(t - s)$ and thus varies for different $(t, s)$ pairs. To maintain RoPE compatibility while benefiting from compression, MLA introduces an additional, smaller key component $\mathbf{K}^R$ (and similarly $\mathbf{Q}^R$) to which RoPE is applied, while the main compressed components $\mathbf{K}^C$ and $\mathbf{V}^C$ (derived from $\mathbf{C}^{KV}$) remain RoPE-free. As we will demonstrate in Section 3.2 of the main paper, TPA offers a different approach to integrate RoPE efficiently with factorized attention through its tensor product formulation.

### F.4  Multi-matrix Factorization Attention (MFA)

[21] proposed Multi-matrix Factorization Attention (MFA), which can be conceptualized as a variation of MQA where the shared key and value projections have a dimension $d_c$, and the query projection for each head is low-rank factorized:

$$\mathbf{Q}_i = \mathbf{X} \boldsymbol{W}^{DQ} \boldsymbol{W}_i^{UQ}, \quad \mathbf{K}_{\text{shared}} = \mathbf{X} \boldsymbol{W}_{\text{shared}}^K, \quad \mathbf{V}_{\text{shared}} = \mathbf{X} \boldsymbol{W}_{\text{shared}}^V,$$

where

$$\boldsymbol{W}^{DQ} \in \mathbb{R}^{d_{\text{model}} \times d_c}, \quad \boldsymbol{W}_i^{UQ} \in \mathbb{R}^{d_c \times d_c}, \quad \boldsymbol{W}_{\text{shared}}^K, \boldsymbol{W}_{\text{shared}}^V \in \mathbb{R}^{d_{\text{model}} \times d_c}.$$

### F.5  Rotary Position Embedding (RoPE)

Many recent LLMs use rotary position embedding (RoPE; 52) to encode positional information in the query/key vectors. Specifically, for a vector at position $t$, RoPE applies a rotation matrix $\mathbf{T}_t \in \mathbb{R}^{d \times d}$ (where $d$ is the dimension of the query/key vectors, typically $d_h$ per head). $\mathbf{T}_t$ is a block-diagonal matrix composed of $d/2$ rotation blocks of the form $\begin{pmatrix} \cos(t\theta_j) & -\sin(t\theta_j) \\ \sin(t\theta_j) & \cos(t\theta_j) \end{pmatrix}$ for $j \in \{1, \ldots, d/2\}$.

The frequencies $\{\theta_j\}$ are typically defined as $\theta_j = \text{base}^{-2j/d}$, with a common base like 10000. If $\mathbf{q}_t \in \mathbb{R}^d$ is a query (or key) row vector for a specific head at position $t$, RoPE is applied as:

$$\text{RoPE}(\mathbf{q}_t) \triangleq \mathbf{q}_t \mathbf{T}_t.$$

A key property of RoPE is that the inner product between RoPE-transformed vectors depends only on their relative position. For a query $\mathbf{q}_t$ and key $\mathbf{k}_s$: $(\mathbf{q}_t\mathbf{T}_t)(\mathbf{k}_s\mathbf{T}_s)^\top = \mathbf{q}_t\mathbf{T}_t\mathbf{T}_s^\top\mathbf{k}_s^\top = \mathbf{q}_t\mathbf{T}_{t-s}\mathbf{k}_s^\top$. This relies on the property:

$$\mathbf{T}_t\mathbf{T}_s^\top = \mathbf{T}_{t-s}, \tag{F.4}$$

which embeds relative positional information $(t-s)$ into the attention scores.

# G  More on TPA

**Parameter Initialization for TPA Factors.**  We initialize the weight matrices for TPA factors, such as $\boldsymbol{W}_r^{a^Q}, \boldsymbol{W}_r^{a^K}, \boldsymbol{W}_r^{a^V}, \boldsymbol{W}_r^{b^Q}, \boldsymbol{W}_r^{b^K}$, and $\boldsymbol{W}_r^{b^V}$ (or their combined forms $\boldsymbol{W}^{a^Q}, \boldsymbol{W}^{b^Q}$, etc.), using Xavier initialization [15]. Specifically, each entry of a weight matrix is drawn from a uniform distribution $\mathcal{U}(-bound, bound)$, where $bound = \sqrt{6/(n_{\text{in}} + n_{\text{out}})}$. Here, $n_{\text{in}}$ and $n_{\text{out}}$ are the input and output dimensions of the respective weight matrix. This initialization strategy is chosen to help maintain the variance of activations and gradients as they propagate through the network layers, contributing to stable training.

**TPA with Non-contextual B.** In Section 4.1, we have introduced TPA with non-contextual A, where head-dimension factors $\mathbf{a}_r^Q, \mathbf{a}_r^K, \mathbf{a}_r^V \in \mathbb{R}^h$ are fixed. Conversely, one may fix the token-dimension factors $\mathbf{b}_r^Q, \mathbf{b}_r^K, \mathbf{b}_r^V \in \mathbb{R}^{d_h}$ as learned parameters, while allowing $\mathbf{a}_r^Q(\mathbf{x}_t), \mathbf{a}_r^K(\mathbf{x}_t), \mathbf{a}_r^V(\mathbf{x}_t)$ to adapt to the input token $\mathbf{x}_t$. The key tensor for token $t$, $\mathbf{K}_t \in \mathbb{R}^{h \times d_h}$, would then be constructed as:

$$\mathbf{K}_t = \frac{1}{R_K}\sum_{r=1}^{R_K}\mathbf{a}_r^K(\mathbf{x}_t) \otimes \mathbf{b}_r^K.$$

A similar formulation applies to values. This configuration might be effective if the fundamental token-level features (captured by $\mathbf{b}_r$) are relatively stable, while their combination across heads (captured by $\mathbf{a}_r(\mathbf{x}_t)$) needs to adapt to the context. Performance comparisons for TPA with non-contextual A factors versus non-contextual B factors on small and medium-sized models are presented in Tables 5, 6, 7, and 8.

Table 5: Evaluation results of small models with TPA using non-contextual A or B factors, pre-trained on FineWeb-Edu 100B dataset (0-shot with lm-evaluation-harness). Abbreviations: HellaSw. = HellaSwag, W.G. = WinoGrande.

| Method | ARC-E | ARC-C | BoolQ | HellaSw. | OBQA | PIQA | W.G. | MMLU | SciQ | Avg. |
|---|---|---|---|---|---|---|---|---|---|---|
| **TPA (non-ctx-A)** | 50.17 | 25.60 | 57.95 | 36.13 | 31.40 | 64.80 | 49.57 | 24.88 | 64.80 | 45.03 |
| **TPA (non-ctx-B)** | 47.39 | 26.37 | 54.8 | 32.71 | 30.2 | 63.38 | 50.2 | 23.13 | 64.8 | 43.66 |

Table 6: Evaluation results of small models with TPA using non-contextual A or B factors, pre-trained on FineWeb-Edu 100B dataset (2-shot with lm-evaluation-harness). Abbreviations: HellaSw. = HellaSwag, W.G. = WinoGrande.

| Method | ARC-E | ARC-C | BoolQ | HellaSw. | OBQA | PIQA | W.G. | MMLU | SciQ | Avg. |
|---|---|---|---|---|---|---|---|---|---|---|
| **TPA (non-ctx-A)** | 55.09 | 27.65 | 53.82 | 36.24 | 30.20 | 64.53 | 50.75 | 26.01 | 78.60 | 46.99 |
| **TPA (non-ctx-B)** | 50.8 | 26.96 | 57.65 | 32.4 | 29.4 | 63.22 | 49.57 | 23.96 | 66.4 | 44.48 |

Table 7: Evaluation results of medium models with TPA using non-contextual A or B factors, pre-trained on FineWeb-Edu 100B dataset (0-shot with lm-evaluation-harness). Abbreviations: HellaSw. = HellaSwag, W.G. = WinoGrande.

| Method | ARC-E | ARC-C | BoolQ | HellaSw. | OBQA | PIQA | W.G. | MMLU | SciQ | Avg. |
|---|---|---|---|---|---|---|---|---|---|---|
| **TPA (non-ctx-A)** | 58.96 | 31.48 | 59.76 | 45.07 | 34.80 | 69.21 | 53.59 | 25.42 | 76.40 | 50.52 |
| **TPA (non-ctx-B)** | 55.43 | 29.69 | 58.32 | 40.77 | 34.40 | 66.92 | 51.38 | 25.66 | 71.10 | 48.19 |

**TPA KV Only.** A simpler variant involves using a standard linear projection for queries,

$$\mathbf{Q}_t = \mathbf{W}^Q\mathbf{x}_t \in \mathbb{R}^{h \times d_h},$$

and factorize only the key and value tensors $(\mathbf{K}_t, \mathbf{V}_t)$. This approach, termed TPA-KVonly, maintains the standard query projection mechanism but still achieves significant KV cache reduction through factorized key and value representations.

Table 8: Evaluation results of medium models with TPA using non-contextual A or B factors, pre-trained on FineWeb-Edu 100B dataset (2-shot with lm-evaluation-harness). Abbreviations: HellaSw. = HellaSwag, W.G. = WinoGrande.

| Method | ARC-E | ARC-C | BoolQ | HellaSw. | OBQA | PIQA | W.G. | MMLU | SciQ | Avg. |
|---|---|---|---|---|---|---|---|---|---|---|
| **TPA (non-ctx-A)** | 65.45 | 33.79 | 56.88 | 45.23 | 33.60 | 68.61 | 54.22 | 25.00 | 85.00 | 51.98 |
| **TPA (non-ctx-B)** | 61.20 | 30.20 | 55.93 | 40.45 | 34.40 | 68.23 | 51.78 | 26.11 | 78.10 | 49.60 |

**TPA KV with Shared B.** Further parameter reduction can be achieved by sharing the token-dimension factors $\mathbf{b}_r$ between keys and values:

$$\mathbf{b}_r^K(\mathbf{x}_t) = \mathbf{b}_r^V(\mathbf{x}_t) \quad \text{(if contextual)}, \quad \text{or} \quad \mathbf{b}_r^K = \mathbf{b}_r^V \quad \text{(if non-contextual)}.$$

This sharing reduces both parameter count and the KV cache footprint. Although it constrains $\mathbf{K}_t$ and $\mathbf{V}_t$ to be constructed from the same token-level basis vectors, this variant can still offer strong performance with additional memory savings.

**Nonlinear Head Factors.** Instead of using purely linear transformations to derive the contextual head-dimension factors $\mathbf{a}_r^Q(\mathbf{x}_t), \mathbf{a}_r^K(\mathbf{x}_t), \mathbf{a}_r^V(\mathbf{x}_t)$, one can introduce element-wise nonlinearities (e.g., sigmoid $\sigma(\cdot)$ or softmax). Applying softmax, for instance, to the coefficients that generate $\mathbf{a}_r(\mathbf{x}_t)$ could be interpreted as a form of Mixture-of-Heads, where the network learns to dynamically weight different head configurations based on the input context.

**Discussion.** These variants highlight the flexibility of the TPA framework, allowing for different trade-offs between memory efficiency, computational cost, and model expressiveness. By carefully choosing which factor components (head-dimension or token-dimension) are contextual versus non-contextual, and by adjusting the ranks $(R_Q, R_K, R_V)$, TPA can not only unify existing mechanisms like MHA, MQA, and GQA but also significantly reduce KV cache size—potentially by an order of magnitude—during autoregressive inference.

# H More on Experiments

## H.1 Experimental Settings

We list the main architecture hyper-parameters and training devices in Table 9. For all models, the head dimension $d_h$ is fixed at 64. Specific architectural choices include: 2 KV heads for GQA models; a residual key dimension $d_h^R = 32$ for MLA models; and ranks $R_K = R_V = 2$ and $R_Q = 6$ for TPA and TPA-KVonly models, unless otherwise specified. Other relevant hyper-parameters are listed in Table 10.

**Training Setup Details.** We follow the `nanoGPT` training configuration [24]. In particular, we use the AdamW [35] optimizer with $(\beta_1, \beta_2) = (0.9, 0.95)$, a weight decay of 0.1, and gradient clipping at 1.0. We follow the same setting as `nanoGPT` that the learning rate is managed by a cosine annealing scheduler [36] with 2,000 warmup steps and a (total) global batch size of 480. For the *small*, *medium*, *large* and *XL* models, we set maximum learning rates of $6 \times 10^{-4}, 3 \times 10^{-4}, 2 \times 10^{-4}$, and $1 \times 10^{-4}$ (respectively), and minimum learning rates of $3 \times 10^{-5}, 6 \times 10^{-5}, 1 \times 10^{-5}$, and $1 \times 10^{-5}$ (respectively).

Table 9: The architecture hyper-parameters and training devices of models. Abbreviations: BS. = Batch Size, GAS. = Gradient Accumulation Steps.

| MODEL SIZE | PARAMETERS | DEVICES | MICRO BS. | GAS. | #LAYERS | $d_{\text{MODEL}}$ |
|---|---|---|---|---|---|---|
| SMALL | 124M | 4× A100 GPUs | 24 | 5 | 12 | 768 |
| MEDIUM | 353M | 8× A100 GPUs | 20 | 3 | 24 | 1024 |
| LARGE | 772M | 8× A100 GPUs | 15 | 4 | 36 | 1280 |
| XL | 1.55B | 8× A100 GPUs | 6 | 10 | 48 | 1600 |

## H.2 Additional Experimental Results

### H.2.1 Perplexity Curves

We display the perplexity curves for medium, large, and XL size models in Figure 9.

Table 10: The architecture hyper-parameters for different models.

| MODEL SIZE | SMALL | MEDIUM | LARGE | XL |
|---|---|---|---|---|
| $h$ (MHA) | 12 | 16 | 20 | 25 |
| $h$ (MQA) | 23 | 31 | 39 | 49 |
| $h$ (GQA) | 22 | 30 | 38 | 48 |
| $h$ (MLA) | 12 | 23 | 34 | 49 |
| $h$ (TPA-KVONLY) | 22 | 29 | 37 | 47 |
| $h$ (TPA) | 34 | 47 | 61 | 78 |
| $d_c$ (MLA) | 256 | 512 | 512 | 512 |
| $d_c'$ (MLA) | 512 | 1024 | 1024 | 1024 |

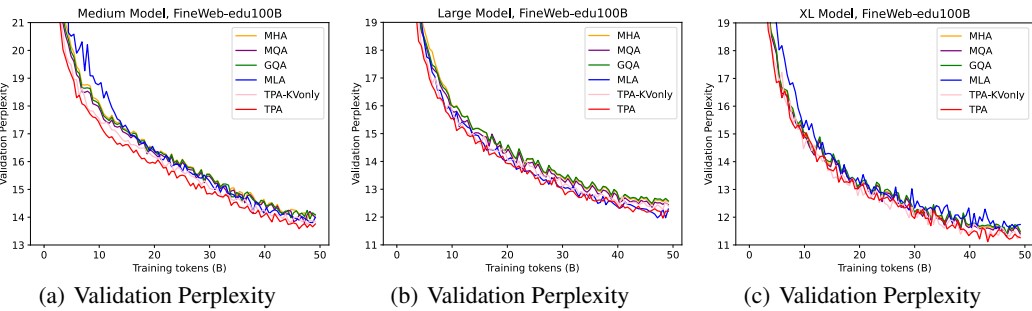

(a) Validation Perplexity  (b) Validation Perplexity  (c) Validation Perplexity

Figure 9: The validation perplexity of medium-size (353M) models, large-size (773M), and XL-size (1.5B) models with different attention mechanisms on the FineWeb-Edu 100B dataset.

### H.2.2 Ablation Study on Different Ranks

Figure 10 illustrates the training loss, validation loss, and validation perplexity for XL-sized (1.5B parameters) TPA models with varying key/value ranks ($R_K = R_V = R$, as indicated in the figure legend), trained on the FineWeb-Edu 100B dataset. Corresponding 0-shot evaluation results are presented in Table 12 (rows for TPA-KVonly with different $R_{K,V}$). These results indicate that increasing the ranks for key and value factorizations generally improves the performance of the TPA models.

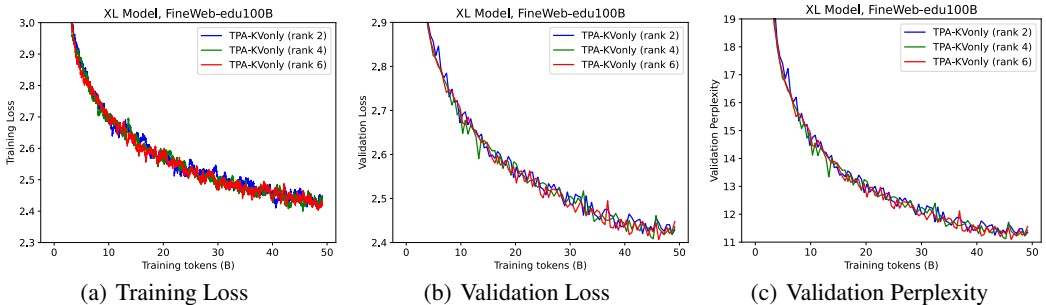

(a) Training Loss  (b) Validation Loss  (c) Validation Perplexity

Figure 10: The training loss, validation loss and validation perplexity curves of XL-size (1.5B) TPA models with different key/value ranks ($R_K = R_V = R$) on the FineWeb-Edu 100B dataset.

### H.2.3 0-shot Evaluation with lm-evaluation-harness

We present 0-shot evaluation results using the lm-evaluation-harness for small (124M parameters) and XL (1.5B parameters) models in Tables 11 and 12, respectively.

### H.2.4 2-shot Evaluation with lm-evaluation-harness

Similarly, 2-shot evaluation results are provided in Tables 13 (Small), 14 (Medium), 15 (Large), and 16 (XL).

Table 11: Evaluation results of small models (124M) with different attention mechanisms, pre-trained on FineWeb-Edu 100B dataset (0-shot with lm-evaluation-harness). The best scores in each column are **bolded**. Abbreviations: HellaSw. = HellaSwag, W.G. = WinoGrande.

| Method | ARC-E | ARC-C | BoolQ | HellaSw. | OBQA | PIQA | W.G. | MMLU | SciQ | Avg. |
|---|---|---|---|---|---|---|---|---|---|---|
| MHA | 50.63 | 26.96 | **59.39** | 36.18 | 32.00 | 64.96 | **51.85** | 23.40 | 70.30 | 46.19 |
| MQA | 49.62 | 25.34 | 55.72 | 35.94 | 31.40 | 64.85 | 51.30 | 23.37 | 68.70 | 45.14 |
| GQA | 48.70 | 25.68 | 56.15 | 35.58 | 31.40 | 64.91 | 51.62 | 23.12 | 68.20 | 45.04 |
| MLA | 50.21 | 26.71 | 58.01 | 36.25 | **32.80** | 64.69 | 50.59 | **24.67** | 71.90 | 46.20 |
| **TPA-KVonly** | 51.05 | 26.54 | 57.25 | **36.77** | 32.60 | **65.02** | 50.91 | 23.64 | 69.70 | 45.94 |
| **TPA** | **51.26** | **27.39** | 57.00 | 36.68 | **32.80** | 64.47 | 49.72 | 24.61 | **72.00** | **46.21** |

Table 12: Evaluation results of XL models (1.5B) with different attention mechanisms, pre-trained on the FineWeb-Edu 100B dataset (0-shot with lm-evaluation-harness). The best scores in each column are **bolded**. Abbreviations: HellaSw. = HellaSwag, W.G. = WinoGrande. If not specified, TPA and TPA-KVonly models use $R_K = R_V = 2$.

| Method | ARC-E | ARC-C | BoolQ | HellaSw. | OBQA | PIQA | W.G. | MMLU | SciQ | Avg. |
|---|---|---|---|---|---|---|---|---|---|---|
| MHA | 64.81 | 35.41 | 61.90 | 54.32 | 37.20 | 72.74 | 55.80 | 25.44 | **82.80** | 54.49 |
| MQA | 64.10 | 36.01 | 62.26 | 54.38 | 39.00 | 72.58 | 56.43 | 23.70 | 81.90 | 54.48 |
| GQA | 63.68 | 35.92 | 60.46 | 54.17 | 38.40 | **73.56** | 56.27 | 24.77 | 81.70 | 54.33 |
| MLA | 64.14 | 35.92 | 60.12 | 53.60 | 39.20 | 72.25 | 55.17 | 24.71 | 81.60 | 54.08 |
| **TPA-KVonly** | 65.61 | 36.77 | **63.02** | 54.17 | 37.00 | 73.34 | 54.62 | 25.02 | 81.60 | 54.57 |
| **TPA-KVonly** ($R_{K,V} = 4$) | 64.52 | **37.03** | 63.27 | **54.89** | 39.80 | 72.91 | 56.51 | 24.74 | 81.60 | **55.03** |
| **TPA-KVonly** ($R_{K,V} = 6$) | 65.78 | 35.92 | 61.71 | 54.86 | 38.60 | 72.69 | **57.93** | **25.59** | 82.20 | **55.03** |
| **TPA** | **66.71** | 36.52 | 61.38 | 54.03 | **40.40** | 72.52 | 56.83 | 24.49 | 82.20 | 55.01 |

Table 13: Evaluation results of small models (124M) with different attention mechanisms, pre-trained on FineWeb-Edu 100B dataset (2-shot with lm-evaluation-harness). The best scores in each column are **bolded**. Abbreviations: HellaSw. = HellaSwag, W.G. = WinoGrande.

| Method | ARC-E | ARC-C | BoolQ | HellaSw. | OBQA | PIQA | W.G. | MMLU | SciQ | Avg. |
|---|---|---|---|---|---|---|---|---|---|---|
| MHA | **57.66** | **28.24** | 57.28 | 36.43 | 29.60 | 64.09 | 51.14 | **26.57** | **82.00** | **48.11** |
| MQA | 53.79 | 26.35 | 44.95 | 34.18 | 28.80 | 62.79 | 52.01 | 25.91 | 78.10 | 45.21 |
| GQA | 55.01 | 25.94 | 55.72 | 35.68 | **31.80** | **65.29** | 51.93 | 25.27 | 77.80 | 47.16 |
| MLA | 54.76 | 27.13 | **58.07** | 36.13 | 31.40 | 65.07 | 51.30 | 25.90 | 78.90 | 47.63 |
| **TPA-KVonly** | 54.25 | 27.90 | 57.06 | 36.36 | **31.80** | 64.31 | **53.59** | 26.18 | 79.20 | 47.85 |
| **TPA** | 57.53 | 28.07 | 56.33 | **36.49** | **31.80** | 64.36 | 51.14 | 25.92 | 79.70 | 47.93 |

Table 14: Evaluation results of medium models (353M) with different attention mechanisms, pre-trained on FineWeb-Edu 100B dataset (2-shot with lm-evaluation-harness, default LR $6 \times 10^{-4}$). The best scores in each column are **bolded**. Abbreviations: HellaSw. = HellaSwag, W.G. = WinoGrande.

| Method | ARC-E | ARC-C | BoolQ | HellaSw. | OBQA | PIQA | W.G. | MMLU | SciQ | Avg. |
|---|---|---|---|---|---|---|---|---|---|---|
| MHA | 64.73 | 32.42 | 58.29 | 45.89 | 34.20 | 68.50 | 53.20 | **25.86** | 88.00 | 52.34 |
| MQA | 64.98 | 33.62 | 55.02 | 45.81 | 34.00 | 69.59 | 53.43 | 24.30 | 85.20 | 51.77 |
| GQA | 65.24 | 33.19 | 56.54 | 45.41 | 34.80 | 69.04 | **55.72** | 24.73 | 87.90 | 52.51 |
| MLA | 64.98 | 33.62 | 53.52 | 45.94 | 33.00 | 68.55 | 51.85 | 25.46 | 89.10 | 51.78 |
| **TPA-KVonly** | 64.69 | 32.34 | **59.48** | 46.23 | **35.40** | **70.08** | 54.06 | 25.64 | 86.30 | 52.69 |
| **TPA** | **67.97** | **34.56** | 57.22 | **46.87** | 34.60 | 69.91 | 52.01 | 25.07 | **89.90** | **53.12** |

Table 15: Evaluation results of large models (772M) with different attention mechanisms, pre-trained on the FineWeb-Edu 100B dataset (2-shot with lm-evaluation-harness). The best scores in each column are **bolded**. Abbreviations: HellaSw. = HellaSwag, W.G. = WinoGrande.

| Method | ARC-E | ARC-C | BoolQ | HellaSw. | OBQA | PIQA | W.G. | MMLU | SciQ | Avg. |
|---|---|---|---|---|---|---|---|---|---|---|
| MHA | 67.85 | 36.35 | 59.82 | 50.22 | 35.00 | 70.67 | 53.35 | 23.92 | 91.10 | 54.25 |
| MQA | 68.86 | 36.09 | 53.79 | 50.50 | **37.00** | 70.89 | **54.70** | 25.01 | 88.00 | 53.87 |
| GQA | 69.15 | 36.09 | 58.84 | 50.29 | 36.20 | 70.73 | 54.22 | **26.08** | 90.00 | 54.62 |
| MLA | 70.54 | **38.74** | **61.50** | **51.86** | 36.00 | 70.89 | 54.22 | 25.47 | **92.40** | **55.74** |
| **TPA-KVonly** | **71.34** | 37.71 | 59.76 | 51.10 | 36.00 | **71.49** | 54.62 | 25.83 | 90.10 | 55.33 |
| **TPA** | 70.41 | 37.71 | 60.06 | 51.30 | 34.00 | 71.06 | 54.54 | 25.79 | 90.30 | 55.02 |

Table 16: Evaluation results of XL models (1.5B) with different attention mechanisms, pre-trained on the FineWeb-Edu 100B dataset (2-shot with lm-evaluation-harness). The best scores in each column are **bolded**. Abbreviations: HellaSw. = HellaSwag, W.G. = WinoGrande. If not specified, $R_K = R_V = 2$ for TPA and TPA-KVonly models.

| Method | ARC-E | ARC-C | BoolQ | HellaSw. | OBQA | PIQA | W.G. | MMLU | SciQ | Avg. |
|---|---|---|---|---|---|---|---|---|---|---|
| MHA | 70.83 | 39.93 | 59.85 | 54.05 | 36.20 | 72.52 | 55.17 | 25.42 | 91.70 | 56.18 |
| MQA | 71.34 | 39.76 | 58.93 | 54.27 | 39.40 | 72.96 | 57.38 | 24.74 | 91.90 | 56.74 |
| GQA | 71.17 | 39.08 | 60.18 | 54.05 | 37.40 | 73.07 | 56.35 | 24.87 | **92.20** | 56.49 |
| MLA | 70.79 | 37.54 | 50.83 | 53.33 | **40.00** | 72.09 | 56.51 | 24.93 | 91.80 | 55.31 |
| **TPA-KVonly** | 72.85 | 39.68 | 60.92 | 53.81 | 37.00 | **73.34** | 56.83 | **26.19** | 91.30 | 56.88 |
| **TPA-KVonly** ($R_{K,V} = 4$) | 72.98 | **40.27** | 60.15 | **54.88** | 36.80 | 73.29 | 56.43 | 25.50 | 92.10 | 56.93 |
| **TPA-KVonly** ($R_{K,V} = 6$) | **73.95** | 39.76 | 58.99 | 54.73 | 36.80 | 72.91 | **59.04** | 24.93 | 92.90 | **57.11** |
| **TPA** | 71.76 | 39.16 | **61.25** | 53.74 | 37.80 | 72.80 | 55.49 | 23.86 | 90.70 | 56.28 |

### H.3 Ablation Studies on Learning Rates

To assess sensitivity to learning rates, we conducted parallel experiments on medium-sized models using a learning rate of $3 \times 10^{-4}$ (compared to the default $6 \times 10^{-4}$ used for other medium model results). The training loss, validation loss, and validation perplexity curves are shown in Figure 11. Performance on standard benchmarks for these models trained with the $3 \times 10^{-4}$ learning rate are reported in Tables 17 (0-shot) and 18 (2-shot). The results demonstrate that TPA and TPA-KVonly maintain their performance advantages over other attention mechanisms even with this alternative learning rate.

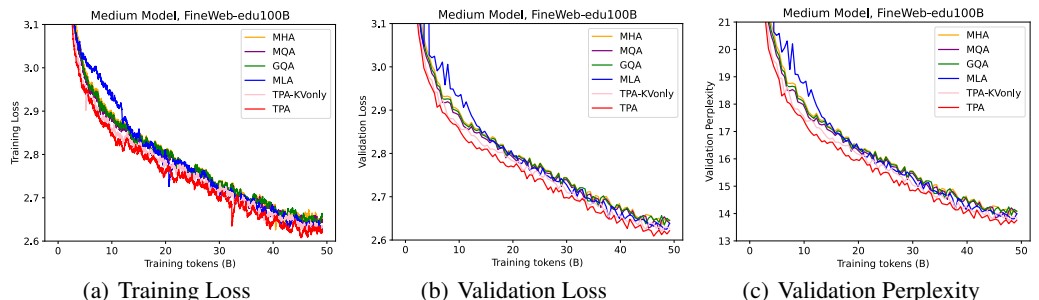

|     (a) Training Loss     |     (b) Validation Loss     |     (c) Validation Perplexity     |

Figure 11: The training loss, validation loss, and validation perplexity of medium-size (353M) models (learning rate $3 \times 10^{-4}$) with different attention mechanisms on the FineWeb-Edu 100B dataset.

Table 17: The evaluation results of medium models (learning rate $3 \times 10^{-4}$) with different attention mechanisms pretrained using the FineWeb-Edu 100B dataset (0-shot with lm-evaluation-harness). The best scores in each column are **bolded**. Abbreviations: HellaSw. = HellaSwag, W.G. = WinoGrande.

| Method | ARC-E | ARC-C | BoolQ | HellaSw. | OBQA | PIQA | W.G. | MMLU | SciQ | Avg. |
|---|---|---|---|---|---|---|---|---|---|---|
| MHA | 56.52 | 29.27 | 58.84 | 44.06 | 35.00 | 68.44 | 51.07 | 25.35 | 76.40 | 49.44 |
| MQA | 55.68 | 28.24 | 60.86 | 44.17 | **35.20** | 68.66 | 52.72 | 25.14 | 72.90 | 49.29 |
| GQA | 54.88 | 29.61 | 56.36 | 43.77 | **35.20** | 68.82 | 52.57 | **25.41** | 74.80 | 49.05 |
| MLA | **59.64** | 29.78 | 60.73 | 45.17 | 34.20 | 68.66 | 52.80 | 25.34 | 75.70 | 50.22 |
| **TPA-KVonly** | 57.11 | 30.03 | **61.25** | 44.83 | 34.60 | 69.04 | **54.54** | 23.35 | 74.60 | 49.93 |
| **TPA** | 59.30 | **31.91** | 60.98 | **45.57** | 34.60 | **69.48** | 53.91 | 24.93 | **77.20** | **50.88** |

## I   Broader Impacts and Limitations

This work allows for the processing of much longer sequences of information with limited hardware resources by reducing the KV cache size. This could make advanced AI capabilities accessible to entities with limited computational budgets, potentially fostering improvement on downstream tasks, including in-depth document analysis, complicated-context reasoning, and code generation, promoting innovation across various sectors in fields of scientific research, education, and software development.

Table 18: The evaluation results of medium models (learning rate $3 \times 10^{-4}$) with different attention mechanisms pre-trained using the FineWeb-Edu 100B dataset (2-shot with lm-evaluation-harness). The best scores in each column are **bolded**. Abbreviations: HellaSw. = HellaSwag, W.G. = WinoGrande.

| Method | ARC-E | ARC-C | BoolQ | HellaSw. | OBQA | PIQA | W.G. | MMLU | SciQ | Avg. |
|---|---|---|---|---|---|---|---|---|---|---|
| MHA | 64.44 | 32.85 | **59.05** | 44.18 | 33.20 | 68.72 | 50.12 | **26.01** | 87.40 | 51.77 |
| MQA | 64.27 | 32.94 | 57.71 | 44.36 | 31.80 | 68.01 | 51.70 | 25.99 | 86.00 | 51.42 |
| GQA | 61.70 | 32.17 | 52.81 | 43.99 | 33.80 | 68.50 | 53.35 | 24.44 | 86.40 | 50.80 |
| MLA | 65.95 | 31.48 | 50.98 | 44.99 | 32.20 | 68.93 | 51.93 | 25.89 | 88.80 | 51.24 |
| **TPA-KVonly** | 65.99 | 33.70 | 57.49 | 44.47 | **34.20** | **69.53** | 53.28 | 24.23 | 86.50 | 52.15 |
| **TPA** | **66.54** | **34.47** | 58.96 | **45.35** | 33.00 | 69.21 | **53.99** | 24.51 | **91.30** | **53.04** |

Although our work proposes a KV-cache efficient architecture for large language models, it may contain certain limitations. For instance, generalization to other modalities deserves more extensive investigation.

