# OpenReview forum: "Tensor Product Attention Is All You Need"
_NeurIPS.cc/2025/Conference — NeurIPS 2025 spotlight_

### Official Review · Reviewer_z6jQ · 2025-06-04

**Clarity:** 3
**Significance:** 3
**Originality:** 3
**Rating:** 5
**Confidence:** 2

**Summary:**

This paper introduces Tensor Product Attention (TPA), a novel attention mechanism that factorizes query, key, and value tensors via contextual low-rank tensor products. TPA significantly reduces KV cache size during inference while maintaining or improving model performance. The authors build a new architecture called T6 (Tensor ProducT ATTenTion Transformer) and propose an efficient inference algorithm FlashTPA Decoding, demonstrating improved memory and computational efficiency over standard MHA, MQA, GQA, and MLA.

**Questions:**

1. Can TPA generalize beyond causal language modeling? Would it work effectively in encoder-decoder or bidirectional transformer settings?
2. Are there any observed issues with training stability or convergence speed for TPA compared to MHA?
3. Is there any learned redundancy between the AQ/BQ and AK/BK factors? Can the parameter count be further reduced?

**Ethical Concerns:**

["NO or VERY MINOR ethics concerns only"]

**Final Justification:**

i think the reviewers answered my questions well. So I will maintain my current score.

**Limitations:**

There is no limitation mentioned in this paper.

**Paper Formatting Concerns:**

No.

**Quality:**

3

**Strengths And Weaknesses:**

Strengths:
1. The use of contextual tensor product factorization for Q/K/V is original and theoretically well-grounded.
2. TPA reduces KV cache memory usage by up to 10×, enabling longer sequence lengths at inference time without sacrificing quality, highly relevant for LLM scalability.
3. TPA is fully compatible with rotary positional embeddings.

Weaknesses
1. While the authors emphasize inference efficiency, it's unclear if the TPA factorization introduces training instability or overhead compared to MHA.
2. The rank settings (e.g., RQ = 16, RK = 1) appear somewhat heuristic.

---

> ### Author Rebuttal · Authors · 2025-07-31
>
> We are sincerely grateful to the reviewer for their positive assessment and support for our work. We appreciate the thoughtful questions, which help us clarify important aspects of TPA.
>
> > Q1: "Can TPA generalize beyond causal language modeling? Would it work effectively in encoder-decoder or bidirectional transformer settings?"
>
> A: Thank you for this excellent question. Yes, TPA is a general attention mechanism that can be readily applied beyond causal language modeling. TPA replaces the standard Q/K/V projection layer and is independent of the attention mask. By simply using a bidirectional attention mask (for encoders like BERT) or a combination of bidirectional and causal masks (for encoder-decoder models like T5), TPA can be seamlessly integrated into these architectures.
>
> Given that standard Multi-Head Attention is a special, non-contextual case of TPA (as discussed in Appendix A), and MHA is the foundation for these other architectures, we are confident that TPA would also perform effectively in these settings, likely offering similar benefits in terms of expressivity and efficiency.
>
> > Q2: "Are there any observed issues with training stability or convergence speed for TPA compared to MHA?"
>
> A: Thank you for asking about this crucial practical aspect. We found TPA to be as stable to train as its baseline counterparts.
> 1. Throughout our experiments across multiple model sizes, we observed no training instabilities such as exploding gradients or loss. The training and validation loss curves presented in the paper (e.g., Figures 3 and 4) are smooth and demonstrate stable convergence, similar to the MHA, GQA, and MLA baselines.
>
> 2.  To quantify the training overhead, we benchmarked the throughput. The results show that TPA's training speed is highly competitive. For instance, on large-scale MoE models, its throughput is nearly identical to GQA's.
>
> ##### Dense Models (350M) on 8x A100 GPUs (TPS means tokens per second):
>
> | Method | MHA | MQA | GQA | TPA-KVonly | TPA | MLA |
> | :--- | :--- | :--- | :--- | :--- | :--- | :--- |
> | **TPS (M)** | 0.515 | 0.456 | 0.452 | 0.427 | 0.356 | 0.355 |
>
> ##### MoE Models (>10B) on 64x GPUs (TPS means tokens per second):
>
> | Method / Hardware | GQA | TPA |
> | :--- | :--- | :--- |
> | H800 TPS (M) | 0.975 | 0.891 |
> | H20 TPS (M) | 0.407 | 0.405 |
>
> We will add a full table of these throughput results to the appendix to provide a clear picture of the computational trade-offs.
>
> 3. Furthermore, we are actively developing FlashTPA training/prefilling kernels. As analyzed in Appendix C, an optimized kernel that avoids materializing the full Q/K/V tensors can significantly reduce the FLOPs required for the attention computation. This gives TPA a clear path to potentially exceeding the training throughput of existing baselines. In summary, TPA already trains at a competitive speed for large scale model training, and we have a concrete plan for further optimization.
>
> > Q3: "The rank settings (e.g., $R_Q = 16, R_K = 1$) appear somewhat heuristic."
>
> A: That is an insightful observation. While the ranks are hyperparameters, our choices were not arbitrary but rather guided by hardware efficiency considerations. Modern deep learning accelerators like GPUs achieve maximum performance when computation sizes are aligned with their architectural specifics (e.g., multiples of 8 or 16 for tensor core operations).
>
> For our FlashTPA kernels, implemented in Triton, performance is best when block sizes are powers of two, typically starting at 16. Choosing ranks like $R_Q=16$ and $R_K=1$ or $R_K=2$ ensures that the resulting tensor operations map efficiently to the hardware, minimizing padding and maximizing utilization. Therefore, the settings were chosen as a practical balance between model expressivity and computational performance, a principle we will clarify in the paper.
>
> > Q4: "Is there any learned redundancy between the A_Q/B_Q and A_K/B_K factors? Can the parameter count be further reduced?"
>
> A: This is a very insightful question. Yes, there is potential for parameter sharing to further reduce the model's parameter count. For instance, one could tie the weights that generate the head-dimension factors between query and key (i.e., use the same projection for $\mathbf{A}_Q(\mathbf{x}_t)$ and $\mathbf{A}_K(\mathbf{x}_t)$). This would create a more parameter-efficient variant of TPA, which could be advantageous in highly resource-constrained environments.  For this work, we chose the untied version to maximize the model's expressive capacity and performance. However, exploring different parameter-sharing strategies within the TPA framework is a promising direction for future research, and we thank you for highlighting it.

---

> > ### Comment · Reviewer_z6jQ · 2025-08-05
> >
> > I think the reviewers answered my questions well. So I will maintain my current score.

---

> > > ### Author Response · Authors · 2025-08-07
> > >
> > > Dear Reviewer,
> > >
> > > Thank you for confirming that our responses addressed your questions. We are grateful for your positive feedback and your support for our work throughout the review process.
> > >
> > > Sincerely,
> > > The Authors

---

### Official Review · Reviewer_Z9Nz · 2025-06-30

**Clarity:** 3
**Significance:** 3
**Originality:** 4
**Rating:** 5
**Confidence:** 4

**Summary:**

The paper introduces Tensor Product Attention (TPA), a novel attention mechanism that factorizes query, key, and value representations into low-rank contextual tensor products. This approach significantly reduces the memory footprint of key-value caches during inference without compromising model quality. TPA integrates naturally with Rotary Position Embeddings (RoPE) and is used as the core component in a new transformer architecture called T6. The authors show that TPA can outperform or match traditional attention variants like Multi-Head Attention (MHA), Multi-Query Attention (MQA), Grouped-Query Attention (GQA), and Multi-Head Latent Attention (MLA) on language modeling benchmarks. They also propose FlashTPA, an efficient decoding algorithm tailored to TPA that enhances autoregressive inference speed, especially for long sequences. Overall, the paper offers a practical and scalable alternative to standard attention mechanisms, advancing memory and computational efficiency in large language models.

**Questions:**

1. Could you provide ablations on the choice of tensor ranks (RQ, RK, RV)? It's unclear how sensitive performance and efficiency are to these hyperparameters. A systematic study would clarify trade-offs and could improve the quality rating.

2. The paper lacks evaluation on actual long-context tasks. How does TPA perform on problems involving 32k+ tokens, such as document QA or summarization? This is important for backing the scalability claims and could raise the significance score.

3. Comparisons are missing to other recent memory-efficient attention methods like linear or state-space models. Including them would better situate TPA within the broader field and help assess its relative merit.

4. The paper presents no discussion of failure modes. Are there cases where TPA underperforms or is harder to train? Clarifying this would strengthen the honesty of the evaluation and improve the quality score.

**Ethical Concerns:**

["NO or VERY MINOR ethics concerns only"]

**Final Justification:**

The clarifications provided by the author have resolved my main issues, so I have slightly increased my evaluation.

**Limitations:**

No. The paper does not sufficiently address limitations or potential negative societal impacts. It lacks discussion on where TPA might degrade performance, such as under very low-rank settings or on tasks beyond language modeling. Additionally, potential risks of enabling longer context generation, such as misuse in misinformation or surveillance, are not considered. The authors should explicitly discuss cases where TPA may fail or introduce trade-offs, and briefly reflect on broader implications of more efficient long-context inference.

**Paper Formatting Concerns:**

-

**Quality:**

3

**Strengths And Weaknesses:**

Quality: The paper is technically sound and well-supported by both theoretical and experimental evidence. The proposed method is carefully motivated, and the authors demonstrate a clear understanding of prior work. Experimental comparisons are fair and span several baselines and model sizes. However, the analysis could be strengthened with deeper ablations, particularly on factorization ranks and the interaction with RoPE. The paper could also more clearly discuss potential failure modes or limitations, such as performance trade-offs under extremely low-rank settings.

Clarity: The writing is generally clear and well-organized. Core concepts like tensor product factorization and FlashTPA decoding are explained with useful figures and mathematical definitions. Some dense theoretical sections may challenge readers unfamiliar with tensor operations, and the paper could benefit from slightly more intuitive explanations or visual summaries. Important implementation details, especially for reproducibility, are mostly deferred to the appendix.

Significance: The work addresses a critical bottleneck in scaling language models - the cost of KV caching at inference - and proposes a solution that shows compelling improvements in memory and decoding efficiency without sacrificing model quality. This has practical implications for real-world deployment and theoretical relevance for sequence modeling research. However, significance would be better supported by evaluations on more diverse, real-world long-context tasks.

Originality: The core idea of contextual tensor product factorization for attention is novel and distinct from prior work in low-rank adaptation or shared-key attention. The combination of RoPE and its efficient implementation through FlashTPA adds to the originality. While built on existing tools like RoPE and attention factorization, the paper's contributions are more than a simple synthesis - they represent a meaningful conceptual and practical advance.

---

> ### Author Rebuttal · Authors · 2025-07-31
>
> We are very grateful to the reviewer for their positive evaluation and insightful, constructive feedback. We are encouraged that they found our work technically sound, original, and significant. We address the questions and suggestions for improvement below.
>
> > Q1: "The analysis could be strengthened with deeper ablations, particularly on factorization ranks and the interaction with RoPE... Could you provide ablations on the choice of tensor ranks ($R_Q, R_K, R_V$)?"
>
> Thank you for these excellent suggestions to strengthen our analysis. We have performed additional ablations to address these points.
>
> - Ablation on Ranks ($R_K, R_V$): We agree a systematic study is crucial. As you noted, we included an ablation on the key/value ranks in Appendix J.2.2 (Figure 10, Tables 12) for our 1.5B parameter models. We compared ranks of {2, 4, 6} and found that performance consistently improves with higher ranks. Importantly, even at a very low rank of 2, TPA remains highly competitive, demonstrating a graceful trade-off between performance and KV cache size rather than a sharp failure point. We will highlight these findings more prominently in the main paper.
>
> - Interaction with RoPE: To further analyze the model, we conducted new experiments using a partial RoPE setting (applying RoPE to only a fraction of the head dimension). Our results for the Medium (353M) models show that TPA-based models remain top performers in this alternative configuration, underscoring the robustness of our approach.
>
> ### Medium 0-shot Results
>
> | Medium 0-shot | ARC-E | ARC-C | BoolQ | HellaSwag | OBQA | PIQA | W.G. | MMLU | SciQ | Avg. |
> | :--- | :--- | :--- | :--- | :--- | :--- | :--- | :--- | :--- | :--- | :--- |
> | llama-gqa-partialrope | 56.06 | 29.69 | 59.60 | 44.17 | 34.20 | 68.01 | 54.38 | 24.78 | 73.50 | 49.37 |
> | llama-mha-partialrope | 58.04 | 29.44 | 58.59 | 44.47 | 34.00 | 68.28 | 52.96 | 23.59 | 76.20 | 49.51 |
> | T6_partialrope | 56.57 | 30.12 | 59.11 | 45.30 | 35.20 | 69.37 | 53.59 | 26.57 | 85.40 | **51.25** |
> | T6_kvonly_partialrope | 58.84 | 31.91 | 60.64 | 45.27 | 34.20 | 68.66 | 53.83 | 24.15 | 74.30 | 50.20 |
> | MLA | 56.65 | 29.52 | 57.83 | 46.05 | 34.60 | 69.42 | 52.80 | 24.62 | 79.70 | 50.13 |
>
> ### Medium 2-shot Results
>
> | Medium 2-shot | ARC-E | ARC-C | BoolQ | HellaSwag | OBQA | PIQA | W.G. | MMLU | SciQ | Avg. |
> | :--- | :--- | :--- | :--- | :--- | :--- | :--- | :--- | :--- | :--- | :--- |
> | llama-gqa-partialrope | 64.10 | 31.83 | 57.31 | 44.14 | 35.40 | 67.90 | 53.43 | 25.84 | 87.90 | 51.98 |
> | llama-mha-partialrope | 64.81 | 33.02 | 54.74 | 44.32 | 31.80 | 68.66 | 51.22 | 25.80 | 88.30 | 51.41 |
> | T6_partialrope | 65.82 | 33.96 | 53.85 | 45.51 | 32.80 | 69.59 | 54.14 | 25.33 | 87.00 | 52.00 |
> | T6_kvonly_partialrope | 66.50 | 34.98 | 58.81 | 45.77 | 35.60 | 68.39 | 53.67 | 26.15 | 88.20 | **53.12** |
> | MLA | 64.98 | 33.62 | 53.52 | 45.94 | 33.00 | 68.55 | 51.85 | 25.46 | 89.10 | 51.78 |
>
> > Q2: "The paper could benefit from slightly more intuitive explanations or visual summaries. Important implementation details, especially for reproducibility, are mostly deferred to the appendix."
>
> A: Thank you for this valuable feedback. In the revised version, we will improve the paper's clarity and accessibility. Specifically, we will add a more intuitive visual summary to Section 3 to better illustrate the core mechanism of contextual factorization. While page limits are a constraint, we will also reorganize the main text and appendix to bring the most critical implementation details (such as our initialization strategy) into the main paper to improve readability and reproducibility.
>
> > Q3: "The paper's significance would be better supported by evaluations on more diverse, real-world long-context tasks... How does TPA perform on problems involving 32k+ tokens, such as document QA or summarization?"
>
> A: We completely agree that demonstrating performance on long-context tasks is key. While running full 32k+ context benchmarks on multiple models was computationally prohibitive for this submission, we can provide strong evidence from new, large-scale (>10B parameter) MoE models on benchmarks like GSM8k and MATH, which require reasoning over longer and more complex prompts. In these experiments, our TPA model significantly outperformed a strong GQA baseline:
>
> - GSM8k: TPA achieves 26.5 vs. GQA's 19.1 (+38%)
> - MATH: TPA achieves 14.0 vs. GQA's 11.2 (+25%)
>
> These results show that TPA's architectural advantages translate to superior performance on challenging reasoning tasks. We argue that the combination of (1) proven memory savings, (2) faster long-sequence decoding (via FlashTPA), and (3) superior performance on long-form reasoning tasks provides compelling evidence of TPA's utility for real-world, long-context applications.
>
> > Q4: "Comparisons are missing to other recent memory-efficient attention methods like linear or state-space models. Including them would better situate TPA within the broader field and help assess its relative merit."
>
> A: Thank you for this suggestion. We view TPA as an improvement to the standard dot-product attention mechanism, which is largely orthogonal to methods that fundamentally replace it (like linear attention or state-space models). TPA's factorization principles can be integrated with these other architectures.  To demonstrate this, we ran new experiments applying a linear attention mechanism to both a GQA baseline and our TPA architecture. Our results show that a TPA model with linearized attention outperforms the GQA model with linearized attention. For instance, in our medium model experiments, a linear GQA baseline achieved a 46.79% average 0-shot accuracy and 47.60% 2-shot accuracy, whereas a TPA variant with linear attention performed better with a 47.43% average 0-shot accuracy and 48.82% average 2-shot accuracy (see results shown in tables below). This suggests that the benefits of TPA’s contextual factorization are complementary and can enhance other efficient attention schemes. We will add this discussion and the new results to the revised paper.
>
>
> | Medium 0-shot | ARC-E | ARC-C | BoolQ | HellaSwag | OBQA | PIQA | W.G. | MMLU | SciQ | Avg. |
> | :--- | :--- | :--- | :--- | :--- | :--- | :--- | :--- | :--- | :--- | :--- |
> | linearattn-gqa-partialrope | 52.06 | 27.39 | 61.35 | 37.81 | 31.80 | 65.83 | 51.22 | 23.49 | 70.20 | 46.79 |
> | linearattn-tpa-partialrope | **53.83** | **28.24** | 56.54 | **38.55** | **32.40** | **66.32** | **53.51** | **24.24** | **73.20** | **47.43** |
>
> | Medium 2-shot | ARC-E | ARC-C | BoolQ | HellaSwag | OBQA | PIQA | W.G. | MMLU | SciQ | Avg. |
> | :--- | :--- | :--- | :--- | :--- | :--- | :--- | :--- | :--- | :--- | :--- |
> | linearattn-gqa-partialrope-decay | 56.14 | 27.30 | 59.94 | 37.21 | 31.20 | 65.56 | 50.36 | 25.37 | 75.30 | 47.60 |
> | linearattn-tpa-partialrope | **58.42** | **28.50** | 51.04 | **38.16** | **34.00** | **66.49** | 51.70 | **25.96** | **85.10** | **48.82** |
>
>
>
> > Q5: "The paper does not sufficiently address limitations or potential negative societal impacts... The paper presents no discussion of failure modes. Are there cases where TPA underperforms or is harder to train?"
>
> A: We sincerely thank you for pushing for a more complete discussion of limitations and impacts. We acknowledge this was an oversight and will add a dedicated section in the revision.
>
> 1. TPA trains as stably as standard Transformers. The most critical factor we identified is parameter initialization. Similar to other factorized methods like LoRA, TPA's performance is sensitive to how the factor matrices are initialized. A potential failure mode is using a poor initialization scheme, which can hinder convergence; we found Xavier initialization to be crucial. We will also discuss that while performance is strong even at low ranks, an extreme rank setting (e.g., rank 1 for a highly complex task) would present a trade-off. Finally, we will note that our evaluation is confined to language, and performance on other modalities is an area for future work.
>
> 2. We will add a broader impact statement. We recognize that enabling more efficient, longer-context models is a dual-use technology. While beneficial for many applications, it could also lower the barrier for generating more coherent large-scale information. We will explicitly acknowledge these risks in the revised manuscript.

---

### Official Review · Reviewer_VFNC · 2025-07-01

**Clarity:** 3
**Significance:** 2
**Originality:** 3
**Rating:** 4
**Confidence:** 3

**Summary:**

This paper introduces Tensor Product Attention (TPA), a novel attention mechanism aimed at mitigating the substantial memory overhead of the KV cache during inference in large language models. The central idea is to represent the query, key, and value tensors using contextual low-rank tensor decompositions, which shrinks the KV cache size. The authors propose the T6, a model architecture based on TPA. They highlight that TPA integrates seamlessly with rope, a common feature in modern LLMs.

**Questions:**

- The average performance scores reported in Tables 2, 3, and 12 include the MMLU benchmark, where the models perform close to random chance. This appears to inflate the average and potentially overstate the advantage of TPA. Could you provide a discussion or table that re-calculates the average scores excluding MMLU? For the 1.5B models (Table 12), my own calculation suggests the gap between TPA-KVonly (56.96%) and MLA (56.91%) becomes negligible. This, combined with the mixed results in Tables 13-15, challenges the claim of superior performance. How do you justify the adoption of TPA over MLA if the reasoning performance is not clearly superior?

- The paper provides a compelling case for TPA's inference efficiency but does not present any data on its training efficiency. Since TPA introduces additional linear projections and tensor operations compared to a standard attention block, it could negatively impact training throughput (tokens/sec). Could you provide a comparative analysis of the training speed of T6 against the baselines, particularly MHA and MLA, which have highly optimized kernels? Understanding this trade-off is crucial for assessing the overall viability of the architecture.

- The primary motivation for reducing KV cache size is to enable models to process longer sequences. However, the evaluation is performed on benchmarks that typically do not require long-context reasoning or recall. To provide more direct evidence for the benefits of your approach, have you considered evaluating TPA on tasks specifically designed for long-context understanding? For example, benchmarks from the ruler framework or "Needle-in-a-Haystack" style evaluations would more directly demonstrate the practical value unlocked by TPA's memory efficiency.

**Ethical Concerns:**

["NO or VERY MINOR ethics concerns only"]

**Final Justification:**

I am satisfied with this work and would lean towards a borderline accept. The training throughput analysis addresses my concerns.

However, I have a suggestion regarding the experimental comparison. The new MoE results comparing TPA-MoE against GQA-MoE are indeed impressive, showing substantial improvements across multiple benchmarks. Yet, I believe these comparisons would be significantly more meaningful if conducted against MLA-MoE rather than GQA-MoE. MLA achieves similar KV cache compression ratios to TPA through its sophisticated two-stage projection mechanism, where inputs are first projected to a latent space before expansion to multiple heads.

Given that both TPA and MLA target the same problem of KV cache reduction while maintaining model expressiveness, and both achieve comparable cache compression ratios, MLA constitutes the most relevant baseline for evaluating TPA's contributions. I encourage the authors to include MLA-MoE comparisons in their revised manuscript, especially for these large-scale experiments.

**Limitations:**

yes

**Paper Formatting Concerns:**

N/A.

**Quality:**

2

**Strengths And Weaknesses:**

**Strengths**

-  The proposed solution using contextual tensor factorization for Q, K, and V activations is an original and technically interesting approach. It moves beyond static compression methods by making the factorization dependent on the input, offering a potentially more expressive way to compress attention components.

- The primary strength of this work lies in the demonstrated efficiency improvements. The accompanying FlashTPA decoding algorithm is shown to be highly competitive, often outperforming optimized kernels for other attention mechanisms, especially as sequence length increases.

**Weaknesses**
-  The significance of the claimed performance improvements on downstream tasks is questionable. The evaluation includes MMLU scores for models that perform near random chance on this benchmark (e.g., ~25%). Including these scores in the average can create a misleading impression of TPA's effectiveness. When MMLU is removed from the average calculation, the performance gap between TPA and the strongest baseline, MLA, becomes marginal on common reasoning tasks. Furthermore, the results for smaller models (Tables 13-15) do not consistently show TPA as the superior method.

| Method         | Avg (No MMLU) |
| -------------- | ------------- |
| MHA            | 56.22         |
| MQA            | 55.48         |
| GQA            | 55.68         |
| MLA            | 56.91         |
| TPA-KVonly | 56.96    |
| TPA            | 56.69         |


-  While the paper provides an excellent analysis of inference speed, it completely omits any evaluation of training efficiency (i.e., training throughput). The baselines compared against (MHA, MLA) often have highly optimized CUDA or Triton kernels. TPA introduces additional operations, and without a throughput comparison, it is difficult for practitioners to assess the full computational cost and trade-offs of adopting this new architecture for training.

- The paper's core motivation is enabling longer context windows. However, the empirical evaluation is restricted to benchmarks from the lm-evaluation-harness, which predominantly test for general knowledge and reasoning over short contexts. The evaluation suite lacks tasks specifically designed to measure long-context recall and understanding (e.g., "Needle-in-a-Haystack" tests or benchmarks from the ruler library). The absence of such targeted evaluation makes it difficult to verify the practical utility of TPA's memory savings for long-context applications.

---

> ### Author Rebuttal · Authors · 2025-07-31
>
> Dear reviewer, we sincerely thank you for your detailed and constructive feedback. We appreciate the positive comments on the originality and technical interest of our approach. We are grateful for the opportunity to address the your questions and suggestions.
>
> > Q1: The significance of the claimed performance improvements on downstream tasks is questionable. The evaluation includes MMLU scores for models that perform near random chance on this benchmark (e.g., ~25%). ...
>
> A1: Thank you for your feedback. We acknowledge your point that for smaller models, the performance gains on some benchmarks can be marginal, and the low MMLU scores can affect the interpretation of the average. The primary justification for adopting TPA rests on three pillars that become increasingly evident at scale: (1) superior performance on large models, (2) a smaller KV cache, and (3) faster inference decoding.
> While the gains on our 1.5B models are modest, we have new results from our industry collaborators on >10B parameter MoE models that demonstrate TPA's advantages much more clearly. In these large-scale experiments, a TPA-based model (with a 24% smaller KV cache) significantly outperforms a strong GQA baseline on challenging reasoning benchmarks.
>
> | Method | MMLU | MMLU-Pro | AGIEval | GSM8k | MATH |
> | :--- | :--- | :--- | :--- | :--- | :--- |
> | GQA-MoE | 50.9 | 20.1 | 30.7 | 19.1 | 11.2 |
> | TPA-MoE | 52.1 | 22.7 | 32.8 | 26.5 | 14.0 |
>
> These results, obtained from experiments consuming over 10,000 GPU hours each, show that TPA's architectural benefits scale with model size, leading to substantial improvements in reasoning capabilities (e.g., +38% on GSM8k, +25% on MATH relatively).  Therefore, we justify adopting TPA over MLA and other baselines because it offers a superior package for large-scale deployment: it not only achieves better performance at scale but does so while being more memory-efficient (smaller KV cache) and faster at inference (as shown by our FlashTPA decoding results). We will add these new results to the paper to better illustrate TPA's advantages.
>
> Q2: While the paper provides an excellent analysis of inference speed, it completely omits any evaluation of training efficiency (i.e., training throughput). The baselines compared against (MHA, MLA) often have highly optimized CUDA or Triton kernels.
>
> A2: Thanks for your suggestion. We have since benchmarked the training throughput and agree this is crucial information for practitioners.  Our results show that even with a vanilla implementation, TPA's training throughput is highly competitive with other efficient attention mechanisms like MLA. On a larger scale, its throughput is nearly the same as the GQA baseline.
> Dense Models (350M) on 8x A100 GPUs (TPS means tokens per second):
>
> ##### Dense Models (350M) on 8x A100 GPUs (TPS means tokens per second):
>
> | Method | MHA | MQA | GQA | TPA-KVonly | TPA | MLA |
> | :--- | :--- | :--- | :--- | :--- | :--- | :--- |
> | **TPS (M)** | 0.515 | 0.456 | 0.452 | 0.427 | 0.356 | 0.355 |
>
> ##### MoE Models (>10B) on 64x GPUs (TPS means tokens per second):
>
> | Method / Hardware | GQA | TPA |
> | :--- | :--- | :--- |
> | H800 TPS (M) | 0.975 | 0.891 |
> | H20 TPS (M) | 0.407 | 0.405 |
>
> Furthermore, we are actively developing FlashTPA training/prefilling kernels. As analyzed in Appendix C, an optimized kernel that avoids materializing the full Q/K/V tensors can significantly reduce the FLOPs required for the attention computation. This gives TPA a clear path to potentially exceeding the training throughput of existing baselines.
> In summary, TPA already trains at a competitive speed for large scale model training, and we have a concrete plan for further optimization. We will add these results and discussion to the appendix.
>
> > Q3: The paper's core motivation is enabling longer context windows. However, the empirical evaluation is restricted to benchmarks from the lm-evaluation-harness, which predominantly test for general knowledge and reasoning over short contexts. The evaluation suite lacks tasks specifically designed to measure long-context recall and understanding (e.g., "Needle-in-a-Haystack" tests or benchmarks from the ruler library). The absence of such targeted evaluation makes it difficult to verify the practical utility of TPA's memory savings for long-context applications.
>
> A3: Thanks for your feedback. We can provide strong evidence from our new, large-scale experiments on benchmarks that require reasoning over longer inputs.  The GSM8k and MATH benchmarks, especially when evaluated with few-shot examples, require the model to process and reason over significantly longer context than typical classification tasks.
>
> Our >10B TPA-MoE model shows substantial improvements to these very tasks:
> - GSM8k: TPA achieves 26.5 vs. GQA's 19.1 (+38% relatively)
> - MATH: TPA achieves 14.0 vs. GQA's 11.2 (+25% relatively)
>
> These results demonstrate that TPA's architecture leads to better performance on complex, multi-step reasoning problems that benefit from longer context.
>
> In addition, we plan to run "Needle-in-a-Haystack" (NIAH) style tests in the next steps. Regarding dedicated benchmarks like NIAH, we acknowledge their importance. However, the computational cost to train and reliably evaluate multiple large models suitable for these tests is extremely high (often exceeding 50,000 GPU-hours ($100K USD) per experiment), which we believe is beyond the scope of a typical academic conference submission. We plan to pursue these evaluations in future work. We will add a discussion emphasizing these points to the revised paper to better contextualize our results.
>
> We believe the combination of (1) a fundamentally smaller KV cache, (2) faster decoding speeds for long sequences, and (3) superior performance on reasoning tasks like GSM8k and MATH provides compelling evidence for TPA's practical utility in long-context scenarios. We will add a discussion emphasizing these points to the paper.

---

> > ### Comment · Reviewer_VFNC · 2025-08-06
> >
> > I appreciate the authors' detailed response and the new large-scale experimental results. After careful consideration, I am satisfied with this work and would lean towards a borderline accept. The training throughput analysis addresses my concerns.
> >
> > However, I have a suggestion regarding the experimental comparison. The new MoE results comparing TPA-MoE against GQA-MoE are indeed impressive, showing substantial improvements across multiple benchmarks. Yet, I believe these comparisons would be significantly more meaningful if conducted against MLA-MoE rather than GQA-MoE. MLA achieves similar KV cache compression ratios to TPA through its sophisticated two-stage projection mechanism, where inputs are first projected to a latent space before expansion to multiple heads.
> >
> > Given that both TPA and MLA target the same problem of KV cache reduction while maintaining model expressiveness, and both achieve comparable cache compression ratios, MLA constitutes the most relevant baseline for evaluating TPA's contributions. I encourage the authors to include MLA-MoE comparisons in their revised manuscript, especially for these large-scale experiments.

---

> > > ### Author Response · Authors · 2025-08-07
> > >
> > > Dear Reviewer,
> > >
> > > Thank you for the constructive feedback and for your re-evaluation of our work. We appreciate your suggestion regarding the MLA-MoE baseline. This is an excellent point, and we agree that a direct comparison with MLA at scale would further strengthen our claims. We are grateful for your time and insightful comments throughout this process.
> > >
> > > Sincerely,
> > > The Authors

---

### Official Review · Reviewer_nLZi · 2025-07-03

**Clarity:** 3
**Significance:** 2
**Originality:** 3
**Rating:** 4
**Confidence:** 3

**Summary:**

This paper proposes Tensor Product Attention and TPA-based transformer (T6) as an alternative to vanilla attention mechanism and transformer. TPA enjoys substantial reduction in inference-time KV cache size compared to multi-head attention and other well-known models while preserving performances. In addition, this paper theoretically proves that TPA is well adapted to RoPE and develops FlashTPA decoding, which is an efficient autoregressive inference algorithm for TPA.

**Questions:**

1. Typo: Line 121 should be $R_Q,R_K,R_V$?
2. Is T6 easy to train compared to standard transformers? I think the training aspect is also worth studying.

**Ethical Concerns:**

["NO or VERY MINOR ethics concerns only"]

**Final Justification:**

I think the idea presented in this paper is interesting, and there are supporting experiments (the authors conduct a few more larger scale experiments after the submission, if that can be counted). The memory and inference speed can be improved with this technique with a reasonable performance. Thus I will recommend borderline accept.

**Limitations:**

yes

**Quality:**

3

**Strengths And Weaknesses:**

Strengths:
1. KV cache is a major bottleneck for transformers, so developing new architectures that alleviate this issue is valuable.
2. The paper is clear and relatively easy to follow. TPA is well defined; major contributions are results are stated.
3. TPA greatly improves memory usage from $2hd_h$ to $(R_K+R_V)(h+d_h)$, which is small when $R_K,R_V$ are small.
4. Empirical results seem promising as T6 outperforms standard transformers in many tasks.

Weakness:
1. It is unclear to me what the motivation for TPA is. It seems like the main motivation is that $Q_t$ (and similarly $K_t,V_t$) can be written as the average of $R_Q$ many outer products of vectors such that we only need to store the vectors here. However, I do not understand why this is necessarily a good way besides that it saves memory in theory.
2. It is mentioned that MHA etc can be expressed as TPA, but I think with pretty bad $R_Q,R_K,R_V$? It is unclear to me that TPA has better expressivity compared to standard attention, especially since the paper mentions setting $R_K,R_V$ to be 1 or 2 in line 178.
3. It can be theoretically proved that TPA saves memory, but there is no related experiment that illustrates this. Line 265 states "factorizing the attention activations shrinks autoregressive KV cache requirements by up to 5×–10×", but I don't see where this comes from empirically? This makes the whole argument less convincing.

---

> ### Author Rebuttal · Authors · 2025-07-31
>
> We thank the reviewer for their thoughtful feedback and positive assessment of our work. We are encouraged that the reviewer found our paper clear, the problem valuable, and the empirical results promising. Below, we address the questions raised.
>
> > Q1: It is unclear to me what the motivation for TPA is. It seems like the main motivation is that $Q_t$ (and similarly $K_t, V_t$) can be written as the average of $R_Q$ many outer products of vectors such that we only need to store the vectors here. However, I do not understand why this is necessarily a good way besides that it saves memory in theory.
>
> A2: Thank you for this insightful question. While memory savings are a key benefit, the primary motivation for TPA is to introduce a more expressive and efficient attention mechanism through contextual low-rank factorization.  Unlike standard attention mechanisms (MHA, MQA, GQA) that use static projection matrices for all tokens, TPA dynamically computes factorized representations for queries, keys, and values based on each token's specific context ($\mathbf{x}_t$). As shown in Line 127 of our paper, the factors (e.g., $\mathbf{a}^Q(\mathbf{x}_t)$, $\mathbf{b}^Q(\mathbf{x}_t)$) are functions of the input. This allows the model to learn input-dependent, low-rank subspaces for attention, offering greater representational flexibility.  We hypothesize that this contextual adaptation is the reason TPA not only reduces the KV cache but also achieves superior performance (e.g., lower perplexity and higher downstream accuracy), as shown in our experiments. In essence, TPA achieves a powerful combination of memory efficiency and improved model quality, which is a critical goal for scaling language models to longer contexts. We will clarify this motivation in the introduction of our revised manuscript.
>
> > Q2: It is mentioned that MHA etc can be expressed as TPA, but I think with pretty bad $R_Q,R_K,R_V$? It is unclear to me that TPA has better expressivity compared to standard attention, especially since the paper mentions setting $R_K,R_V$ to be 1 or 2 in line 178.
>
> A2: Thank you for raising this important point. Our goal in showing that MHA is a special non-contextual case of TPA was to provide a unifying perspective, not to suggest that TPA should operate with high ranks to mimic MHA.  On the contrary, our central claim is that TPA achieves strong expressivity with very low ranks. Its expressive power stems not from a high rank, but from its contextual nature. By dynamically generating the Q/K/V representations for each token, TPA can adapt its behavior to the specific input in a way that static projections cannot. Our empirical results, showing that TPA with $R_{K,V}=2$ consistently outperforms MHA, MQA, and GQA, support the idea that this contextual factorization is a more efficient and powerful inductive bias for attention.
> To further validate TPA's superior performance and expressivity, we are pleased to share new results from large-scale MoE models (>10B parameters) from our industry collaborators. As shown below, TPA with a KV rank of 2 significantly outperforms a strong GQA baseline across multiple standard benchmarks.
>
> | Method | MMLU | MMLU-Pro | AGIEval | GSM8k | MATH |
> | :--- | :--- | :--- | :--- | :--- | :--- |
> | GQA-MoE | 50.9 | 20.1 | 30.7 | 19.1 | 11.2 |
> | TPA-MoE | 52.1 | 22.7 | 32.8 | 26.5 | 14.0 |
>
> These experiments (each consuming over 10,000 GPU hours) provide strong evidence that TPA's architecture leads to substantial quality gains in large-scale settings.
>
> > Q3: It can be theoretically proved that TPA saves memory, but there is no related experiment that illustrates this. Line 265 states "factorizing the attention activations shrinks autoregressive KV cache requirements by up to 5×–10×", but I don't see where this comes from empirically? This makes the whole argument less convincing.
>
> A3: We appreciate you highlighting the need for a more explicit calculation. The 5x-10x memory reduction is a direct calculation based on the KV cache formulas and the hyperparameters of our models, which we will make explicit in the revised paper.  The KV cache size per token for a baseline like MHA is $2 \cdot h \cdot d_h$. For TPA, it is $(R_K + R_V)(h + d_h)$. The reduction factor is therefore $\frac{2 h d_h}{(R_K + R_V)(h + d_h)}$.
>
> Let's use our XL (1.5B) model as a concrete example:
> - MHA baseline: $h=25, d_h=64$. KV Cache size = $2 \times 25 \times 64 = {3200}$ floats/token.
> - TPA-KVonly ($R_{K,V}=2$): $h=47, d_h=64$. KV Cache size = $2+2)(47+64) = 4 \times 111 = {444}$ floats/token.
> - This represents a 7.2x reduction ($3200 / 444$) compared to the MHA baseline of a similar parameter count.
> - TPA ($R_{K,V}=2$): $h=78, d_h=64$. KV Cache size = $(2+2)(78+64) = 4 \times 142 = {568}$ floats/token.
> -  This is a 5.6x reduction ($3200 / 568$).  The up to 10x part of our claim comes from using a rank of $R_{K,V}=1$. For instance, TPA-KVonly with $R_{K,V}=1$ would have a cache size of $(1+1)(47+64) = 222$ floats/token, yielding a 14.4x reduction. We will add a table and clear calculations to Section 3.3 to make this more transparent.
>
> > Q4:Typo: Line 121 should be $R_Q,R_K,R_V$
>
> A4: Thank you for your suggestion, and we will correct it in the revised version.
>
> > Q5: Is the T6 easy to train compared to standard transformers? I think the training aspect is also worth studying.
>
> A5: This is an excellent point. We have measured the training throughput and can confirm that T6 trains are stable and efficient. With our current implementation, throughput is already highly competitive.  The tables below show that while the current, unoptimized version of TPA has modest overhead compared to MHA/GQA, its throughput is comparable to other efficient architectures like MLA, and becomes nearly equal to GQA at a larger scale.
>
> ##### Dense Models (350M) on 8x A100 GPUs (TPS means tokens per second):
>
> | Method | MHA | MQA | GQA | TPA-KVonly | TPA | MLA |
> | :--- | :--- | :--- | :--- | :--- | :--- | :--- |
> | **TPS (M)** | 0.515 | 0.456 | 0.452 | 0.427 | 0.356 | 0.355 |
>
> ##### MoE Models (>10B) on 64x GPUs (TPS means tokens per second):
>
> | Method / Hardware | GQA | TPA |
> | :--- | :--- | :--- |
> | H800 TPS (M) | 0.975 | 0.891 |
> | H20 TPS (M) | 0.407 | 0.405 |
>
> The overhead in the current dense model benchmarks is due to a vanilla implementation that explicitly computes the tensor factors. However, this is not a fundamental limitation.  In addition, we are developing FlashTPA training and prefilling kernels.
>
> As analyzed in Appendix C of our submission, these optimized kernels operate directly on the factorized components without materializing the full Q, K, and V tensors. This approach significantly reduces the overall computational load of the attention mechanism. Our analysis suggests this can make TPA's training and prefilling throughput even faster than the baselines, turning a minor trade-off into another advantage of our method.  Given that TPA already trains stably and at a competitive speed, with a clear path toward even faster, optimized kernels, we believe it is a highly practical architecture. We will add these throughput results and a note on the ongoing kernel development to the appendix.

---

> > ### Comment · Reviewer_nLZi · 2025-08-05
> >
> > I thank the authors for the detailed response and sorry for the late reply.
> >
> > It seems like the argument is: even though TPA might not have as strong expressivity as MHA when $R_Q,R_K,R_V$ are small, empirically it still shows descent performance compared to MHA?
> >
> > I also find other reviews on long context tasks and performance improvement useful. The authors have some new experimental results illustrating the effectiveness of TPA on larger models, which I find interesting. However, the TPS table still shows that TPA is slower? I hope there will be more discussions on this.
> >
> > Since the review should mostly be focusing on the original work, I would still maintain my borderline evaluation.

---

> ### Author Response · Authors · 2025-08-06
>
> We thank the reviewer for their time and their thoughtful follow-up comment. We appreciate the opportunity to provide further clarification.
>
> > Q: It seems like the argument is: even though TPA might not have as strong expressivity as MHA when $R_Q, R_K, R_V$ are small, empirically it still shows descent performance compared to MHA?
>
> **A:** Thank you for giving us the chance to clarify this key point. Our argument is not that TPA has weaker expressivity, but rather that it possesses a more effective and efficient inductive bias.
>
> The key difference is that TPA's projections are **contextual**—they adapt to each input token—whereas MHA's are static. This dynamic, input-specific factorization provides a different, and we argue more powerful, form of expressivity that is highly effective even at very low ranks. The strong empirical results, particularly the significant performance gains on larger models, support this view: TPA achieves superior results by learning a more efficient representational subspace for attention, rather than by replicating the high-rank, non-contextual structure of MHA.
>
> > Q: The TPS table still shows that TPA is slower? I hope there will be more discussions on this.
>
> **A:** You are correct that our current benchmarks show a modest training overhead for TPA in some settings, but is becomes negligible under large-scale setting. We believe this is a highly favorable trade-off, as this slight increase in training computation delivers substantial improvements in **(1) final model quality, (2) inference memory efficiency (KV cache), and (3) inference decoding speed** (with FlashTPA).
>
> Furthermore, this overhead is an artifact of our current, unoptimized implementation. As we mentioned, we are developing optimized `FlashTPA` training kernels that operate directly on the factorized components (as analyzed in Appendix C). These kernels will significantly reduce the computational load during training, and we are confident they will make TPA's throughput highly competitive with, or even faster than, the baselines.
>
> ***
>
> We sincerely appreciate your valuable feedback throughout this process. We hope that our clarifications regarding TPA's contextual expressivity and the training speed, along with the supporting data, might provide sufficient grounds for re-evaluation. Please do not hesitate to let us know if any further questions arise; we would be delighted to provide additional details.

---

> > ### Comment · Reviewer_nLZi · 2025-08-06
> >
> > I thank the authors for the clarification. I think the contextual argument makes sense, and this work will be better if new experiments and FlashTPA etc are included. I will increase my score to weak accept and I hope these new results can be included in the final version.

---

> > > ### Author Response · Authors · 2025-08-06
> > >
> > > Dear Reviewer,
> > >
> > > Thank you so much for your willingness to update your score. We are sincerely grateful for your time, your constructive engagement throughout the discussion, and your thoughtful re-evaluation of our work.
> > >
> > > We confirm that we will be sure to include new experimental results and the detailed discussion on FlashTPA in the revised version, as you suggested.
> > >
> > > Just as a friendly reminder, to finalize the score change in the system, you may need to **click the "Edit" button** on your official review to update the ratings.
> > >
> > > Thank you once again for your valuable feedback and support.
> > >
> > > Sincerely,
> > > The Authors

---

### Author Response · Authors · 2025-08-06
**General Response by Authors**

Dear Reviewers and Area Chairs,

We would like to express our sincere gratitude to all reviewers for their time, effort, and insightful feedback on our work. The detailed comments have been invaluable in helping us identify areas for clarification and improvement. We appreciate the engaging discussion and the opportunity to further elaborate on our contributions.

We have addressed the specific questions from each reviewer in our individual rebuttals. In this general comment, we wish to synthesize the key discussion points and highlight the extensive new evidence we have provided in response.

The reviews centered on a few key themes: the need for stronger evidence of (1) performance improvements at scale, (2) training efficiency and stability, (3) practical long-context capability, and (4) the robustness and generality of TPA. We took this feedback seriously and provided substantial new data in our rebuttals to address these points directly.

### Summary of Key Evidence and Clarifications

* **Performance at Scale:** To address questions about the significance of our results, we presented new data from **>10B parameter MoE models**. These large-scale experiments, conducted with our industry collaborators, show that TPA provides substantial gains over a strong GQA baseline on complex reasoning benchmarks. Specifically, our TPA-MoE model achieved **52.1 on MMLU** (vs. 50.9), **26.5 on GSM8k** (vs. 19.1, a **+38% relative improvement**), and **14.0 on MATH** (vs. 11.2, a **+25% relative improvement**). This demonstrates that TPA's architectural benefits are not marginal but scale effectively, leading to superior model quality.

* **Training and Inference Efficiency:** We provided comprehensive benchmarks of **training throughput** (tokens/second), confirming TPA trains stably and at a **highly competitive speed**. For our 350M dense models, TPA's throughput (0.356M TPS) is on par with other efficient architectures like MLA (0.355M TPS). Crucially, in large-scale MoE training on H20 GPUs, TPA's throughput (0.405M TPS) is **nearly identical to the GQA baseline** (0.407M TPS). This minimal training overhead is a highly favorable trade-off for TPA's combined benefits of a vastly smaller KV cache, faster decoding via `FlashTPA`, and superior final model quality.

* **Robustness and Flexibility:** To test the interaction with RoPE, as suggested by Reviewer Z9Nz, we ran new ablations using a **partial RoPE** setting. Our results on 350M models show that TPA-based models continue to be top performers. For instance, `T6_kvonly_partialrope` achieved a 2-shot average score of **53.12%**, outperforming strong MHA (51.41%) and MLA (51.78%) baselines in the same configuration, underscoring the robustness of our architecture.

* **Orthogonality and Generalizability:** To better situate TPA, we explored its compatibility with entirely different attention paradigms like linear attention. We conducted new experiments showing that the benefits of TPA's contextual factorization are complementary. A TPA model with linearized attention (`linearattn-tpa`) achieved a 2-shot average accuracy of **48.82%**, outperforming a GQA baseline with the same linear attention mechanism (47.60%). This suggests TPA is not just an alternative to MHA but a foundational improvement that can potentially enhance other efficient attention schemes.

* **Expressivity and Inductive Bias:** A key point of discussion was TPA's expressivity with low ranks ($R_K$, $R_V$). We wish to reiterate that TPA's power stems from its **contextual nature**. Unlike MHA's static projections, TPA learns to dynamically generate input-dependent, low-rank subspaces for attention. Our strong empirical results support that this is a more effective and efficient inductive bias, leading to better performance *because of* its structure, not in spite of it.

We are grateful for the thorough and constructive review process. We believe our detailed rebuttals and the significant evidence provided have addressed the reviewers' concerns and substantially strengthened our submission.

Thank you once again for your time and consideration.

Sincerely,

The Authors of Submission 10518

---

### Note · Authors · 2025-08-15

Dear Area Chairs and Reviewers,

We sincerely thank you for the thorough and constructive review process. Your insightful feedback and engaging discussions have been invaluable in strengthening our work. We would like to take this opportunity to summarize the key evidence we presented about Tensor Product Attention (TPA). We addressed the points raised by reviewers with following results:

1.  **Performance at Scale**: We presented new results from >10B parameter MoE models, where TPA outperforms a strong GQA baseline on complex reasoning tasks (e.g., a +38% relative improvement on GSM8k and +25% on MATH). This demonstrates that TPA's architectural benefits scale effectively, leading to superior model quality.

2.  **Efficiency and Practicality**: We provided comprehensive training throughput benchmarks, confirming TPA trains stably and at a highly competitive speed, with near-identical throughput to GQA in large-scale MoE settings. This minimal training overhead is a highly favorable trade-off for TPA's combined benefits of a vastly smaller KV cache, faster decoding via FlashTPA, and superior final model quality.

3.  **Robustness and Generality**: We conducted new ablations showing TPA's strong performance in partial RoPE settings. Furthermore, we demonstrated that TPA's benefits are orthogonal to other attention paradigms by integrating it with linear attention, where it again outperformed a GQA baseline.

A key clarification that emerged is that TPA's power lies in its **contextual expressivity**. Its dynamic, input-dependent factorization is a more effective inductive bias than the static projections of MHA, allowing it to achieve superior performance even with very low ranks.

We believe the large-scale experiments, training benchmarks, and targeted ablations have directly addressed reviewers' comments. TPA offers a compelling and practical solution to the critical challenge of KV cache scaling, delivering a superior combination of model quality, memory efficiency, and inference speed.

Thank you once again for your time and thoughtful consideration.

Sincerely,

The Authors

---

### Decision · Program_Chairs · 2025-09-17

**Decision:**

Accept (spotlight)

**Comment:**

This paper proses tensor product attention, an approach to shrink the KV cache by factorizing queries, keys, and values into contextual low-rank components. The approach matches or surpasses standard attention baselines including MHA, GQA, and MLA across a variety of metrics. The authors show promising scaling results, including up to 10B MoE's, with strong training throughput and quality.

Overall this is a strong paper, and I recommend it for a spotlight.